# Conditional Bayesian Quadrature

**Zonghao Chen**[1,*]          **Masha Naslidnyk**[1,*]          **Arthur Gretton**[2]          **François-Xavier Briol**[3]

[1]Department of Computer Science, University College London, London, UK
[2]Gatsby Computational Neuroscience Unit, University College London, London, UK
[3]Department of Statistical Science, University College London, London, UK

## Abstract

We propose a novel approach for estimating conditional or parametric expectations in the setting where obtaining samples or evaluating integrands is costly. Through the framework of probabilistic numerical methods (such as Bayesian quadrature), our novel approach allows to incorporates prior information about the integrands especially the prior smoothness knowledge about the integrands and the conditional expectation. As a result, our approach provides a way of quantifying uncertainty and leads to a fast convergence rate, which is confirmed both theoretically and empirically on challenging tasks in Bayesian sensitivity analysis, computational finance and decision making under uncertainty.

## 1 INTRODUCTION

This paper considers the computational challenge of estimating certain intractable expectations which arise in machine learning, statistics, and beyond. Given a function $f : \mathcal{X} \times \Theta \rightarrow \mathbb{R}$, we are interested in estimating *conditional expectations* (sometimes also called parametric expectations) $I : \Theta \rightarrow \mathbb{R}$ uniformly over the parameter space $\Theta$, where:

$$I(\theta) = \mathbb{E}_{X \sim \mathbb{P}_\theta}[f(X, \theta)] = \int_\mathcal{X} f(x, \theta) \mathbb{P}_\theta(\mathrm{d}x),$$

and $\{\mathbb{P}_\theta\}_{\theta \in \Theta}$ is a family of distributions on the integration domain $\mathcal{X}$. We will assume that $I(\theta)$ is sufficiently smooth in $\theta$ so that $I(\theta), I(\theta')$ are similar given close enough parameters $\theta, \theta'$, but that $I$ is not available in closed-form and must be approximated through samples and function evaluations.

The computational challenge of approximating conditional expectations arises in many fields. It must be tackled when calculating tail probabilities in rare-event simulation [Tang, 2013], and when computing moment generating, characteristic, or cumulative distribution functions [Giles et al., 2015, Krumscheid and Nobile, 2018]. It also arises when computing the conditional value at risk or various valuations of options [Longstaff and Schwartz, 2001, Alfonsi et al., 2022], for Bayesian sensitivity analysis [Lopes and Tobias, 2011, Kallioinen et al., 2021], or even more broadly for scientific sensitivity analysis; see for example Sobol indices [Sobol, 2001]. Conditional expectations $I(\theta)$ are also often computed as an intermediate quantity. For example, given $\phi : \mathbb{R} \rightarrow \mathbb{R}$ and some probability distribution $\mathbb{Q}$ on $\Theta$, we are often interested in the *nested expectation* given by $\mathbb{E}_{\theta \sim \mathbb{Q}}[\phi(I(\theta))]$ [Hong and Juneja, 2009, Rainforth et al., 2018]. This problems comes about when computing the expected information gain in Bayesian experimental design [Chaloner and Verdinelli, 1995], and for computing the expected value of partial perfect information in health economics [Heath et al., 2017].

Methods for computing $I(\theta)$ generally select $T$ parameter values $\theta_1, \cdots, \theta_T \in \Theta$, then simulate $N$ realisations from each corresponding probability distribution $\mathbb{P}_{\theta_1}, \cdots, \mathbb{P}_{\theta_T}$ at which they evaluate the integrand $f$, leading to a total of $NT$ evaluations. The usual approach is to use classical Monte Carlo methods to estimate $I(\theta_1), \cdots, I(\theta_T)$, but in many applications we are also interested in estimating either $I(\theta)$ for a fixed $\theta \notin \{\theta_1, \cdots, \theta_T\}$, or $I(\theta)$ uniformly over $\theta \in \Theta$. As a result, a second step combining the estimates of $I(\theta_1), \cdots, I(\theta_T)$ is often required to complete the task.

The most straightforward approach to estimating conditional expectation is importance sampling [Glynn and Igelhart, 1989, Madras and Piccioni, 1999, Tang, 2013, Demange-Chryst et al., 2022], where $I(\theta)$ is estimated by weighting function evaluations to account for the fact that the samples were not obtained from $\mathbb{P}_\theta$ but from the importance distributions $\mathbb{P}_{\theta_1}, \cdots, \mathbb{P}_{\theta_T}$. Unfortunately, this approach is only applicable when $f$ does not depend on $\theta$ (otherwise new expensive function evaluations are needed), and it is usually difficult to identify importance distributions that can

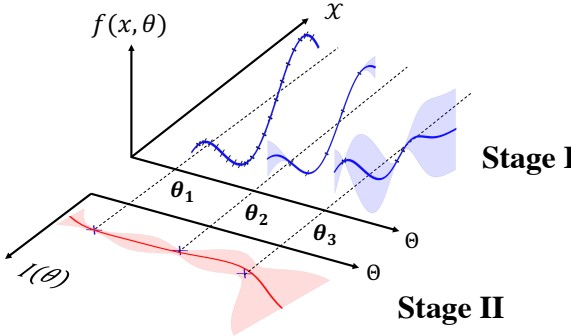

**Stage I**

**Stage II**

**Figure 1:** Illustration for *conditional Bayesian quadrature* (CBQ) in Section 3. The first stage gives a GP posterior of $f(x, \theta)$ for each $\theta \in \{\theta_1, \cdots, \theta_T\}$, which are then integrated to give $\hat{I}_{\text{BQ}}(\theta_1), \cdots, \hat{I}_{\text{BQ}}(\theta_T)$. The second stage then combines all BQ estimates from the first stage to give a GP posterior of $I(\theta)$: $\hat{I}_{\text{CBQ}}(\theta)$. All shared areas represent Bayesian quantification of uncertainty.

lead to an accurate estimator for small $N$ and $T$. Alternatively, least-squares Monte Carlo [Longstaff and Schwartz, 2001, Alfonsi et al., 2022] first estimates $I(\theta_1), \cdots, I(\theta_T)$ through Monte Carlo, then estimates $I(\theta)$ through linear or polynomial regression based on these estimates. These methods are therefore dependent on the accuracy of the Monte Carlo estimators and the regression method.

Overall and in addition, there are two main limitations which all of these existing methods suffer from. Firstly, they are very sample-intensive; i.e. they require a relatively large number of function evaluations (i.e. $N$ and $T$) to reach a given level of accuracy, which makes them infeasible if sampling or evaluating the integrand is expensive. Secondly, obtaining a finite-sample quantification of uncertainty for $I(\theta)$ is often infeasible. This is a significant limitation for challenging integration problems, for which we would ideally like to know how accurate our estimator is.

To tackle these limitations, we propose a novel algorithm called *conditional Bayesian quadrature* (CBQ). The name comes from the fact that our approach extends the Bayesian quadrature algorithm [Diaconis, 1988, O'Hagan, 1991, Rasmussen and Ghahramani, 2003, Briol et al., 2019] to the computation of conditional expectations. As such, CBQ falls in the line of work on probabilistic numerical methods [Hennig et al., 2015, Cockayne et al., 2019, Oates and Sullivan, 2019, Hennig et al., 2022]. Our algorithm is based on a hierarchical Bayesian model consisting of two-stages of Gaussian process regression, and leads to a univariate Gaussian posterior distribution on $I(\theta)$ whose mean and variance are parametrised by $\theta$. See Figure 1 for an illustration.

This approach allows us to mitigate the two main limitations of existing methods. Firstly, we show both theoretically and empirically that our method converges rapidly to the true value and is hence more sample efficient than baselines. This

result holds under mild smoothness conditions on $f$ and $I(\theta)$ whenever the dimension of $\mathcal{X}$ and $\Theta$ is not too large. As a result, a desired accuracy can be reached with smaller $N$ and $T$, and the method will therefore be preferable for expensive problems. Secondly, the fact that we have an entire posterior distribution on $I(\theta)$ allows us to provide finite-sample Bayesian quantification of uncertainty.

The remainder of the paper is structured as follows: In Section 2, we review existing methods for computing conditional expectations and Bayesian quadrature. In Section 3, we formalise our novel *conditional Bayesian quadrature* algorithm. In Section 4, we establish the theoretical convergence of our method. In Section 5, we provide empirical results and compare with baseline methods on challenging tasks in Bayesian sensitivity analysis, computational finance and decision making under uncertainty.

## 2 BACKGROUND

We aim to compute the conditional expectation $I(\theta) = \mathbb{E}_{X \sim \mathbb{P}_\theta}[f(X, \theta)]$, where we assume that $\mathcal{X} \subseteq \mathbb{R}^d$, $\Theta \subseteq \mathbb{R}^p$, and $f(\cdot, \theta)$ is in $\mathcal{L}^2(\mathbb{P}_\theta) := \{h : \mathcal{X} \to \mathbb{R} : \mathbb{E}_{X \sim \mathbb{P}_\theta}[h(X)^2] < \infty\}$, the space of square-integrable functions with respect to $\mathbb{P}_\theta$ for all $\theta \in \Theta$. The latter is a minimal assumption which ensures that Monte Carlo estimators satisfy the central limit theorem. Our observations and corresponding functional evaluations are:

$$\theta_{1:T} := [\theta_1, \cdots, \theta_T]^\top \in \Theta^T,$$
$$x_{1:N}^t := [x_1^t, \cdots, x_N^t]^\top \in \mathcal{X}^N,$$
$$f(x_{1:N}^t, \theta_t) := [f(x_1^t, \theta_t), \cdots, f(x_N^t, \theta_t)]^\top \in \mathbb{R}^N,$$

for all $t \in \{1, \cdots, T\}$, where we use square brackets to indicate vectors. This could straightforwardly be extended to allow a different number of samples $N_t$ per parameter value $\theta_t$, but we do not consider this case in order to simplify notations throughout. In this section, we will review existing methods for computing conditional expectations and the core ingredient for our method: Bayesian quadrature.

### 2.1 EXISTING METHODS FOR COMPUTING CONDITIONAL EXPECTATIONS

Existing methods fall into two categories: sampling-based methods and regression-based methods. Throughout, we will assume that $x_{1:N}^t \sim \mathbb{P}_{\theta_t}$ for all $t \in \{1, \cdots, T\}$.

**Sampling-based Methods** We can construct a *Monte Carlo* (MC) estimator [Robert et al., 1999] for $I(\theta_t)$ through $\hat{I}_{\text{MC}}(\theta_t) := \frac{1}{N} \sum_{i=1}^N f(x_i^t, \theta_t)$. Unfortunately, we cannot estimate $I(\theta)$ for $\theta \notin \{\theta_1, \cdots, \theta_T\}$, and we can only use $N$ rather than $NT$ points to estimate each $I(\theta_t)$, making MC inappropriate for our task. A more suitable alternative is *importance sampling* (IS). Assume $\mathbb{P}_\theta$ has a Lebesgue density $p_\theta : \mathcal{X} \to \mathbb{R}$ which has full support on $\mathcal{X}$ for

all $\theta \in \Theta$, and the integrand does not depend on $\theta$ (i.e. $f(x, \theta) = f(x)$). Then the IS estimator is able to make use of all $NT$ samples and can estimate $I(\theta)$ for any parameter $\theta \in \Theta$: $\hat{I}_{\text{IS}}(\theta) := \frac{1}{T} \sum_{t=1}^{T} \sum_{i=1}^{N} p_\theta(x_i^t)/p_{\theta_t}(x_i^t) f(x_i^t)$. The choice of importance distributions $\mathbb{P}_{\theta_1}, \cdots, \mathbb{P}_{\theta_T}$ has been studied in Glynn and Igelhart [1989], Madras and Piccioni [1999], Tang [2013], but alternatives beyond this parametric family of distributions could also be used [Demange-Chryst et al., 2022].

**Regression-based Methods** The main regression-based method is least-squares Monte Carlo (LSMC) [Longstaff and Schwartz, 2001], which is a two-stage approach. Stage 1 consists of computing MC estimators $\hat{I}_{\text{MC}}(\theta_1), \cdots, \hat{I}_{\text{MC}}(\theta_T)$, whilst stage 2 consists of estimating $I(\theta)$ through linear or polynomial regression based on the estimates from stage 1. Other non-parametric regression method could be used though; for kernel ridge regression [Han et al., 2009, Hu and Zastawniak, 2020], we will refer to the algorithm as kernelised least-squares Monte Carlo (KLSMC). Note that KLSMC is identical to standard estimators for conditional kernel mean embeddings based on vector-valued kernel ridge regression and can be recognised as a generalisation of the kernel mean shrinkage estimators of Muandet et al. [2016], Chau et al. [2021]. Clearly, both the performance and computational cost of these estimators will depend on the regression method. LSMC costs $\mathcal{O}(TN + p^3)$ with $p$ being the order of polynomial, whereas KLSMC costs $\mathcal{O}(TN + T^3)$. On the other hand, KLSMC is more flexible and will outperform LSMC when $I(\theta)$ cannot be approximated well by a low-order polynomial.

**Other Related Work** Alternative approaches for estimating $I(\theta)$ are based on multi-task or meta- learning [Xi et al., 2018, Gessner et al., 2020, Sun et al., 2023a,b]. This line of research tends to assume that several related expectations need to be computed, and the relationship between these expectations is encoded through a vector-valued RKHS, or that they are independent draws from a set of tasks. Notably, they do not explicitly encode properties of the mapping $\theta \mapsto I(\theta)$, and will therefore be sub-optimal for our setting. Multilevel Monte Carlo methods are also popular in estimating expensive expectations, by combining samples from multiple levels of resolution [Giles et al., 2015]. However, they are not able to estimate new integrals $I(\theta^*)$ or $I(\theta)$ uniformly over $\theta \in \Theta$.

## 2.2 BAYESIAN QUADRATURE

In this section, we present Bayesian quadrature, the foundational component of our approach. Consider the expectation $I = \mathbb{E}_{X \sim \mathbb{P}}[f(X)]$ of some function $f : \mathcal{X} \to \mathbb{R}$, where we emphasise that neither $f$ nor $\mathbb{P}$ depend on $\theta$ in this subsection. In Bayesian quadrature (BQ) [Diaco-

nis, 1988, O'Hagan, 1991, Rasmussen and Ghahramani, 2003, Briol et al., 2019], we begin by positing a Gaussian process (GP) prior on $f$. We will denote this prior $\mathcal{GP}(m_\mathcal{X}, k_\mathcal{X})$, where $m_\mathcal{X} : \mathcal{X} \to \mathbb{R}$ is the mean function and $k_\mathcal{X} : \mathcal{X} \times \mathcal{X} \to \mathbb{R}$ is the covariance (or reproducing kernel) function. These two functions fully characterise the distribution, and can be used to encode prior knowledge about smoothness, periodicity, or sparsity of $f$. Once a GP prior has been selected, we condition on noiseless function evaluations $f(x_{1:N}) = [f(x_1), \cdots, f(x_N)]^\top$ for $x_{1:N} \in \mathcal{X}^N$. This leads to a posterior GP on $f$, which induces a univariate Gaussian posterior distribution $\mathcal{N}(\hat{I}_{\text{BQ}}, \sigma_{\text{BQ}}^2)$ on $I$, where:

$$\hat{I}_{\text{BQ}} = \mathbb{E}_{X \sim \mathbb{P}}[m_\mathcal{X}(X)] + \mu(x_{1:N})^\top (k_\mathcal{X}(x_{1:N}, x_{1:N}) +$$
$$\lambda_\mathcal{X} \text{Id}_N)^{-1} (f(x_{1:N}) - m_\mathcal{X}(x_{1:N})),$$
$$\sigma_{\text{BQ}}^2 = \mathbb{E}_{X, X' \sim \mathbb{P}}[k_\mathcal{X}(X, X')] - \mu(x_{1:N})^\top (k_\mathcal{X}(x_{1:N}, x_{1:N})$$
$$+ \lambda_\mathcal{X} \text{Id}_N)^{-1} \mu(x_{1:N}).$$

Here, $\text{Id}_N$ is an $N$-dimensional identity matrix and $\lambda_\mathcal{X} \geq 0$ is a regularisation parameter, often called "nugget" or "jitter", which, although not essential from a statistical viewpoint, is often used to ensure the matrix can be numerically inverted [Ababou et al., 1994, Andrianakis and Challenor, 2012].

The function $\mu(x) = \mathbb{E}_{X \sim \mathbb{P}}[k_\mathcal{X}(X, x)]$ is known as the kernel mean embedding [Muandet et al., 2017] of the distribution $\mathbb{P}$ and $\mathbb{E}_{X, X' \sim \mathbb{P}}[k_\mathcal{X}(X, X')]$ is known as the initial error. To implement BQ, we need the kernel mean embedding and the initial error to be available in closed-form, which is a rather strong requirement and does not hold for all pairs of kernel and distribution. Fortunately, there are multiple solutions for this problem; see Table 1 in [Briol et al., 2019], [Nishiyama and Fukumizu, 2016], the `ProbNum` package [Wenger et al., 2021], or Stein reproducing kernels [Anastasiou et al., 2023]. A discussion is provided in Appendix B.1.

The posterior mean $\hat{I}_{\text{BQ}}$ provides a point estimate for $I$ whilst the posterior variance $\sigma_{\text{BQ}}^2$ gives a notion of uncertainty for $I$ which arises due to having only observed $f$ at $N$ points. For BQ to be well-calibrated and the posterior variance $\sigma_{\text{BQ}}^2$ to be meaningful, we need to select the GP prior and all associated hyperparameters carefully; this is usually achieved through empirical Bayes [Casella, 1985]. A detailed discussion on hyperparameter selection is provided in Appendix B.2. It is noteworthy that BQ does not impose restrictions on how $x_{1:N}$ is selected, and as such does not require independent realisations from $\mathbb{P}$. In fact, a number of active learning approaches have proven popular, see Gunter et al. [2014], Gessner et al. [2020].

The convergence rate of the BQ estimator has been studied extensively [Briol et al., 2019, Kanagawa and Hennig, 2019, Wynne et al., 2021] and is particularly fast for low- to mid-dimensional smooth integrands. This has to be contrasted

with the computational cost, which is inherited from GP regression and is $\mathcal{O}(N^3)$. For this reason, BQ has principally been applied to problems where sampling or evaluating the integrand is very expensive and usually only a small number of samples are available (i.e. small $N$). Examples range from differential equation solvers [Kersting and Hennig, 2016], neural ensemble search [Hamid et al., 2023], variational inference [Acerbi, 2018] and simulator-based inference [Bharti et al., 2023] to applications in computer graphics [Marques et al., 2013, Xi et al., 2018], cardiac modelling [Oates et al., 2017] and tsunami modelling [Li et al., 2022]. For cheaper problems, Jagadeeswaran and Hickernell [2019], Karvonen and Sarkka [2018], Karvonen et al. [2018] propose BQ methods where the computational cost is much lower, but these are applicable only with specific point sets $x_{1:N}$ and distributions $\mathbb{P}$. Hayakawa et al. [2023] also studies Nyström-type of approximations, whilst Adachi et al. [2022] studies parallelisation techniques. Finally, several alternatives with linear cost in $N$ have also been proposed using tree-based [Zhu et al., 2020] or neural-network [Ott et al., 2023] models, but these tend to require approximate inference methods such as Laplace approximations or Markov chain Monte Carlo.

# 3 METHODOLOGY

*Conditional Bayesian quadrature* (CBQ) provides a Bayesian hierarchical model for $I(\theta^*)$ for any $\theta^* \in \Theta$, and the posterior mean of this hierarchical model is called the CBQ estimator. The algorithm falls into the realm of regression-based methods and can therefore be expressed in two stages:

- **Stage 1:** Compute $\hat{I}_{\text{BQ}}(\theta_{1:T}), \sigma^2_{\text{BQ}}(\theta_{1:T})$ to obtain the BQ posterior mean and variance on $I(\theta_1), \ldots, I(\theta_T)$.

- **Stage 2:** Perform GP regression over $I(\theta)$ using the outputs of stage 1. The posterior mean $\hat{I}_{\text{CBQ}}(\theta)$ is the CBQ estimator for $I(\theta)$, and the variance $k_{\text{CBQ}}(\theta, \theta)$ quantifies uncertainty.

An illustrative figure is provided in Figure 1. This two-stage algorithm can also be summarised using the directed acyclic graph in Figure 2, where the first stage corresponds to the part of the model inside the largest plate, and the second stage corresponds to the remainder of the graph. The CBQ posterior mean and covariance are given by

$$\hat{I}_{\text{CBQ}}(\theta) := m_\Theta(\theta) + k_\Theta(\theta, \theta_{1:T})\big(k_\Theta(\theta_{1:T}, \theta_{1:T})$$
$$+ \text{diag}(\lambda_\Theta + \sigma^2_{\text{BQ}}(\theta_{1:T}))\big)^{-1}(\hat{I}_{\text{BQ}}(\theta_{1:T}) - m_\Theta(\theta_{1:T})),$$
$$k_{\text{CBQ}}(\theta, \theta') := k_\Theta(\theta, \theta') - k_\Theta(\theta, \theta_{1:T})\big(k_\Theta(\theta_{1:T}, \theta_{1:T})$$
$$+ \text{diag}(\lambda_\Theta + \sigma^2_{\text{BQ}}(\theta_{1:T}))\big)^{-1}k_\Theta(\theta_{1:T}, \theta')$$

where the observations $\{x^t_{1:N}, f(x^t_{1:N}, \theta_t)\}^T_{t=1}$ enters implicitly through $\hat{I}_{\text{BQ}}(\theta_{1:T})$. The terms $\hat{I}_{\text{BQ}}(\theta_t)$ and $\sigma^2_{\text{BQ}}(\theta_t)$ are the BQ posterior mean and variance for $I(\theta_t)$, $\text{diag}(\lambda_\Theta +$

$\sigma^2_{\text{BQ}}(\theta_{1:T})))$ is the diagonal matrix with vector $\lambda_\Theta + \sigma^2_{\text{BQ}}(\theta_{1:T}))$ on the diagonal and where $\lambda_\Theta \geq 0$ acts as a regulariser. We also have $m_\Theta : \Theta \to \mathbb{R}$ and $k_\Theta : \Theta \times \Theta \to \mathbb{R}$ which are the prior mean and covariance for the stage 2 GP. Similarly to BQ, the "quadrature" terminology is justified since $\hat{I}_{\text{CBQ}}(\theta) := \sum^T_{t=1} \sum^N_{i=1} w^{\text{CBQ}}_{i,t} f(x^t_i, \theta_t)$ for some weights $w^{\text{CBQ}}_{i,t} \in \mathbb{R}$ when $m_\Theta(\theta) = 0$.

The first stage corresponds to the BQ procedure highlighted in Section 2.2: we model $f(\cdot, \theta_t)$ with independent $\text{GP}(m^t_{\mathcal{X}}, k^t_{\mathcal{X}})$ priors, condition on observations $f(x^t_{1:N}, \theta_t)$, and consider the posterior distribution on $I(\theta_t)$ for all $t \in \{1, \ldots, T\}$. We therefore require access to closed-form expressions for each of the $T$ kernel mean embeddings and initial errors (see discussion in Appendix B.1 on the pairs of kernel and distribution that have a closed form kernel mean embedding). Note that at this stage, we do not share any samples across the estimators of $I(\theta_1), \ldots, I(\theta_T)$.

In the second stage, we place a $\text{GP}(m_\Theta, k_\Theta)$ prior on $I : \Theta \to \mathbb{R}$, and assume $\hat{I}_{\text{BQ}}(\theta_t)$ are noisy evaluations of $I(\theta_t)$: $\hat{I}_{\text{BQ}}(\theta_t) = I(\theta_t) + \varepsilon_t$, where the noise terms $\varepsilon_t$ are independent zero-mean Gaussian noise with variance $\sigma^2_{\text{BQ}}(\theta_t)$ for all $t \in \{1, \ldots, T\}$. Note that $\hat{I}_{\text{BQ}}(\theta_t)$ is a deterministic function of independent samples $\theta_t, x^t_1, \cdots, x^t_N$ across $t = 1, \cdots, T$, so $\hat{I}_{\text{BQ}}(\theta_1), \ldots, \hat{I}_{\text{BQ}}(\theta_T)$ are also independent. As the variance $\epsilon_t$ is input-dependent, this corresponds to heteroscedastic GP regression [Le et al., 2005]. We now briefly comment on the choice of prior and likelihood in this second stage:

- The $\text{GP}(m_\Theta, k_\Theta)$ prior can be used to encode prior knowledge about how the expectation $I(\theta)$ varies with the parameter $\theta$. Typically, the stronger this prior information, the faster the CBQ estimator's convergence rate will be; this statement will be made formal in Section 4.

- The likelihood for the heteroscedastic GP is directly inherited from the BQ posteriors in the first stage: the posterior on $I(\theta_t)$ is a univariate normal with mean $\hat{I}_{\text{BQ}}(\theta_t)$ and variance $\sigma^2_{\text{BQ}}(\theta_t)$. As expected, when the number of samples $N$ grows, the BQ variance $\sigma^2_{\text{BQ}}(\theta_t)$ will decrease, indicating that we are more certain about $I(\theta_t)$. This is then directly taken into account in stage 2. Note that heteroscedasticity has previously been shown to be common in practice for LSMC [Fabozzi et al., 2017].

CBQ is closely related to LSMC and KLSMC as it simply corresponds to different choices for the two stages. The main difference is in stage 1, where we use BQ rather than MC. This is where we expect the greatest gains for our approach due to the fast convergence rate of BQ estimators (this will be confirmed in Section 4). For stage 2, we use heteroscedastic GP regression rather than polynomial or kernel ridge regression. As such, the second stage of KLSMC and CBQ is identical up to a minor difference in the way in which the Gram matrix $k_\Theta(\theta_{1:T}, \theta_{1:T})$ is regularised before

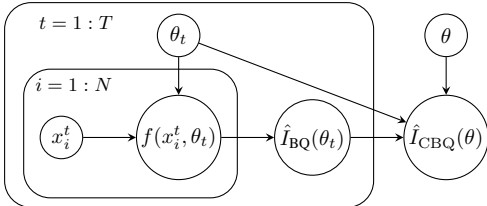 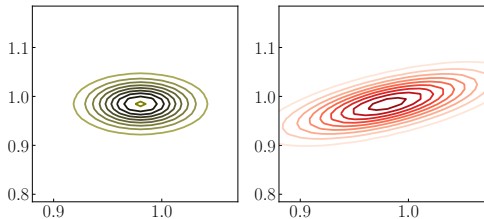

**Figure 2:** *Illustration of CBQ.* **Left:** Directed acyclic graph representation. Circle nodes indicate random variables and rectangles correspond to independent replications over indices. **Right:** BQ and CBQ posteriors on $I(\theta_{1:2}) = [I(\theta_1), I(\theta_2)]^\top$ for $\theta_1 \approx \theta_2$. Unlike BQ, the CBQ posterior accounts for the relation between the two quantities.

inversion. Finally, one significant advantage of CBQ over LSMC and KLSMC is that it is a fully Bayesian model, meaning that we obtain a posterior distribution on $I(\theta)$ for any $\theta \in \Theta$.

The total computational cost of our approach is $\mathcal{O}(TN^3 + T^3)$ due to the need to compute $T$ BQ estimators in the first stage and heteroscedastic GP regression in the second stage. Approximate GP approaches such as Titsias [2009] could *not* be used to reduce the cost because they introduce an additional layer of approximation which will slow down the convergence rate of CBQ. The cost of CBQ is higher than the cost of $\mathcal{O}(TN + p^3)$ or $\mathcal{O}(TN + T^3)$ of LSMC and KLSMC respectively, but as we will see in Section 5, the higher computational cost of CBQ will be offset competitive by faster convergence (derived in Theorem 1) and is more competitive compared to baseline methods (see Section 5). Additionally in many applications (such as the SIR model in Section 5), the cost of evaluating the integrand will be much larger than the cost of estimation methods, so data-efficient method like CBQ will be more efficient overall.

Interestingly, CBQ also provides us with a joint Gaussian posterior on the expectation at $\theta_1^*, \ldots, \theta_{T_{\text{Test}}}^* \in \Theta$ which has mean vector $\hat{I}_{\text{CBQ}}(\theta_{1:T_{\text{Test}}}^*)$ and covariance matrix $k_{\text{CBQ}}(\theta_{1:T_{\text{Test}}}^*, \theta_{1:T_{\text{Test}}}^*)$. This can be computed at an $\mathcal{O}(T^2 T_{\text{test}})$ cost, and is illustrated in the right plot of Figure 2 on a synthetic example from Section 5; as observed, CBQ takes into account of covariances between test points in that the integral value will be similar for similar parameter values, whereas standard BQ treats each integral value independently.

A natural alternative would be to place a GP prior directly on $(x, \theta) \mapsto f(x, \theta)$ and condition on all $N \times T$ observations. The implied distribution on $I(\theta_1), \ldots, I(\theta_T)$ would also be a multivariate Gaussian distribution. This approach coincides with the multi-output Bayesian quadrature (MOBQ) approach of Xi et al. [2018] where multiple integrals are considered simultaneously. However, the computational cost of MOBQ is $\mathcal{O}(N^3 T^3)$, due to fitting a GP on $NT$ observations, and quickly becomes intractable as $N$ or $T$ grow. A further comparison of BQ and MOBQ can be found in Appendix C.5. The same holds true if $f$ does not depend on $\theta$, in which case the task reduces to the conditional mean

process studied in Proposition 3.2 of Chau et al. [2021], and when $T = 1$, we recover standard Bayesian quadrature.

**Hyperparameters** The hyperparameter selection for CBQ boils down to the choice of GP interpolation hyperparameters at stage 1 and the choice of GP regression hyperparameters at stage 2. To simplify this choice, we renormalise all our function values before performing GP regression and interpolation. This is done by first subtracting the empirical mean and then dividing by the empirical standard deviation. The choice of covariance functions $k_\mathcal{X}$ and $k_\Theta$ is made on a case-by-case basis in order to both encode properties we expect the target functions to have, but also to ensure that the corresponding kernel mean is available in closed-form (see Appendix B.1). Once this is done, we typically still need to make a choice of hyperparameters for both kernel: lengthscales $l_\mathcal{X}$, $\ell_\Theta$ and amplitudes $A_\mathcal{X}$, $A_\Theta$. We also need to select the regularizer $\lambda_\mathcal{X}$, $\lambda_\Theta$. $\lambda_\mathcal{X}$ is fixed to be 0 as suggested by Theorem 1, and the rest of the hyperparameters are selected through empirical Bayes, which consists of maximising the log-marginal likelihood. For more details on hyperparameter selection, please refer to Appendix B.2.

## 4 THEORETICAL RESULTS

Our main theoretical result in Theorem 1 below guarantees that CBQ is able to recover the true value of $I(\theta)$ when $N$ and $T$ grow. The result of this theorem depends on the smoothness of the problem. We will say a function has smoothness $s$ if it is in the Sobolev space $\mathcal{W}^{s,2}$ of functions with at least $s$ (weak) derivatives that are square Lebesgue-integrable [Adams and Fournier, 2003]. For a multi-index $\alpha = (\alpha_1, \ldots \alpha_p) \in \mathbb{N}^p$, by $D_\theta^\alpha f$ we denote the $|\alpha| = \sum_{i=1}^d \alpha_i$ order weak derivative of a function $f$ on $\Theta$. Similarly, we will say a kernel has smoothness $s$ whenever its corresponding RKHS is a space of functions of smoothness $s$. This is for example the case of the Matérn$-\nu$ kernel in dimension $d$ whenever $s = \nu + d/2$, defined as $k_\nu(x, y) = \frac{\eta}{\Gamma(\nu)2^{\nu-1}}(\frac{\sqrt{2\nu}}{l}\|x - y\|_2)^\nu K_\nu(\frac{\sqrt{2\nu}}{l}\|x - y\|_2)$ where $K_\nu$ is the modified Bessel function of the second kind and $\eta, l > 0$ are hyperparameters.

**Theorem 1.** *Let* $x \mapsto f(x, \theta)$ *be a function of smoothness* $s_f > d/2$, *and* $\theta \mapsto f(x, \theta)$ *be*

*a function of smoothness $s_I > p/2$ such that $\sup_{\theta \in \Theta} \max_{|\alpha| < s_I} \|D_\theta^\alpha f(\cdot, \theta)\|_{\mathcal{W}^{s_I, 2}(\mathcal{X})} < \infty$. Suppose the following assumptions hold:*

*A1 The domains $\mathcal{X} \subset \mathbb{R}^d$ and $\Theta \subset \mathbb{R}^p$ are open, convex, and bounded.*

*A2 The parameters and samples satisfy: $\theta_{1:T} \sim \mathbb{Q}$, and $x_{1:N}^t \sim \mathbb{P}_{\theta_t}$ for all $t \in \{1, \ldots, T\}$.*

*A3 $\mathbb{Q}$ has density $q$ such that $\inf_{\theta \in \Theta} q(\theta) > 0$ and $\sup_{\theta \in \Theta} q(\theta) < \infty$, and $\mathbb{P}_\theta$ has density $p_\theta$ such that $\theta \mapsto p_\theta(x)$ is of smoothness $s_I > p/2$, and for any $\theta \in \Theta$, it holds that $\inf_{\theta \in \Theta, x \in \mathcal{X}} p_\theta(x) > 0$ and $\sup_{\theta \in \Theta} \max_{|\alpha| \le s} \|D_\theta^\alpha p_\theta(x)\|_{\mathcal{L}^\infty(\mathcal{X})} < \infty$.*

*A4 The kernels $k_\mathcal{X}$ and $k_\Theta$ are of smoothness $s_\mathcal{X} \in (d/2, s_f]$ and $s_\Theta \in (p/2, s_I]$ respectively.*

*A5 The regularisers satisfy $\lambda_\mathcal{X} = 0$ and $\lambda_\Theta = \mathcal{O}(T^{\frac{1}{2}})$.*

*Then, we have that for any $\delta \in (0, 1)$ there is an $N_0 > 0$ such that for any $N \ge N_0$ with probability at least $1 - \delta$ it holds that*

$$\left\| \hat{I}_{\mathrm{CBQ}} - I \right\|_{\mathcal{L}^2(\Theta)} \le C_0(\delta) N^{-\frac{s_\mathcal{X}}{d} + \varepsilon} + C_1(\delta) T^{-\frac{1}{4}},$$

*for any arbitrarily small $\varepsilon > 0$, and the constants $C_0(\delta) = \mathcal{O}(1/\delta)$ and $C_1(\delta) = \mathcal{O}(\log(1/\delta))$ are independent of $N, T, \varepsilon$.*

To prove the result, we represent the CBQ estimator as a *noisy importance-weighted kernel ridge regression* (NIW-KRR) estimator. Then, we extend convergence results for the *noise-free* IW-KRR estimator established in Gogolashvili et al. [2023, Theorem 4] to bound Stage 2 error in terms of the error in Stage 1, which in turn we bound via results on the convergence of GP interpolation from Wynne et al. [2021]. See Appendix A for the detailed proof.

We now briefly discuss our assumptions. Many of these were simplified to improve readability, in which case we highlight possible generalisations. A1 is used to guarantee the points eventually cover the domain, and could straightforwardly be generalised to any open and bounded domain with Lipschitz boundary satisfying an interior cone condition; see Kanagawa et al. [2020], Wynne et al. [2021]. A2 ensures $\theta_{1:T}$ and $x_{1:N}^t$ cover $\mathcal{X}$ and $\Theta$ sufficiently fast in probability as $N$ and $T$ grow. The assumption on the point sets could also be straightforwardly generalised to active learning designs or grids following existing work on BQ convergence [Kanagawa and Hennig, 2019, Kanagawa et al., 2020, Wynne et al., 2021]. A3 ensures that the points will fill $\mathcal{X}$. A4 guarantees that our first and second stage GPs have the right level of regularity for the problem, although the range of smoothness values could be significantly extended following the approach of Kanagawa et al. [2020]. For simplicity, we also implicitly assume that the kernel hyperparameters (such as lengthscales and amplitudes) are known, but this could be extended to estimation in bounded sets; see [Teckentrup, 2020]. Finally, A5 requires $\lambda_\mathcal{X} = 0$,

but this could be relaxed at the cost of slowing down convergence (see Appendix A). In contrast, growing $\lambda_\Theta > 0$ in $T$ is natural since we work in a bounded domain and we expect the conditioning of the Gram matrix to become worse as $T \to \infty$.

We are now ready to discuss the implications of the theorem. Firstly, the result is expressed in probability to account for randomness in $\theta_{1:T}$ and $x_{1:N}^t$, and provides a rate of $\mathcal{O}(T^{-1/4} + N^{-s_\mathcal{X}/d + \varepsilon})$. We can see that growing $N$ will only help up to some extent (as the second terms approaches zero fast), but that growing $T$ is essential to ensure convergence. This is intuitive since we cannot expect to approximate $I(\theta)$ uniformly simply by increasing $N$ at some fixed points in $\Theta$. Despite this, we will see in Section 5 that increasing $N$ will be essential to improving performance in practice. The rate in $N$ will typically be very fast for smooth targets, but is significantly slowed down for large $d$, demonstrating that our method is mostly suitable for low-to mid-dimensional problems, a common feature shared by Bayesian quadrature based algorithms [Briol et al., 2019, Frazier, 2018]. There have been some attempts to scale BQ/CBQ to high dimensions; for example in section 5.4 of Briol et al. [2019] where the integrand can be decomposed into a sum of low-dimensional functions, however, this is only possible in limited settings when the integrand has certain forms of sparsity.

Although the bound is dominated by a term $\mathcal{O}(T^{-1/4})$ in $T$, the proof can be extended to provide a more general result with rate up to $\mathcal{O}(T^{-1/3})$ under an additional "source condition" which requires stronger regularity from $f$; this is further discussed in Appendix A. The latter rate is minimax optimal for any nonparametric regression-based method [Stone, 1982]. Compared to baselines, we note that we cannot expect a similar result for IS since IS does not apply when $f$ depends on $\theta$. For LSMC, we also cannot guarantee consistency of the algorithm when $I(\theta)$ is not a polynomial (unless $p \to \infty$; see Stentoft [2004]). Although we are not aware of any such result, we expect KLSMC to have the same rate in $T$ as CBQ, and for CBQ to be significantly faster than KLSMC in $N$. This is due to the second stage of KLSMC being essentially the same as that for CBQ, and KLSMC using MC rather than BQ in the first stage: by Novak [1988], the convergence rate of BQ, $N^{-s_\mathcal{X}/d}$, is faster than that of MC, $N^{-1/2}$, in the case where the function $x \to f(x, \theta)$ is of smoothness at least $s_\mathcal{X} > d/2$.

## 5 EXPERIMENTS

We will now evaluate the empirical performance of CBQ against baselines including IS, LSMC and KLSMC. For the first three experiments, we focus on the case where $f$ does not depend on $\theta$ (i.e. $f(x, \theta) = f(x)$), and for the fourth experiment we focus on the case where $f$ depends on both $x$ and $\theta$. All methods use $\theta_{1:T} \sim \mathbb{Q}$ ($\mathbb{Q}$ is specified

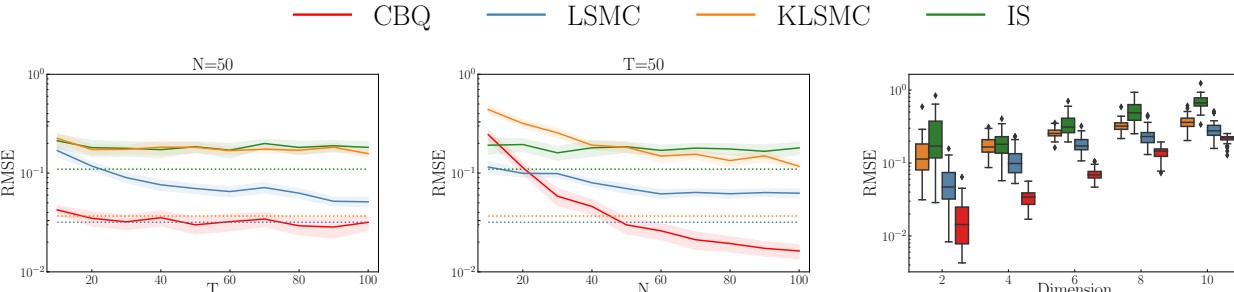

**Figure 3:** *Bayesian sensitivity analysis for linear models.* **Left:** RMSE of all methods when $d = 2$ and $N = 50$. **Middle:** RMSE of all methods when $d = 2$ and $T = 50$. **Right:** RMSE of all methods when $N = T = 100$.

individually for each experiment) and $x_{1:N}^t \sim \mathbb{P}_{\theta_t}$ to ensure a fair comparison, and we therefore use $\mathbb{P}_{\theta_1}, \dots, \mathbb{P}_{\theta_T}$ as our importance distributions in IS. For experiments on nested expectations, we use standard Monte Carlo for the outer expectation and use CBQ along with all baseline methods to compute conditional expectation for the inner expectation.

Detailed descriptions of hyperparameter selection for CBQ and all baseline methods can be found in Appendix B. Detailed experimental settings can be found in Appendix C.1 to Appendix C.4 along with detailed checklists on whether the assumptions of Theorem 1 can be satisfied in each experiment. We also provide additional experiments in Appendix C. Appendix C.5 includes experiments which show MOBQ obtains similar performance to CBQ, but with a computational cost which is between 10 and 100 times larger. Appendix C.6 includes experiments with quasi-Monte Carlo points [Hickernell, 1998], which demonstrates that CBQ is not limited to independent samples. Appendix C.7 includes ablation studies on various kernels $k_\mathcal{X}$ and $k_\Theta$. Appendix C.8 demonstrates the calibration of CBQ uncertainty. The code to reproduce all the results in this section is available at the following GitHub repository https://github.com/hudsonchen/cbq.

**Synthetic Experiment: Bayesian Sensitivity Analysis for Linear Models.** The prior and likelihood in a Bayesian analysis often depend on hyperparameters, and determining the sensitivity of the posterior to these is critical for assessing robustness [Oakley and O'Hagan, 2004, Kallioinen et al., 2021]. One way to do this is to study how posterior expectations of interest depend on these hyperparameters, a task usually requiring the computation of conditional expectations. We consider this problem in the context of Bayesian linear regression with a zero-mean Gaussian prior with covariance $\theta\mathrm{Id}_d$ where $\mathrm{Id}_d$ is identity matrix and $\theta \in (1,3)^d$. Using a Gaussian likelihood, we can obtain a conjugate Gaussian posterior $\mathbb{P}_\theta$ on the regression weights. We can then analyse sensitivity by computing the conditional expectation $I(\theta)$ of some quantity of interest $f$. For example, if $f(x) = x^\top x$, then $I(\theta)$ is the second moment of the posterior, whereas if $f(x) = x^\top y^*$ for some new observa-

tion $y^*$, then $I(\theta)$ is the predictive mean. In these simple settings, $I(\theta)$ can be computed analytically, making this a good synthetic example for benchmarking.

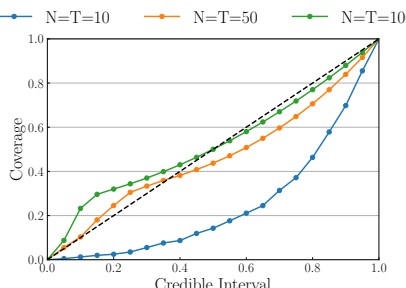

**Figure 4:** *Bayesian linear model sensitivity analysis in $d = 2$.*

Our results in Figure 3 are for the second moment, whilst the results for the predictive mean are in Appendix C.1. We measure performance in terms of root mean squared error (RMSE) and use $\mathbb{Q} = \mathrm{Unif}(1,3)^d$. For CBQ, $k_\mathcal{X}$ is chosen to be a Gaussian kernel so that the kernel mean embedding $\mu$ has a closed form, and $k_\Theta$ is a Matérn-3/2 kernel. Figure 3 shows the performance of CBQ against baselines with varying $N, T$ and $d$. LSMC performs well for this problem, and this can be explained by the fact that $I(\theta)$ is a polynomial in $\theta$. Despite this, the left and middle plots show that CBQ consistently outperforms all competitors. Specifically, its rate of convergence is initially much faster in $N$ than in $T$, which confirms the intuition from Theorem 1. The dotted lines also give the performance of baselines under a very large number of samples $N = T = 1000$, and we see that CBQ is either comparable or better than these even when it has access only to much smaller $N$ and $T$. In the right-most panel, we see that the baselines gradually catch up with CBQ as $d$ grows, which is again expected since the rate in Theorem 1 is $O(N^{-2s_\mathcal{X}/d+\varepsilon})$ in $N$. Additional experimental results demonstrating these are consistent conclusions for different values of $N, T$ can be found in Appendix C.1.

Our last plot is in Figure 4 and studies the calibration of the CBQ posterior. The coverage is the % of times a credible interval contains $I(\theta)$ under repetitions of the experiment.

The black diagonal line represents perfect calibration, whilst any curve lying above or below the black line indicates underconfidence or overconfidence respectively. We observe that when $N$ and $T$ are as small as 10, CBQ is overconfident. When $N$ and $T$ increase, CBQ becomes underconfident, meaning that our posterior variance is more inflated than needed from a frequentist viewpoint. Calibration plots for the rest of the experiments can be found in Appendix C and demonstrate similar results. It is generally preferable to be under-confident than overconfident, and CBQ does a good job most of the time. We expect that overconfidence in small $N$ and $T$ can be explained by a poor performance of empirical Bayes, and therefore caution users to not overly rely on the reported uncertainty in this regime.

**Bayesian Sensitivity Analysis for the Susceptible-Infectious-Recovered (SIR) Model.** The SIR model is commonly used to simulate the dynamics of infectious diseases through a population [Kermack and McKendrick, 1927]. In this model, the dynamics are governed by a system of differential equations parametrised by a positive infection rate and a recovery rate (see Appendix C.2). The accuracy of the numerical solution to this system typically hinges on the step size. While smaller step sizes yield more accurate solutions, they are also associated with a much higher computational cost. For example, using a step size of 0.1 days for simulating a 150-day period would require a computation time of 3 seconds for generating a single sample, which is more costly than running CBQ on $N = 40, T = 15$ samples. The cost would become even larger as the step size gets smaller, as depicted in the middle panel of Figure 5. Consequently, when performing Bayesian sensitivity for SIR, there is clear necessity for more data-efficient algorithms such as CBQ.

We perform a sensitivity analysis for the parameter $\theta$ of our $\mathrm{Gamma}(\theta, 10)$ prior on the infection rate $x$. The parameter $\theta$ represents the initial belief of the infection rate deduced from the study of the virus in the laboratory at the beginning of the outbreak. We are interested in the expected peak number of infected individuals: $f(x) = \max_r N_I^r(x)$, where $N_I^r(x)$ is the solution to the SIR equations and represents the number of infections at day $r$. It is important to study the sensitivity of $I(\theta)$ to the shape parameter $\theta$. The total population is set to be $10^6$ and $\mathbb{Q} = \mathrm{Unif}(2, 9)$ and $\mathbb{P}_{\theta_t} = \mathrm{Gamma}(\theta_t, 10)$. We use a Monte Carlo estimator with 5000 samples as the pseudo ground truth and evaluate the RMSE across all methods. For CBQ, we employ a Stein kernel for $k_{\mathcal{X}}$, with the Matérn-3/2 as the base kernel, and $k_{\Theta}$ is selected to be a Matérn-3/2 kernel.

We can see in the left panel of Figure 5 that CBQ clearly outperforms baselines including IS, LSMC and KLSMC in terms of RMSE. Although the CBQ estimator exhibits a higher computational cost compared to baselines, we have demonstrated in the middle panel of Figure 5 that, due to the increased computational expense of obtaining samples with

smaller step size, using CBQ is ultimately more efficient overall within the same period of time. Additional experimental results demonstrating these are consistent conclusions for different values of $T$ can be found in Appendix C.2.

**Option Pricing in Mathematical Finance.** Financial institutions are often interested in computing the expected loss of their portfolios if a shock were to occur in the economy, which itself requires the computation of conditional expectations (it is in fact in this context that LSMC and KLSMC was first proposed). This is typically a challenging computational problem since simulating from the stock of interest often requires the numerical solution of stochastic differential equations over a long time horizon (see Achdou and Pironneau [2005]), making data-efficient methods such as CBQ particularly desirable.

Our next experiment is representative of this class of problems, but has been chosen to have a closed-form expected loss and to be amenable to cheap simulation of the stock to enable extensive benchmarking. We consider a butterfly call option whose price $S(\tau)$ at time $\tau \in [0, \infty)$ follows the Black-Scholes formula; see Appendix C.3 for full details. The payoff at time $\tau$ can be expressed as $\psi(S(\tau)) = \max(S(\tau) - K_1, 0) + \max(S(\tau) - K_2, 0) - 2\max(S(\tau) - (K_1 + K_2)/2, 0)$ for two fixed constants $K_1, K_2 \geq 0$. We follow the set-up in Alfonsi et al. [2021, 2022] assuming that a shock occurs at time $\eta$ when the price is $S(\eta) = \theta \in (0, \infty)$, and this shock multiplies the price by $1 + s$ for some $s \geq 0$. As a result, the expected loss of the option is $\mathcal{L} = \mathbb{E}_{\theta \sim \mathbb{Q}}[\max(I(\theta), 0)]$, where $I(\theta) = \int_0^\infty f(x)\mathbb{P}_\theta(dx)$, $x = S(\zeta)$ is the price at the time $\zeta$ at which the option matures, $f(x) = \psi(x) - \psi((1 + s)x)$, and $\mathbb{P}_\theta$ and $\mathbb{Q}$ are two log-normal distributions induced from the Black-Scholes model.

Results are presented in the right-most panel of Figure 5. We take $K_1 = 50, K_2 = 150, \eta = 1, s = 0.2$ and $\zeta = 2$. For CBQ, $k_{\Theta}$ is selected to be a Matérn-3/2 kernel and $k_{\mathcal{X}}$ is either a Stein kernel with Matérn-3/2 as base kernel or a logarithmic Gaussian kernel (see Appendix C.3) in which case $k_{\mathcal{X}}$ is too smooth to satisfy the assumption of our theorem.

As expected, CBQ exhibits much faster convergence in $N$ than IS, LSMC or KLSMC, and outperforms these baselines even when they are given a substantial sample size of $N = T = 1000$ (see dotted lines). We can also see that CBQ with the log-Gaussian kernel or with Stein kernel have similar performance, despite the log-Gaussian kernel not satisfying the smoothness assumptions of our theory. Additional experiments in Appendix C.3 show that these results are consistent for different values of $T$.

**Uncertainty Decision Making in Health Economics.** In the medical world, it is important to trade-off the costs and benefits of conducting additional experiments on pa-

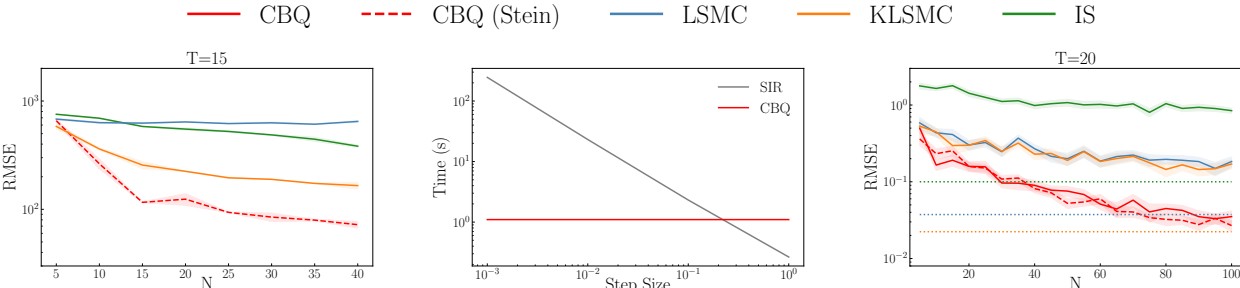

**Figure 5:** *Bayesian sensitivity analysis for SIR Model & Option pricing in mathematical finance.* **Left:** RMSE of all methods for the SIR example with $T = 15$. **Middle:** The computational cost (in wall clock time) for CBQ ($T = 15$, $N = 40$) and for obtaining one single numerical solution from SIR under different discretization step sizes. In practice, the process of obtaining samples from SIR equations is repeated $NT$ times. **Right:** RMSE of all methods for the finance example with $T = 20$.

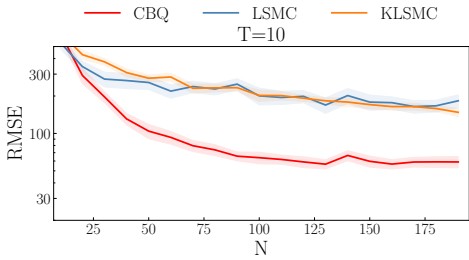

**Figure 6:** *Uncertainty decision making in health economics.* We study RMSE for different estimators of EVPPI.

tients. One important measure in this context is the expected value of partial perfect information (EVPPI), which quantifies the expected gain from conducting experiments to obtain precise knowledge of some unknown variables [Brennan et al., 2007]. The EVPPI can be expressed as $\mathbb{E}_{\theta \sim \mathbb{Q}}[\max_c I_c(\theta)] - \max_c \mathbb{E}_{\theta \sim \mathbb{Q}}[I_c(\theta)]$ where $f_c$ represents a measure of patient outcome (such as quality-adjusted life-years) under treatment $c$ among a set of potential treatments $\mathcal{C}$, $\theta$ denotes the additional variables we could measure, and $I_c(\theta) = \int_{\mathcal{X}} f_c(x, \theta)\mathbb{P}_\theta(dx)$ denotes the expected patient outcome given our measurement of $\theta$. We highlight that for these applications $N$ and $T$ are often small due to the very high monetary cost and complexity of collecting patient data in real world.

We study the potential use of CBQ for this problem using the synthetic problem of Giles and Goda [2019], where $\mathbb{P}_\theta$ and $\mathbb{Q}$ are Gaussians (see Appendix C.4). We compute EVPPI with $f_1(x, \theta) = 10^4(\theta_1 x_5 x_6 + x_7 x_8 x_9) - (x_1 + x_2 x_3 x_4)$ and $f_2(x, \theta) = 10^4(\theta_2 x_{13} x_{14} + x_{15} x_{16} x_{17}) - (x_{10} + x_{11} x_{12} x_4)$. The exact practical meanings of $x$ and $\theta$ can be found in Appendix C.4. We draw $10^6$ samples from the joint distribution to generate a pseudo ground truth, and evaluate the RMSE across different method. Note that IS is no longer applicable here because $f$ depends on both $x$ and $\theta$, so we only compare against KLSMC and LSMC. For CBQ, $k_{\mathcal{X}}$ is a Matérn-3/2 kernel and $k_{\Theta}$ is also a Matérn-3/2 kernel. In Figure 6, we can see that CBQ consistently outperforms baselines with much smaller RMSE. The results are also consistent with different values of $T$; see Appendix C.4.

## 6  CONCLUSIONS

We propose CBQ, a novel algorithm which is tailored for the computation of conditional expectations in the setting where obtaining samples or evaluating functions is costly. We show both theoretically and empirically that CBQ exhibits a fast convergence rate, and provides the additional benefit of Bayesian quantification of uncertainty. Looking forward, we believe further gains in accuracy could be obtained by developing active learning schemes to $N$, $T$, and the location of $\theta_{1:T}$ and $x_{1:N}^t$ for all $t$ in an adaptive manner. Additionally, CBQ could be extended for nested expectation problems by using a second level of BQ based on the output of second stage heteroscedastic GP, potentially leading to a further increase in accuracy.

**Ackowledgments** The authors would like to thank Motonobu Kanagawa for some helpful discussions and pointers to the literature. ZC, MN and FXB acknowledge support from the Engineering and Physical Sciences Research Council (ESPRC) through grants [EP/S021566/1] and [EP/Y022300/1]. AG was partly supported by the Gatsby Charitable Foundation.

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

# SUPPLEMENTARY MATERIAL

**TABLE OF CONTENTS**

# APPENDIX A   THEORETICAL RESULTS

To validate our methodology, we established a rate at which the CBQ estimator converges to the true value of the conditional expectation $I$ in the $\mathcal{L}^2(\Theta)$ norm, $\|I_{\text{CBQ}} - I\|_{\mathcal{L}^2(\Theta)} = \int_\Theta (I_{\text{CBQ}}(\theta) - I(\theta))^2 \mathrm{d}\theta$ in Theorem 2. The more specific version of this result was presented in the main text in Theorem 1. In this section, we prove a more general version of Theorem 1 (as well as several intermediate results), and expand on the technical background required.

For the duration of the appendix, we will denote by $M$ the total number of points in $\Theta$ instead of $T$, to avoid notation clashes with the integral operator $T$. Additionally, we will be explicit on the dependency of the BQ mean $I_{\text{BQ}}$ and variance $\sigma_{\text{BQ}}^2$ at the point $\theta$ on the realisations $x_{1:N}^\theta \sim \mathbb{P}_\theta$, meaning

$$I_{\text{BQ}}(\theta; x_{1:N}^\theta) = \mu_\theta^\top(x_{1:N}^\theta)\left(k_{\mathcal{X}}(x_{1:N}^\theta, x_{1:N}^\theta) + \lambda_{\mathcal{X}}\text{Id}_N\right)^{-1} f(x_{1:N}^\theta, \theta)$$

$$\sigma_{\text{BQ}}^2(\theta; x_{1:N}^\theta) = \mathbb{E}_{X, X' \sim \mathbb{P}_\theta}[k_{\mathcal{X}}(X, X')] - \mu_\theta^\top(x_{1:N}^\theta)\left(k_{\mathcal{X}}(x_{1:N}^\theta, x_{1:N}^\theta) + \lambda_{\mathcal{X}}\text{Id}_N\right)^{-1} \mu_\theta(x_{1:N}^\theta)$$

Finally, whenever $x_{1:N}^{\theta_t}$, we will shorten it to $x_{1:N}^t$ to avoid bulky notation. The rest of the section in structured as follows. In Appendix A.1 we present technical assumptions, and state in Theorem 2 the main convergence result the proof of which is deferred until the necessary Stage 1 and 2 results are proven. In Appendix A.2, we provide the necessary Stage 1 bounds that will be used in the proof of the main result. In Appendix A.3, we provide the necessary auxiliary results and the bound for Stage 2 in terms of Stage 1 errors. Finally, in Appendix A.4 we combine the bounds from both stages to prove Theorem 2, the more general version of Theorem 1.

## A.1   MAIN RESULT

Prior to presenting our findings, we present and justify the assumptions we have made. Throughout we use Sobolev spaces to quantify a function's smoothness. A Sobolev space $\mathcal{W}^{s,2}(\mathcal{X}, \mu)$, with $s > d/2$ and a measure $\mu$ on $\mathcal{X} \subseteq \mathbb{R}^d$, consists of functions that satisfy certain conditions: they are square integrable under the measure $\mu$, and all weak derivatives up to and including order $s$ are also square integrable under $\mu$. Weak derivatives are a generalization of ordinary derivatives, allowing for functions that are not necessarily differentiable everywhere. We write $\theta = \begin{bmatrix} \theta_{(1)} & \dots & \theta_{(p)} \end{bmatrix}$ for any $\theta \in \Theta \subseteq \mathbb{R}^p$. For a multi-index $\alpha = (\alpha_1, \dots \alpha_p) \in \mathbb{N}^p$, by $D_\theta^\alpha g$ we denote the $|\alpha| = \sum_{i=1}^d \alpha_i$ order weak derivative $D_\theta^\alpha g = D_{\theta_{(1)}}^{\alpha_1} \dots D_{\theta_{(p)}}^{\alpha_p} g$ for a function $g$ on $\Theta \subseteq \mathbb{R}^p$. Further, we assume the kernels $k_\Theta, k_{\mathcal{X}}$ are Sobolev kernels, meaning they induce Hilbert spaces that are norm-equivalent to Sobolev spaces (two normed spaces $(\mathcal{X}, \|\cdot\|_{\mathcal{X}}), (\mathcal{Y}, \|\cdot\|_{\mathcal{Y}})$ are said to be norm-equivalent when $\mathcal{X} = \mathcal{Y}$ as sets, and there are real constants $C_1', C_2' > 0$ such that for any $x \in \mathcal{X}$ it holds that $C_1'\|x\|_{\mathcal{Y}} \le \|x\|_{\mathcal{X}} \le C_2'\|x\|_{\mathcal{Y}}$).

Matérn kernels are important examples of Sobolev kernels. It is well-known that the RKHS of a Matérn kernel of order $\nu_\Theta$ over an open, convex and bounded $\Theta \subset \mathbb{R}^p$ is norm-equivalent to the Sobolev space $W^{2, \nu_\Theta + p/2}(\Theta)$ when $\nu_\Theta + p/2 \in \mathbb{Z}$; this is proven in [Wendland and Rieger, 2005, Corollary 10.48]. For $\Theta = \mathbb{R}^p$, the result can be straightforwardly extended to fractional order Sobolev-Slobodeckij spaces, $\nu_\Theta + p/2 \in \mathbb{R}$: by [Wendland and Rieger, 2005, Corollary 10.13] the RKHS of a Matérn kernel on $\mathbb{R}^p$ is norm-equivalent to a Bessel potential space, which in turn is norm-equivalent to the Sobolev-Slobodeckij space by [Adams and Fournier, 2003, Section 7.62]. Finally, one can use an extension operator in [DeVore and Sharpley, 1993, Theorems 6.1 and 6.7] to restrict the norm-equivalence result to open, convex and bounded $\Theta \subset \mathbb{R}^p$.

The following is a more general form of the assumptions in Theorem 1: specifically, we allow for the case when $\theta_{1:T}$ came from a distribution that doesn't necessarily have a density, and do not assume $\lambda_{\mathcal{X}} = 0$.

B0   (a)  $f(x, \theta)$ lies in the Sobolev space $\mathcal{W}^{s_f, 2}(\mathcal{X})$ for any $\theta \in \Theta$.

     (b)  $f(x, \theta)$ lies in the Sobolev space $\mathcal{W}^{s_I, 2}(\Theta)$ for any $x \in \mathcal{X}$.

     (c)  $M_f = \sup_{\theta \in \Theta} \max_{|\alpha| < s_I} \|D_\theta^\alpha f(\cdot, \theta)\|_{\mathcal{W}^{s_I, 2}(\mathcal{X})} < \infty$.

B1   (a)  $\mathcal{X} \subset \mathbb{R}^d$ is open, convex, and bounded.

     (b)  $\Theta \subset \mathbb{R}^p$ is open, convex, and bounded.

B2   (a)  $\theta_t$ were sampled i.i.d. from some $\mathbb{Q}$, and $\mathbb{Q}$ is equivalent to the uniform distribution on $\Theta$, meaning $\mathbb{Q}(A) = 0$ for a set $A \subset \Theta$ if and only if $\text{Unif}(A) = 0$.

     (b)  $x_{1:N}^t \sim \mathbb{P}_{\theta_t}$ for all $t \in \{1, \cdots, T\}$.

B3   $\mathbb{P}_\theta$ has a density $p_\theta$ for any $\theta \in \Theta$, and the densities are such that

     (a)  $\inf_{\theta \in \Theta, x \in \mathcal{X}} p_\theta(x) = \eta > 0$ and $\sup_{\theta \in \Theta} \|p_\theta\|_{\mathcal{L}^2(\mathcal{X})} = \eta_0 < \infty$.

(b) $p_\theta(x)$ lies in the Sobolev space $\mathcal{W}^{s_I,2}(\Theta)$ for any $x \in \mathcal{X}$.

(c) $M_p = \sup_{\substack{\theta \in \Theta \\ x \in \mathcal{X}}} \max_{|\alpha| \le s_I} |D_\theta^\alpha p_\theta(x)| < \infty$.

B4 (a) $k_\mathcal{X}$ is a Sobolev kernel of smoothness $s_\mathcal{X} \in (d/2, s_f]$.

(b) $k_\Theta$ is a Sobolev kernel of smoothness $s_\Theta \in (p/2, s_I]$.

(c) $\kappa = \sup_{\theta \in \Theta} k_\Theta(\theta, \theta) < \infty$.

B5 (a) $\lambda_\Theta = cM^{1/2}$, for $c > (4/C_6)\kappa \log(4/\delta)$ for some $C_6 \le 1$.

(b) $\lambda_\mathcal{X} \ge 0$.

Assumption B0 corresponds to conditions specified in the text of Theorem 1 prefacing the list of assumptions. Assumption B0.(b) implies $I(\theta) \in \mathcal{W}^{s_I,2}(\Theta)$: $f(x, \theta)p_\theta(x) \in \mathcal{W}^{s_I,2}(\Theta)$ by the product rule for weak derivatives (see, for instance, Evans [2018, Section 4.2.2]), and the integral lies in $\mathcal{W}^{s_I,2}(\Theta)$ by $\mathcal{W}^{s_I,2}(\Theta)$ being a complete space. Assumption B0.(c) ensures the $\mathcal{X}$-Sobolev norm of any weak derivative of $\theta \to f(\cdot, \theta)$ is uniformly bounded across all $\theta$; this will be satisfied unless $f$ is so irregular said Sobolev norms can get arbitrarily close to infinity. Assumption B3.(c), similarly, ensures that any weak derivative of $\theta \to p_\theta(x)$ is bounded across all $\theta$ and $x$. It is worth pointing out assumption B4.(c), boundedness of the kernel, follows from assumption B4.(b); however, we keep it separate as some results will only require that the kernel is bounded, not necessarily that it is Sobolev.

Crucially, in the proofs in the next section we will see that the assumptions imply that the setting of the model in Stage 1 satisfies the assumptions of [Wynne et al., 2021, Theorem 4], and the setting of the model in Stage 2 satisfies the assumptions necessary to establish convergence of a noisy importance-weighted kernel ridge regression estimator—the two key results we will use to prove the convergence rate of the estimator.

We now state the main convergence result, which is a version of Theorem 1 for $\lambda_\mathcal{X} \ge 0$. The proof of both this result and the more specific Theorem 1 are postponed until Section BLAH, as they rely on intermediary results.

**Theorem 2** (Generalised Theorem 1). *Suppose all technical assumptions in Appendix A.1 hold. Then for any $\delta \in (0, 1)$ there is an $N_0 > 0$ such that for any $N \ge N_0$, with probability at least $1 - \delta$ it holds that*

$$\|I_{\mathrm{CBQ}} - I\|_{\mathcal{L}^2(\Theta, \mathbb{Q})} \le \left(1 + c^{-1}M^{-\frac{1}{2}}\left(\lambda_\mathcal{X} + C_2 N^{-1+2\varepsilon}\left(N^{-\frac{s_\mathcal{X}}{d}+\frac{1}{2}+\varepsilon} + C_3\lambda_\mathcal{X}\right)^2\right)\right)$$
$$\times \left(C_7(\delta)N^{-\frac{1}{2}+\varepsilon}\left(N^{-\frac{s_\mathcal{X}}{d}+\frac{1}{2}+\varepsilon} + C_5\lambda_\mathcal{X}\right) + C_8(\delta)M^{-\frac{1}{4}}\|I\|_{\mathcal{H}_\Theta}\right)$$

*for any arbitrarily small $\varepsilon > 0$, constants $C_2, C_3, C_5, C_7(\delta) = O(1/\delta)$ and $C_8(\delta) = O(\log(1/\delta))$ independent of $N, M, \varepsilon$.*

## A.2 STAGE 1 BOUNDS

Recall that we use the shorthand $x_{1:N}^t$ for $x_{1:N}^{\theta_t}$. In this section, we bound the BQ variance $\sigma_{\mathrm{BQ}}^2(\theta; x_{1:N}^\theta)$ in expectation in Theorem 3, and the difference between $I_{\mathrm{BQ}}(\theta; x_{1:N}^\theta)$ and $I$ in the norm of the RKHS $\mathcal{H}_\Theta$ induced by the kernel $k_\Theta$ in Theorem 4. Later in Appendix A.3, the error of the estimator $I_{\mathrm{CBQ}}$ will be bounded in terms of these quantities.

**Theorem 3.** *Suppose Assumptions B0.(a), B1.(a), B3.(b), B4.(a), and B5.(b) hold. Then there is a $N_0 > 0$ such that for all $N \ge N_0$ it holds that*

$$\mathbb{E}_{y_{1:N}^\theta \sim \mathbb{P}_\theta} \sigma_{\mathrm{BQ}}^2(\theta; y_{1:N}^\theta) \le \lambda_\mathcal{X} + C_2 N^{-1+2\varepsilon}\left(N^{-\frac{s_\mathcal{X}}{d}+\frac{1}{2}+\varepsilon} + C_3\lambda_\mathcal{X}\right)^2$$

*for any $\theta \in \Theta$, any arbitrarily small $\varepsilon > 0$, and $C_2, C_3$ independent of $\theta, N, \varepsilon, \lambda_\mathcal{X}$.*

The term $N_0$ quantifies how likely the points $y_{1:N}^\theta$ are to "fill out" the space $\mathcal{X}$—for any $\theta$. Intuitively speaking, $N_0$ is smallest when for all $\theta$, the $\mathbb{P}_\theta$ is uniform.

*Proof.* Recall

$$I_{\mathrm{BQ}}(\theta; y_{1:N}^\theta) = \mu_\theta(y_{1:N}^\theta)^\top \left(k_\mathcal{X}(y_{1:N}^\theta, y_{1:N}^\theta) + \lambda_\mathcal{X}\mathrm{Id}_N\right)^{-1} f(y_{1:N}^\theta, \theta),$$
$$\sigma_{\mathrm{BQ}}^2(\theta; y_{1:N}^\theta) = \mathbb{E}_{X, X' \sim \mathbb{P}_\theta}[k_\mathcal{X}(X, X')] - \mu_\theta(y_{1:N}^\theta)^\top \left(k_\mathcal{X}(y_{1:N}^\theta, y_{1:N}^\theta) + \lambda_\mathcal{X}\mathrm{Id}_N\right)^{-1} \mu_\theta(y_{1:N}^\theta).$$

We seek to bound $\sigma^2_{\mathrm{BQ}}(\theta; y^\theta_{1:N})$. [Kanagawa et al., 2018, Proposition 3.8] pointed out that the Gaussian noise posterior is the worst-case error in the $\mathcal{H}^{\lambda_\mathcal{X}}_\mathcal{X}$, the RKHS induced by the kernel $k^{\lambda_\mathcal{X}}_\mathcal{X}(x, x') = k_\mathcal{X}(x, x') + \lambda_\mathcal{X}\delta(x, x')$ (where $\delta(x, x') = 1$ if $x = x'$, and 0 otherwise). Through straightforward algebraic manipulations and using the reproducing property, one can show that for the vector $w_\theta = k(x, y^\theta_{1:N})^\top \left(k_\mathcal{X}(y^\theta_{1:N}, y^\theta_{1:N}) + \lambda_\mathcal{X}\mathrm{Id}_N\right)^{-1} \in \mathbb{R}^N$,

$$\sigma^2_{\mathrm{BQ}}(\theta; y^\theta_{1:N}) - \lambda_\mathcal{X} = \sup_{\|f\|_{\mathcal{H}^{\lambda_\mathcal{X}}_\mathcal{X}} \leq 1} \left| w_\theta f(y^\theta_{1:N}) - \int_\mathcal{X} f(x)\mathbb{P}_\theta(\mathrm{d}x) \right|^2, \tag{A.1}$$

Since $\mathcal{H}^{\lambda_\mathcal{X}}_\mathcal{X}$ is induced by the sum of kernels, $k^{\lambda_\mathcal{X}}_\mathcal{X}(x, x') = k_\mathcal{X}(x, x') + \lambda_\mathcal{X}$, it holds that $\mathcal{H}_\mathcal{X} \subseteq \mathcal{H}^{\lambda_\mathcal{X}}_\mathcal{X}$, and $\|f\|_{\mathcal{H}^{\lambda_\mathcal{X}}_\mathcal{X}} \leq \|f\|_{\mathcal{H}_\mathcal{X}}$ [Aronszajn, 1950, Theorem I.13.IV]. Therefore, the class of functions $f$ for which $\|f\|_{\mathcal{H}_\mathcal{X}} \leq 1$ is larger than that for which $\|f\|_{\mathcal{H}^{\lambda_\mathcal{X}}_\mathcal{X}} \leq 1$, and

$$\sup_{\|f\|_{\mathcal{H}^{\lambda_\mathcal{X}}_\mathcal{X}} \leq 1} \left| w_\theta f(y^\theta_{1:N}) - \int_\mathcal{X} f(x)\mathbb{P}_\theta(\mathrm{d}x) \right| \leq \sup_{\|f\|_{\mathcal{H}_\mathcal{X}} \leq 1} \left| w_\theta f(y^\theta_{1:N}) - \int_\mathcal{X} f(x)\mathbb{P}_\theta(\mathrm{d}x) \right|. \tag{A.2}$$

Next, note that for $\hat{f}_\theta(x) = k(x, y^\theta_{1:N})^\top \left(k_\mathcal{X}(y^\theta_{1:N}, y^\theta_{1:N}) + \lambda_\mathcal{X}\mathrm{Id}_N\right)^{-1} f(y^\theta_{1:N})$,

$$\left| w_\theta f(y^\theta_{1:N}) - \int_\mathcal{X} f(x)\mathbb{P}_\theta(\mathrm{d}x) \right| = \left| \int_\mathcal{X} \left( \hat{f}_\theta(x) - f(x) \right) \mathbb{P}_\theta(\mathrm{d}x) \right| \leq \int_\mathcal{X} \left| \hat{f}_\theta(x) - f(x) \right| \mathbb{P}_\theta(\mathrm{d}x)$$
$$\leq \|\hat{f}_\theta - f\|_{\mathcal{L}^2(\mathcal{X})} \|p_\theta\|_{\mathcal{L}^2(\mathcal{X})}, \tag{A.3}$$

where the last inequality is an application of Hölder inequality. By Assumption B3.(b), $\|p_\theta\|_{\mathcal{L}^2(\mathcal{X})}$ is bounded above by $\eta_0$. In order to apply [Wynne et al., 2021, Theorem 4] to bound $\|\hat{f}_\theta - f\|_{\mathcal{L}^2(\mathcal{X})}$, we show the assumptions of that Theorem hold.

Assumption 1 (Assumptions on the Domain): An open, bounded, and convex $\mathcal{X}$ satisfies the assumption, as discussed in Wynne et al. [2021].

Assumption 2 (Assumptions on the Kernel Parameters) and Assumption 3 (Assumptions on the Kernel Smoothness Range): Our setting is more specific than the one [Wynne et al., 2021, Theorem 4]: the kernel $k_\mathcal{X}$ is Matérn, and therefore all smoothness constants mentioned in Assumptions 2 and 3 have the same value, $s_\mathcal{X}$.

Assumption 4 (Assumptions on the Target Function and Mean Function): The target function $f$ was assumed to have higher smoothness than $k_\mathcal{X}$ in B0.(a), and B4.(a); the mean function was taken to be zero.

Assumption 5 (Additional Assumptions on Kernel Parameters): By B4.(a) and B0.(a) the smoothness of the true function $s_f \geq s_\mathcal{X} > d/2$, which verifies both statements in the Assumption since all smoothness constants of the kernel are equal to $s_\mathcal{X}$.

Therefore [Wynne et al., 2021, Theorem 4] holds, and for $\mathcal{W}_{0,2}(\mathcal{X}) = \mathcal{L}^2(\mathcal{X})$

$$\|\hat{f}_\theta - f\|_{\mathcal{L}^2(\mathcal{X})} \leq K_3 \|f\|_{\mathcal{H}_\mathcal{X}} h^{\frac{d}{2}}_{y^\theta_{1:N}} \left( h^{s_\mathcal{X} - \frac{d}{2}}_{y^\theta_{1:N}} + \lambda_\mathcal{X} \right),$$

for any $N$ for which the fill distance $h_{y^\theta_{1:N}} \leq h_0$ for some $h_0$, and $K_3$ and $h_0$ that depend on $\mathcal{X}, s_f, s_\mathcal{X}$.[1]

For $y^\theta_{1:N} \sim \mathbb{P}_\theta$, we can guarantee that $h_{y^\theta_{1:N}} \leq h_0$ in expectation using [Oates et al., 2019, Lemma 2], which says that provided the density $\inf_x p_\theta(x) > 0$, there is a $C_\theta$ such that $\mathbb{E}\, h_{y^\theta_{1:N}} \leq C_\theta N^{-1/d+\varepsilon}$ for an arbitrarily small $\varepsilon > 0$, for $C_\theta$ that depends on $\theta$ through $\inf_x p_\theta(x)$. The smaller $\inf_x p_\theta(x)$, the larger $C_\theta$. Since we assumed $\inf_{x,\theta} p_\theta(x) = \eta > 0$ there is a $K_4$ such that $C_\theta \leq K_4$ for any $\theta$. Therefore, we may take $N_0$ to be the smallest $N$ for which $\mathbb{E}\, h_{y^\theta_{1:N}} \leq K_4 N^{-1/d+\varepsilon}$ holds, and have for all $N \geq N_0$

$$\mathbb{E}_{y^\theta_{1:N} \sim \mathbb{P}_\theta} \|\hat{f}_\theta - f\|_{\mathcal{L}^2(\mathcal{X})} \leq K_3 K_4^{\frac{d}{2}} \|f\|_{\mathcal{H}_\mathcal{X}} N^{-\frac{1}{2}+\varepsilon} \left( K_4^{s_\mathcal{X} - \frac{d}{2}} N^{-\frac{s_\mathcal{X}}{d} + \frac{1}{2} + \varepsilon} + \lambda_\mathcal{X} \right) \tag{A.4}$$

---

[1] Note that the result in [Wynne et al., 2021, Theorem 4] features $\|f\|_{\mathcal{W}^{s_\mathcal{X},2}(\mathcal{X})}$, not $\|f\|_{\mathcal{H}_\mathcal{X}}$. The bound in terms $\|f\|_{\mathcal{H}_\mathcal{X}}$ holds since $\mathcal{H}_\mathcal{X}$ was assumed to be a Sobolev RKHS.

Putting together Equations (A.1) to (A.4) and Assumption B3.(b), we get the result,

$$
\begin{aligned}
\mathbb{E}_{y_{1:N}^\theta \sim \mathbb{P}_\theta}\, \sigma^2_{\mathrm{BQ}}(\theta; y_{1:N}^\theta) - \lambda_{\mathcal{X}} &= \sup_{\|f\|_{\mathcal{H}_{\mathcal{X}}^{\lambda_{\mathcal{X}}}} \leq 1} \mathbb{E}_{y_{1:N}^\theta \sim \mathbb{P}_\theta}\left| w_\theta f(y_{1:N}^\theta) - \int_{\mathcal{X}} f(x)\mathbb{P}_\theta(\mathrm{d}x) \right|^2 \\
&\leq \sup_{\|f\|_{\mathcal{H}_{\mathcal{X}}} \leq 1} \mathbb{E}_{y_{1:N}^\theta \sim \mathbb{P}_\theta}\left| w_\theta f(y_{1:N}^\theta) - \int_{\mathcal{X}} f(x)\mathbb{P}_\theta(\mathrm{d}x) \right|^2 \\
&\leq \sup_{\|f\|_{\mathcal{H}_{\mathcal{X}}} \leq 1} \mathbb{E}_{y_{1:N}^\theta \sim \mathbb{P}_\theta} \|\hat{f}_\theta - f\|^2_{\mathcal{L}^2(\mathcal{X})} \|p_\theta\|^2_{\mathcal{L}^2(\mathcal{X})} \\
&\leq \eta_0^2 K_3^2 K_4^d N^{-1+2\varepsilon}\left( K_4^{s_{\mathcal{X}} - \frac{d}{2}} N^{-\frac{s_{\mathcal{X}}}{d} + \frac{1}{2} + \varepsilon} + \lambda_{\mathcal{X}} \right)^2 \\
&=: C_2 N^{-1+2\varepsilon}\left( N^{-\frac{s_{\mathcal{X}}}{d} + \frac{1}{2} + \varepsilon} + C_3 \lambda_{\mathcal{X}} \right)^2 .
\end{aligned}
$$

$\square$

Before bounding the error $\|I_{\mathrm{BQ}} - I\|_{\mathcal{H}_\Theta}$, we give the following general auxiliary result for an arbitrary Sobolev space of function over some open $\Omega \subseteq \mathbb{R}^d$.

**Proposition 1.** *Suppose $f, g$ lie in a Sobolev space $\mathcal{W}^{s,2}(\Omega)$ for some of smoothness $s$, and for all $|\alpha| \leq s$ the weak derivative $D^\alpha g$ is bounded. Take $M = \max_{|\alpha| \leq s} \|D^\alpha g\|_{\mathcal{L}^\infty(\Omega)}$. Then, there is a constant $K$ such that*

$$
\|fg\|_{\mathcal{W}^{s,2}(\Omega)} \leq KM\|f\|_{\mathcal{W}^{s,2}(\Omega)}.
$$

*Proof.* Recall that the norm in a Sobolev space is defined as

$$
\|fg\|^2_{\mathcal{W}^{s,2}(\Omega)} = \sum_{|\alpha| \leq s} \|D^\alpha[fg]\|^2_{\mathcal{L}^2(\Omega)}. \tag{A.5}
$$

Fix some $\alpha$ such that $|\alpha| \leq s$. By the product rule to weak derivatives (see, for instance, Evans [2018, Section 4.2.2]), it holds that

$$
D^\alpha[fg] = \sum_{|\alpha'| \leq |\alpha|} \sum_{|\alpha''| \leq |\alpha|} C_{\alpha',\alpha'',\alpha} D^{\alpha'}[f] D^{\alpha''}[g],
$$

for all $\alpha', \alpha''$ being multi-indices of the same dimension as $\alpha$, and some real constants $C_{\alpha',\alpha'',\alpha} > 0$ that only depend on $\alpha$ and not $f$ or $g$. Then

$$
\begin{aligned}
\|D^\alpha[fg]\|^2_{\mathcal{L}^2(\Omega)} &= \left\| \sum_{|\alpha'| \leq |\alpha|} \sum_{|\alpha''| \leq |\alpha|} C_{\alpha',\alpha'',\alpha} D^{\alpha'}[f] D^{\alpha''}[g] \right\|^2_{\mathcal{L}^2(\Omega)} \\
&\overset{(A)}{\leq} \left( \sum_{|\alpha'| \leq |\alpha|} \sum_{|\alpha''| \leq |\alpha|} C_{\alpha',\alpha'',\alpha} \|D^{\alpha'}[f] D^{\alpha''}[g]\|_{\mathcal{L}^2(\Omega)} \right)^2 \\
&\overset{(B)}{\leq} 2\binom{d}{|\alpha|} \sum_{|\alpha'| \leq |\alpha|} \sum_{|\alpha''| \leq |\alpha|} C_{\alpha',\alpha'',\alpha} \|D^{\alpha'}[f] D^{\alpha''}[g]\|^2_{\mathcal{L}^2(\Omega)} \\
&\overset{(C)}{\leq} 2M^2\binom{d}{|\alpha|} \sum_{|\alpha'| \leq |\alpha|} \sum_{|\alpha''| \leq |\alpha|} C_{\alpha',\alpha'',\alpha} \|D^{\alpha'}[f]\|^2_{\mathcal{L}^2(\Omega)} \\
&\leq 2M^2\binom{d}{|\alpha|} \sum_{|\alpha'| \leq |\alpha|} \sum_{|\alpha''| \leq |\alpha|} C_{\alpha',\alpha'',\alpha} \|f\|^2_{\mathcal{W}^{s,2}(\Omega)},
\end{aligned}
$$

where $(A)$ holds by triangle inequality, $(B)$ holds as, by Cauchy-Schwartz, $\left(\sum_{i=1}^n a_i\right)^2 \leq n\sum_{i=1}^n a_i^2$ for any real $a_i$, and as the number of multi-indices in $\mathbb{N}^d$ of size at most $\alpha$ is "$d$ choose $|\alpha|$", and $(C)$ by

the definition $M = \max_{|\alpha|\leq s}\|D^\alpha g\|_{\mathcal{L}^\infty(\Omega)}$. Substituting this into Equation (A.5), we get that for $\sqrt{K} = 2\sum_{|\alpha|<s}\binom{d}{|\alpha|}\sum_{|\alpha'|\leq|\alpha|}\sum_{|\alpha''|\leq|\alpha|}C_{\alpha',\alpha'',\alpha}$,

$$\|fg\|^2_{\mathcal{W}^{s,2}(\Omega)} \leq K^2M^2\|f\|^2_{\mathcal{W}^{s,2}(\Omega)}.$$

$\square$

With the Sobolev norm bound in place, we are ready to give the bound on $\|I_{\mathrm{BQ}}-I\|_{\mathcal{H}_\Theta}$.

**Theorem 4.** *Suppose Assumptions B0.(a), B0.(c), B1.(a), B2.(b), B3.(b), B3.(c), B4.(a), B4.(b) and B5.(b) hold. Then there is a $N_0 > 0$ such that for all $N \geq N_0$ with probability at least $1 - \delta/2$ it holds that*

$$\|I_{\mathrm{BQ}}-I\|_{\mathcal{H}_\Theta} \leq \frac{2}{\delta}C_4 N^{-\frac{1}{2}+\varepsilon}\left(N^{-\frac{s_\mathcal{X}}{d}+\frac{1}{2}+\varepsilon}+C_5\lambda_\mathcal{X}\right).$$

*for any arbitrarily small $\varepsilon > 0$, and $C_4, C_5$ independent of $N, \varepsilon, \lambda_\mathcal{X}$.*

*Proof.* Recall that, as $\mathcal{H}_\Theta$ is a Sobolev RKHS (meaning $k_\Theta$ is a Sobolev kernel) of smoothness $s_\Theta$, it holds that $C_1'\|g\|_{\mathcal{W}^{s_\Theta,2}(\Theta)} \leq \|g\|_{\mathcal{H}_\Theta} \leq C_2'\|g\|_{\mathcal{W}^{s_\Theta,2}(\Theta)}$ for some constants $C_1', C_2' > 0$ and any $g \in \mathcal{H}_\Theta$. Take $\hat{f}(x,\theta) = k(x,x^\theta_{1:N})^\top\left(k_\mathcal{X}(x^\theta_{1:N},x^\theta_{1:N})+\lambda_\mathcal{X}\mathrm{Id}_N\right)^{-1}f(x^\theta_{1:N},\theta)$. Then,

$$\begin{aligned}
\|I_{\mathrm{BQ}}-I\|^2_{\mathcal{H}_\Theta} &= \langle I_{\mathrm{BQ}}-I, I_{\mathrm{BQ}}-I\rangle_{\mathcal{H}_\Theta}\\
&= \left\langle \int_\mathcal{X}\left(\hat{f}(x,\theta)-f(x,\theta)\right)p_\theta(x)\mathrm{d}x, \int_\mathcal{X}\left(\hat{f}(x',\theta)-f(x',\theta)\right)p_\theta(x')\mathrm{d}x'\right\rangle_{\mathcal{H}_\Theta}\\
&\leq \int_\mathcal{X}\int_\mathcal{X}\left\langle\left(\hat{f}(x,\theta)-f(x,\theta)\right)p_\theta(x),\left(\hat{f}(x',\theta)-f(x',\theta)\right)p_\theta(x')\right\rangle_{\mathcal{H}_\Theta}\mathrm{d}x\mathrm{d}x'\\
&\overset{(A)}{\leq}\left(\int_\mathcal{X}\left\|\left(\hat{f}(x,\theta)-f(x,\theta)\right)p_\theta(x)\right\|_{\mathcal{H}_\Theta}\mathrm{d}x\right)^2\\
&\overset{(B)}{\leq}C_2'^2 K^2 M_p^2\left(\int_\mathcal{X}\left\|\hat{f}(x,\theta)-f(x,\theta)\right\|_{\mathcal{W}^{s_\Theta,2}(\Theta)}\mathrm{d}x\right)^2,
\end{aligned}$$

where $(A)$ holds by the Cauchy-Schwarz, $(B)$ by Proposition 1 and $\mathcal{H}_\Theta$ being a Sobolev RKHS. As for the remaining term,

$$\begin{aligned}
\int_\mathcal{X}\left\|\hat{f}(x,\theta)-f(x,\theta)\right\|^2_{\mathcal{W}^{s_\Theta,2}(\Theta)}\mathrm{d}x &= \sum_{|\alpha|\leq s_\Theta}\int_\mathcal{X}\int_\Theta\left(D^\alpha_\theta\hat{f}(x,\theta)-D^\alpha_\theta f(x,\theta)\right)^2\mathrm{d}\theta\mathrm{d}x\\
&= \sum_{|\alpha|\leq s_\Theta}\int_\Theta\int_\mathcal{X}\left(D^\alpha_\theta\hat{f}(x,\theta)-D^\alpha_\theta f(x,\theta)\right)^2\mathrm{d}x\mathrm{d}\theta\\
&= \sum_{|\alpha|\leq s_\Theta}\int_\Theta\left\|D^\alpha_\theta\hat{f}(x,\theta)-D^\alpha_\theta f(x,\theta)\right\|^2_{\mathcal{L}^2(\mathcal{X})}\mathrm{d}\theta
\end{aligned}$$

Since $D^\alpha_\theta\hat{f}(x,\theta) = k(x,x^\theta_{1:N})^\top\left(k_\mathcal{X}(x^\theta_{1:N},x^\theta_{1:N})+\lambda_\mathcal{X}\mathrm{Id}_N\right)^{-1}D^\alpha_\theta f(x^\theta_{1:N},\theta)$, and the $\mathcal{X}$-smoothness of $D^\alpha_\theta f$ is the same as that of $f$, we may use Wynne et al. [2021, Theorem 4] to bound $\|D^\alpha_\theta\hat{f}(x,\theta)-D^\alpha_\theta f(x,\theta)\|_{\mathcal{L}^2(\mathcal{X})}$ identically to the proof of Theorem 3. Then, we have that

$$\begin{aligned}
\mathbb{E}_{x^\theta_{1:N}\sim\mathbb{P}_\theta}\|D^\alpha_\theta\hat{f}(x,\theta)-D^\alpha_\theta f(x,\theta)\|_{\mathcal{L}^2(\mathcal{X})} &\leq K_3 K_4^{\frac{d}{2}}\|D^\alpha_\theta f\|_{\mathcal{H}_\mathcal{X}}N^{-\frac{1}{2}+\varepsilon}\left(K_4^{s_\mathcal{X}-\frac{d}{2}}N^{-\frac{s_\mathcal{X}}{d}+\frac{1}{2}+\varepsilon}+\lambda_\mathcal{X}\right)\\
&\overset{(A)}{\leq}K_3 K_4^{\frac{d}{2}}C_2'M_f N^{-\frac{1}{2}+\varepsilon}\left(K_4^{s_\mathcal{X}-\frac{d}{2}}N^{-\frac{s_\mathcal{X}}{d}+\frac{1}{2}+\varepsilon}+\lambda_\mathcal{X}\right),
\end{aligned}$$

where $(A)$ holds by Assumption B0.(c), and $k_\mathcal{X}$ being a Sobolev kernel and $C_2'$ being a norm equivalence constant. Define By Markov's inequality, for any $\delta/2 \in (0,1)$ it holds with probability at least $1-\delta/2$ that

$$\|D^\alpha_\theta\hat{f}(x,\theta)-D^\alpha_\theta f(x,\theta)\|_{\mathcal{L}^2(\mathcal{X})} \leq \frac{2}{\delta}K_3 K_4^{\frac{d}{2}}C_2'M_f N^{-\frac{1}{2}+\varepsilon}\left(K_4^{s_\mathcal{X}-\frac{d}{2}}N^{-\frac{s_\mathcal{X}}{d}+\frac{1}{2}+\varepsilon}+\lambda_\mathcal{X}\right)$$

Lastly, the number of $\alpha$ such that $|\alpha| < s_\Theta$ is the combination "$p$ select $s_\Theta$". Then,

$$\|I_{\mathrm{BQ}} - I\|_{\mathcal{H}_\Theta}^2 \le C_2'^2 K^2 M_p^2 \binom{p}{s_\Theta} \left( \frac{2}{\delta} K_3 K_4^{\frac{d}{2}} C_2' M_f N^{-\frac{1}{2}+\varepsilon} \left( K_4^{s_\mathcal{X}-\frac{d}{2}} N^{-\frac{s_\mathcal{X}}{d}+\frac{1}{2}+\varepsilon} + \lambda_\mathcal{X} \right) \right)^2$$

$$=: \frac{4}{\delta^2} \sqrt{C_4} N^{-1+2\varepsilon} \left( N^{-\frac{s_\mathcal{X}}{d}+\frac{1}{2}+\varepsilon} + C_5 \lambda_\mathcal{X} \right)^2 .$$

$\square$

## A.3   STAGE 2 BOUNDS

In this section, we establish convergence of the estimator $I_{\mathrm{CBQ}}$ to the true function $I$ in the norm $\mathcal{L}^2(\Theta, \mathbb{Q})$, first in terms of the error $\|I_{\mathrm{BQ}}(\cdot; x_{1:N}^\theta) - I(\cdot)\|_{\mathcal{H}_\Theta}$ in Theorem 5, and additionally in the variance $\sigma_{\mathrm{BQ}}^2(\theta; x_{1:N}^\theta)$ in Corollary 1. To do so, we represent the CBQ estimator as

$$I_{\mathrm{CBQ}}(\theta) = k_\Theta(\theta, \theta_{1:M}) \left( k_\Theta(\theta_{1:M}, \theta_{1:M}) + \mathrm{diag}\left[ \frac{M\lambda}{w(\theta_{1:M}) + \varepsilon(\theta_{1:M}; x_{1:N}^{1:M})} \right] \right)^{-1} I_{\mathrm{BQ}}(\theta_{1:M}; x_{1:N}^{1:M}). \qquad (\mathrm{A.6})$$

for vector notation $\varepsilon(\theta_{1:M}; x_{1:N}^{1:M}) = [\varepsilon(\theta_1; x_{1:N}^1), \dots, \varepsilon(\theta_M; x_{1:N}^M)]^\top \in \mathbb{R}^M$, and $\lambda$, the weight $w : \Theta \to \mathbb{R}$ and the noise term $\varepsilon : \Theta \to \mathbb{R}$ given by

$$\lambda = \lambda_\Theta M^{-1}$$
$$w(\theta) = \mathbb{E}_{y_{1:N}^\theta \sim \mathbb{P}_\theta} \frac{\lambda_\Theta}{\lambda_\Theta + \sigma_{\mathrm{BQ}}^2(\theta; y_{1:N}^\theta)}, \qquad (\mathrm{A.7})$$
$$\varepsilon(\theta; x_{1:N}^\theta) = \frac{\lambda_\Theta}{\lambda_\Theta + \sigma_{\mathrm{BQ}}^2(\theta; x_{1:N}^\theta)} - \mathbb{E}_{y_{1:N}^\theta \sim \mathbb{P}_\theta} \frac{\lambda_\Theta}{\lambda_\Theta + \sigma_{\mathrm{BQ}}^2(\theta; y_{1:N}^\theta)}.$$

The equality to the CBQ estimator given in the main text can be easily seen, as the term under the $\mathrm{diag}$ is

$$\frac{M\lambda}{w(\theta_{1:M}) + \varepsilon(\theta_{1:M}; x_{1:N}^{1:M})} = \frac{M\lambda_\Theta M^{-1}}{\frac{\lambda_\Theta}{\lambda_\Theta + \sigma_{\mathrm{BQ}}^2(\theta; x_{1:N}^\theta)}} = \lambda_\Theta + \sigma_{\mathrm{BQ}}^2(\theta; x_{1:N}^\theta).$$

If the noise term in Equation (A.6) were absent (meaning, equal to zero), the estimator would become the *importance-weighted kernel ridge regression* (IW-KRR) estimator. The convergence of the IW-KRR estimator was studied in Gogolashvili et al. [2023, Theorem 4]. In this section, we extend their results to the case of noisy weights ($\varepsilon \not\equiv 0$), which are additionally correlated with the noise in $I_{\mathrm{BQ}}(\theta_i; x_{1:N}^i)$ (through the shared datapoints $x_{1:N}^i$).

Note that, while we only provide results specific for $I_{\mathrm{CBQ}}$, the proof can be extended with minor modifications to the more general case of arbitrary noisy IW-KRR with weights that satisfy conditions in Gogolashvili et al. [2023], and zero-mean weight noise.

The convergence results for the *noisy importance-weighted kernel ridge regression* estimator in Appendix A.3.3 will rely on a representation of $I_{\mathrm{CBQ}}$ in terms of a sample-level version of a certain weighted integral operator. Then, we bound the gap between $I_{\mathrm{CBQ}}$ and $I$ in terms of (1) the gap between the sample-level version of said operator, and the population-level version, and (2) the gap between $I_{\mathrm{BQ}}$ and $I$. Next, we define said operator, and additional notation used in the proofs.

### A.3.1   Notation

We will be working on positive, bounded, self-adjoint $\mathcal{H}_\Theta \to \mathcal{H}_\Theta$ operators

$$T[g](\theta) = \int_\Theta k_\Theta(\theta, \theta') g(\theta') w(\theta') \mathbb{Q}(\mathrm{d}\theta') \qquad \hat{T}[g](\theta) = \frac{1}{M} k_\Theta(\theta, \theta_{1:M}) \mathrm{diag}\left[ w(\theta_{1:M}) + \varepsilon(\theta_{1:M}; x_{1:N}^{1:M}) \right] g(\theta_{1:M}).$$
$$(\mathrm{A.8})$$

for the weight function $w$ and noise term $\varepsilon$ as defined in Equation (A.7). We will denote HS to be the Hilbert space of Hilbert-Schmidt operators $\mathcal{H}_\Theta \to \mathcal{H}_\Theta$, $\|\cdot\|_{\mathrm{HS}}$ to be the Hilbert-Schmidt norm, and $\|\cdot\|_{\mathrm{op}}$ to be the operator norm. As is customary, we will write $T + \lambda$ to mean the operator $T + \lambda \mathrm{Id}_{\mathcal{H}_\Theta}$, where $\mathrm{Id}_{\mathcal{H}_\Theta}$ is the identity operator $\mathcal{H}_\Theta \to \mathcal{H}_\Theta$.

### A.3.2 Auxiliary results

The results given in this section are key to proving the main Stage 2 result, Theorem 5. The following result bounds the Hilbert-Schmidt norm on the "gap" between the population-level $T$ and the sample-level $\hat{T}$, when their difference is "sandwiched" between $(T + \lambda)^{-1/2}$. With some manipulation, this term will appear in the proof of Theorem 5.

**Lemma 1** (Modified Lemma 18 in Gogolashvili et al. [2023]). *Suppose Assumptions B2.(b), B4.(c) hold, and the operators $T, \hat{T}$ be as defined in Appendix A.3.1. Then, with probability greater than $1 - \delta/2$,*

$$S_1 := \|(T + \lambda)^{-1/2}(T - \hat{T})(T + \lambda)^{-1/2}\|_{\mathrm{HS}} \leq \frac{4\kappa}{\lambda\sqrt{M}} \log(4/\delta).$$

*Additionally, if $\lambda\sqrt{M} > (4/C_6)\kappa \log(4/\delta)$ for some $C_6 \leq 1$, it holds that $S_1 < C_6 \leq 1$.*

The fact that $S_1$ is strictly less than 1 will be important in the proof of the main Stage 2 result, Theorem 5, as it will allow us to apply Neumann series expansion to $\|(\mathrm{Id} - (T + \lambda)^{-1/2}(T - \hat{T})(T + \lambda)^{-1/2})^{-1}\|_{\mathrm{op}}$.

*Proof.* Denote a feature function $\varphi_\theta(\cdot) := k_\Theta(\theta, \cdot)$. Let $\xi, \xi_1, \ldots, \xi_M$ be random variables in HS defined as

$$\xi = (T + \lambda)^{-1/2}(w(\theta) + \varepsilon(\theta; x_{1:N}^\theta))\varphi_\theta\langle\varphi_\theta, \cdot\rangle_{\mathcal{H}_\Theta}(T + \lambda)^{-1/2}$$
$$\xi_i = (T + \lambda)^{-1/2}(w(\theta_i) + \varepsilon(\theta_i; x_{1:N}^i))\varphi_{\theta_i}\langle\varphi_{\theta_i}, \cdot\rangle_{\mathcal{H}_\Theta}(T + \lambda)^{-1/2}$$

First, note that as $(\theta, x_{1:N}), (\theta_1, x_{1:N}^1), \ldots, (\theta_N, x_{1:N}^N)$ are i.i.d., it follows that $\xi, \xi_1, \ldots, \xi_N$ are i.i.d. random variables in HS. Further, as $\mathbb{E}_{x_{1:N}^\theta \sim \mathbb{P}_\theta} \varepsilon(\theta; x_{1:N}^\theta) = 0$, it holds that

$$\mathbb{E}_{\theta \sim \mathbb{Q}} \mathbb{E}_{x_{1:N}^\theta \sim \mathbb{P}_\theta} \xi = \mathbb{E}_{\theta \sim \mathbb{Q}}\left[(T + \lambda)^{-1/2}w(\theta)\varphi_\theta\langle\varphi_\theta, \cdot\rangle_{\mathcal{H}_\Theta}(T + \lambda)^{-1/2}\right] = (T + \lambda)^{-1/2}T(T + \lambda)^{-1/2},$$

where the last equality, $\mathbb{E}_{\theta \sim \mathbb{Q}} w(\theta)\varphi_\theta\langle\varphi_\theta, \cdot\rangle_{\mathcal{H}_\Theta} = T$, holds since $\mathbb{E}_{\theta \sim \mathbb{Q}}[w(\theta)\varphi_\theta\langle\varphi_\theta, \cdot\rangle_{\mathcal{H}_\Theta}]g(\theta') = \mathbb{E}_{\theta \sim \mathbb{Q}} w(\theta)k(\theta, \theta')g(\theta)$ for any $g \in \mathcal{H}_\Theta$. Therefore

$$S_1 = \left\|\frac{1}{M}\sum_{i=1}^M \xi_i - \mathbb{E}_{\theta \sim \mathbb{Q}} \mathbb{E}_{x_{1:N}^\theta \sim \mathbb{P}_\theta} \xi\right\|_{\mathrm{HS}}.$$

Then, by the Bernstein inequality for Hilbert space-valued random variables [Caponnetto and De Vito, 2007, Proposition 2], the claimed bound on $S_1$ holds if there exist $L > 0, \sigma > 0$ such that

$$\mathbb{E}_{\theta \sim \mathbb{Q}} \mathbb{E}_{x_{1:N}^\theta \sim \mathbb{P}_\theta}\left[\|\xi - \mathbb{E}_{\theta \sim \mathbb{Q}} \mathbb{E}_{x_{1:N}^\theta \sim \mathbb{P}_\theta} \xi\|_{\mathrm{HS}}^m\right] \leq \frac{1}{2}m!\sigma^2 L^{m-2}$$

holds for all integer $m \geq 2$. We will show the condition holds. For convenience, denote $\mathbb{E}_\xi f(\xi) := \mathbb{E}_{\theta \sim \mathbb{Q}} \mathbb{E}_{x_{1:N}^\theta \sim \mathbb{P}_\theta} f(\xi)$. First, suppose $\xi'$ is an independent copy of $\xi$. Identically to the proof of Gogolashvili et al. [2023, Lemma 18], it holds that

$$\mathbb{E}_\xi [\|\xi - \mathbb{E}_\xi \xi\|_{\mathrm{HS}}^m] \overset{(A)}{\leq} \mathbb{E}_\xi \mathbb{E}_{\xi'} [\|\xi - \xi'\|_{\mathrm{HS}}^m] \overset{(B)}{\leq} 2^{m-1} \mathbb{E}_\xi \mathbb{E}_{\xi'} [\|\xi\|_{\mathrm{HS}}^m + \|\xi'\|_{\mathrm{HS}}^m] = 2^m \mathbb{E}_\xi \|\xi\|_{\mathrm{HS}}^m$$

where $(A)$ holds by Jensen inequality, and $(B)$ uses the fact that $|a + b|^m \leq 2^{m-1}(|a|^m + |b|^m)$. Next, observe that

$$\mathbb{E}_\xi \|\xi\|_{\mathrm{HS}}^m = \mathbb{E}_{\theta \sim \mathbb{Q}} \mathbb{E}_{x_{1:N}^\theta \sim \mathbb{P}_\theta} \|(T + \lambda)^{-1/2}(w(\theta) + \varepsilon(\theta; x_{1:N}^\theta))\varphi_\theta\langle\varphi_\theta, \cdot\rangle_{\mathcal{H}_\Theta}(T + \lambda)^{-1/2}\|_{\mathrm{HS}}^m$$

$$\overset{(A)}{=} \mathbb{E}_{\theta \sim \mathbb{Q}}\left[\mathbb{E}_{x_{1:N}^\theta \sim \mathbb{P}_\theta}\left[(w(\theta) + \varepsilon(\theta; x_{1:N}^\theta))^m\right] \|(T + \lambda)^{-1/2}\varphi_\theta\langle\varphi_\theta, \cdot\rangle_{\mathcal{H}_\Theta}(T + \lambda)^{-1/2}\|_{\mathrm{HS}}^m\right]$$

$$\overset{(B))}{\leq} \mathbb{E}_{\theta \sim \mathbb{Q}}\left[\|(T + \lambda)^{-1/2}\varphi_\theta\langle\varphi_\theta, \cdot\rangle_{\mathcal{H}_\Theta}(T + \lambda)^{-1/2}\|_{\mathrm{HS}}^m\right]$$

$$\overset{(C)}{\leq} \kappa^m\lambda^{-m}$$

$$= \frac{1}{2}m!\sigma^2 L^{m-2}$$

where $L = \sigma = \kappa\lambda^{-1}$, $(A)$ holds by linearity of norms as $w(\theta) + \varepsilon(\theta; x^\theta_{1:N}) \in \mathbb{R}$, $(B)$ holds since $\sigma^2_{\mathrm{BQ}}(\theta; x^\theta_{1:N}) \geq 0$, so

$$\left(w(\theta) + \varepsilon(\theta; x^\theta_{1:N})\right)^m = \left(\frac{\lambda_\Theta}{\lambda_\Theta + \sigma^2_{\mathrm{BQ}}(\theta; x^\theta_{1:N})}\right)^m \leq 1.$$

To show $(C)$ holds, take $\{e_j\}_{j=1}^\infty$ to be some orthonormal basis of $\mathcal{H}_\Theta$. Then,

$$\begin{aligned}
\|(T+\lambda)^{-1/2}\varphi_\theta\langle\varphi_\theta, \cdot\rangle_{\mathcal{H}_\Theta}(T+\lambda)^{-1/2}\|^2_{\mathrm{HS}} &= \sum_{j=1}^\infty \|(T+\lambda)^{-1/2}\varphi_\theta\langle\varphi_\theta, \cdot\rangle_{\mathcal{H}_\Theta}(T+\lambda)^{-1/2}e_j\|^2_{\mathcal{H}_\Theta}\\
&= \sum_{j=1}^\infty \|(T+\lambda)^{-1/2}\varphi_\theta\langle\varphi_\theta, (T+\lambda)^{-1/2}e_j\rangle_{\mathcal{H}_\Theta}\|^2_{\mathcal{H}_\Theta}\\
&\leq \|(T+\lambda)^{-1/2}\varphi_\theta\|^2_{\mathcal{H}_\Theta}\sum_{j=1}^\infty\langle\varphi_\theta, (T+\lambda)^{-1/2}e_j\rangle^2_{\mathcal{H}_\Theta}\\
&= \|(T+\lambda)^{-1/2}\varphi_\theta\|^2_{\mathcal{H}_\Theta}\sum_{j=1}^\infty\langle(T+\lambda)^{-1/2}\varphi_\theta, e_j\rangle^2_{\mathcal{H}_\Theta}\\
&\overset{(A)}{\leq} \|(T+\lambda)^{-1/2}\varphi_\theta\|^2_{\mathcal{H}_\Theta}\|(T+\lambda)^{-1/2}\varphi_\theta\|^2_{\mathcal{H}_\Theta}\\
&= \langle(T+\lambda)^{-1}\varphi_\theta, \varphi_\theta\rangle^2_{\mathcal{H}_\Theta}\\
&\leq \kappa^2\lambda^{-2},
\end{aligned}$$

where $(A)$ holds by Bessel's inequality. Then by the Bernstein inequality in Caponnetto and De Vito [2007, Proposition 2], it holds that

$$S_1 \leq \frac{2\kappa}{\lambda\sqrt{M}}\left(\frac{1}{\sqrt{M}} + 1\right)\log(4/\delta) \leq \frac{4\kappa}{\lambda\sqrt{M}}\log(4/\delta),$$

with probability at least $1 - \delta/2$. Finally as $\lambda\sqrt{M} > (16/3)\kappa\log(4/\delta)$, $S_1 < 3/4$. $\qquad\square$

Next, we bound another relevant term that also quantifies the "gap" between $T$ and $\hat{T}$. Unlike $S_1$, we will not require it to be upper bounded by 1—as it will only appear in Theorem 5 as a bounding term to the error.

**Lemma 2.** *Suppose Assumptions B2.(b), B4.(c) hold, and the operators $T, \hat{T}$ be as defined in Appendix A.3.1. Then, with probability greater than $1 - \delta/2$,*

$$S_2 := \|(T+\lambda)^{-1/2}(T-\hat{T})\|_{\mathrm{HS}} \leq \frac{4\kappa}{\sqrt{\lambda M}}\log(4/\delta).$$

*Additionally, if $\lambda\sqrt{M} > (4/C_6)\kappa\log(4/\delta)$, it holds that $S_2 < C_6\sqrt{\lambda}$.*

*Proof.* The proof is identical to that of Lemma 1. $\qquad\square$

The last auxiliary result we need is a simple bound on the following operator norm.

**Lemma 3.** *Let $T : \mathcal{H}_\Theta \to \mathcal{H}_\Theta$ be a positive operator. Then,*

$$\|T(T+\lambda)^{-1}\|_{\mathrm{op}} \leq 1.$$

*Proof.* Since $T$ is positive, for any $f \in \mathcal{H}_\Theta$ it holds that $\|Tf\|_{\mathcal{H}_\Theta} \leq \|(T+\lambda)f\|_{\mathcal{H}_\Theta}$. Therefore, by taking $f = (T+\lambda)^{-1}g$, we get that

$$\|T(T+\lambda)^{-1}\|_{\mathrm{op}} = \sup_{\substack{g\in\mathcal{H}_\Theta\\\|g\|_{\mathcal{H}_\Theta}=1}} \|T(T+\lambda)^{-1}g\|_{\mathcal{H}_\Theta} \leq \sup_{\substack{g\in\mathcal{H}\\\|g\|_{\mathcal{H}_\Theta}=1}} \|(T+\lambda)(T+\lambda)^{-1}g\|_{\mathcal{H}_\Theta} = 1.$$

$\qquad\square$

### A.3.3 Convergence of the noisy IW-KRR estimator

With the auxiliary results in place, we now extend Gogolashvili et al. [2023, Theorem 4] to the case of noisy weights. We start by establishing convergence in $\mathcal{L}^2(\Theta, \mathbb{Q}_w)$, where $\mathbb{Q}_w$ is the measure defined as $\mathbb{Q}_w(A) = \int_A w(\theta)\mathbb{Q}(\mathrm{d}\theta)$ that must be finite and positive. By [Fremlin, 2000, Proposition 232D], for $\mathbb{Q}_w(A)$ to be a finite positive measure, it is sufficient for $w(\theta)$ to be continuous and bounded. By their definition in Equation (A.7),

$$w(\theta) = \mathbb{E}_{y_{1:N}^\theta \sim \mathbb{P}_\theta} \frac{\lambda_\Theta}{\lambda_\Theta + \sigma_{\mathrm{BQ}}^2(\theta; y_{1:N}^\theta)}$$

the weights are bounded by 1, and are continuous in $\theta$ if $p_\theta$ is continuous in $\theta$ (as the dependance of $\sigma_{\mathrm{BQ}}^2(\theta; y_{1:N}^\theta)$ on $\theta$ for a fixed $y_{1:N}^\theta$ is, again, only through $p_\theta$ appearing under integrals and in polynomials). The continuity of $p_\theta$ holds as, by B0.(b), B4.(b), $p_\theta$ lies in a Sobolev space of smoothness over $p/2$, and therefore by Sobolev embedding theorem [Adams and Fournier, 2003, Theorem 4.12] $p_\theta$ is continuous in $\theta$.

**Theorem 5.** *Suppose Assumptions B0.(b), B1.(b), B2.(a), B2.(b), and B4.(c) hold, and $\lambda\sqrt{M} > (4/C_6)\kappa \log(4/\delta)$ for some $C_6 \leq 1$. Then,*

$$\|I_{\mathrm{CBQ}} - I\|_{\mathcal{L}^2(\Theta, \mathbb{Q}_w)} \leq (1 - C_6)^{-1}\left(C_6\sqrt{\lambda} + 1\right)\|I_{\mathrm{BQ}} - I\|_{\mathcal{H}_\Theta} + \left(\frac{8(1-C_6)^{-1}\kappa}{\sqrt{\lambda M}}\log(4/\delta) + \sqrt{\lambda}\right)\|I\|_{\mathcal{H}_\Theta}.$$

*Proof.* First, note that $I_{\mathrm{CBQ}}(\theta) = (\hat{T} + \lambda)^{-1}\hat{T}[I_{\mathrm{BQ}}]$, which can be checked easily by seeing that $(\hat{T} + \lambda)I_{\mathrm{CBQ}}(\theta) = \hat{T}[I_{\mathrm{BQ}}]$ for the *weighted* operator $\hat{T}$ as defined in Appendix A.3.1 and $I_{\mathrm{CBQ}}$ as defined in Equation (A.6). Then, for $I_\lambda = (T + \lambda)^{-1}T[I]$, by triangle inequality the error is bounded as

$$\|I_{\mathrm{CBQ}} - I\|_{\mathcal{L}^2(\Theta, \mathbb{Q}_w)} \leq \|I_{\mathrm{CBQ}} - I_\lambda\|_{\mathcal{L}^2(\Theta, \mathbb{Q}_w)} + \|I_\lambda - I\|_{\mathcal{L}^2(\Theta, \mathbb{Q}_w)} \tag{A.9}$$

The second term, $\|I_\lambda - I\|_{\mathcal{L}^2(\Theta, \mathbb{Q}_w)}^2$, can be bounded in terms of $\lambda$ as

$$\begin{aligned}
\|I_\lambda - I\|_{\mathcal{L}^2(\Theta, \mathbb{Q}_w)} &= \|\lambda(T + \lambda)^{-1}[I]\|_{\mathcal{L}^2(\Theta, \mathbb{Q}_w)} \\
&= \|\lambda T^{1/2}(T + \lambda)^{-1}[I]\|_{\mathcal{H}_\Theta} \\
&\overset{(A)}{\leq} \lambda\|T(T + \lambda)^{-1}\|_{\mathrm{op}}^{1/2}\|(T + \lambda)^{-1/2}\|_{\mathrm{op}}\|I\|_{\mathcal{H}_\Theta} \\
&\leq \sqrt{\lambda}\|I\|_{\mathcal{H}_\Theta}, \tag{A.10}
\end{aligned}$$

where $(A)$ holds by Lemma 3 and $T$ being a positive operator. Next, the $\mathcal{L}^2(\Theta, \mathbb{Q}_w)$ norm between $I_{\mathrm{CBQ}} - I_\lambda$ can be bounded as

$$\begin{aligned}
\|I_{\mathrm{CBQ}} - I_\lambda\|_{\mathcal{L}^2(\Theta, \mathbb{Q}_w)} &= \|T^{1/2}(I_{\mathrm{CBQ}} - I_\lambda)\|_{\mathcal{H}_\Theta} = \|T^{1/2}((\hat{T} + \lambda)^{-1}\hat{T}[I_{\mathrm{BQ}}] - (T + \lambda)^{-1}T[I])\|_{\mathcal{H}_\Theta} \\
&\overset{(A)}{\leq} \|T(T + \lambda)^{-1}\|_{\mathrm{op}}^{1/2}\|(\mathrm{Id} - (T + \lambda)^{-1/2}(T - \hat{T})(T + \lambda)^{-1/2})^{-1}\|_{\mathrm{op}} \\
&\quad \times \Big(\|(T + \lambda)^{-1/2}(\hat{T}[I_{\mathrm{BQ}}] - T[I])\|_{\mathcal{H}_\Theta} \\
&\qquad\qquad + \|(T + \lambda)^{-1/2}(T - \hat{T})(T + \lambda)^{-1}T[I]\|_{\mathcal{H}_\Theta}\Big) \\
&\overset{(B)}{\leq} \|T(T + \lambda)^{-1}\|_{\mathrm{op}}^{1/2}\|(\mathrm{Id} - (T + \lambda)^{-1/2}(T - \hat{T})(T + \lambda)^{-1/2})^{-1}\|_{\mathrm{op}} \\
&\quad \times \Big(\|(T + \lambda)^{-1/2}\hat{T}[I_{\mathrm{BQ}} - I]\|_{\mathcal{H}_\Theta} \\
&\qquad\qquad + \|(T + \lambda)^{-1/2}(T - \hat{T})[I]\|_{\mathcal{H}_\Theta} \\
&\qquad\qquad + \|(T + \lambda)^{-1/2}(T - \hat{T})(T + \lambda)^{-1}T[I]\|_{\mathcal{H}_\Theta}\Big) \\
&=: U_0 \times U_1 \times (U_2 + U_3 + U_4),
\end{aligned}$$

where $\|\cdot\|_{\mathrm{op}}$ denotes the operator norm, $(A)$ holds by Gogolashvili et al. [2023, Lemma 17], and $(B)$ is an application of triangle inequality,

$$\|(T + \lambda)^{-1/2}(\hat{T}[I_{\mathrm{BQ}}] - T[I])\|_{\mathcal{H}_\Theta} \leq \|(T + \lambda)^{-1/2}\hat{T}[I_{\mathrm{BQ}} - I]\|_{\mathcal{H}_\Theta} + \|(T + \lambda)^{-1/2}(T - \hat{T})[I]\|_{\mathcal{H}_\Theta}.$$

We will bound the terms $U_0, U_1, U_2, U_3, U_4$, and the result will follow. First, we have that $U_0 = \|T(T+\lambda)^{-1}\|_{\text{op}}^{1/2} \leq 1$ by Lemma 3. To upper bound $U_1 = \|(\text{Id} - (T+\lambda)^{-1/2}(T-\hat{T})(T+\lambda)^{-1/2})^{-1}\|_{\text{op}}$ we may expand it as Neumann series, provided $\|(T+\lambda)^{-1/2}(T-\hat{T})(T+\lambda)^{-1/2})^{-1}\|_{\text{op}} < 1$. This condition holds as

$$\|(T+\lambda)^{-1/2}(T-\hat{T})(T+\lambda)^{-1/2}\|_{\text{op}} \overset{(A)}{\leq} \|(T+\lambda)^{-1/2}(T-\hat{T})(T+\lambda)^{-1/2}\|_{\text{HS}} \overset{(B)}{<} C_6 \leq 1,$$

where $(A)$ holds as the operator norm is bounded by the Hilbert-Schmidt norm, and $(B)$ by Lemma 1. Therefore,

$$\left\|(\text{Id} - (T+\lambda)^{-1/2}(T-\hat{T})(T+\lambda)^{-1/2})^{-1}\right\|_{\text{op}} \overset{(A)}{=} \left\|\sum_{i=0}^{\infty}\left((T+\lambda)^{-1/2}(T-\hat{T})(T+\lambda)^{-1/2}\right)^i\right\|_{\text{op}}$$

$$\overset{(B)}{\leq} \sum_{i=0}^{\infty}\left\|(T+\lambda)^{-1/2}(T-\hat{T})(T+\lambda)^{-1/2}\right\|_{\text{op}}^i$$

$$\overset{(C)}{\leq} \sum_{i=0}^{\infty}\left\|(T+\lambda)^{-1/2}(T-\hat{T})(T+\lambda)^{-1/2}\right\|_{\text{HS}}^i$$

$$\overset{(D)}{=} \left(1 - \left\|(T+\lambda)^{-1/2}(T-\hat{T})(T+\lambda)^{-1/2}\right\|_{\text{HS}}\right)^{-1}$$

$$\overset{(E)}{\leq} (1 - C_6)^{-1},$$

where $(A)$ holds by the Neumann series expansion, $(B)$ by the triangle inequality, and the fact that operator norm is sub-multiplicative for bounded operators, $(C)$ since the operator norm is bounded by the Hilbert-Schmidt norm, $(D)$ by the geometric series, and $(E)$ by Lemma 1.

To bound $U_2 = \|(T+\lambda)^{-1/2}\hat{T}[I_{\text{BQ}} - I]\|_{\mathcal{H}_\Theta}$, observe that

$$U_2 = \|(T+\lambda)^{-1/2}\hat{T}[I_{\text{BQ}} - I]\|_{\mathcal{H}_\Theta} \leq \|(T+\lambda)^{-1/2}\hat{T}\|_{\text{op}}\|I_{\text{BQ}} - I\|_{\mathcal{H}_\Theta}$$

$$\leq \left(\|(T+\lambda)^{-1/2}(T-\hat{T})\|_{\text{op}} + \|(T+\lambda)^{-1/2}T\|_{\text{op}}\right)\|I_{\text{BQ}} - I\|_{\mathcal{H}_\Theta}$$

$$\overset{(A)}{\leq} (S_2 + 1)\|I_{\text{BQ}} - I\|_{\mathcal{H}_\Theta},$$

where $(A)$ holds by Lemmas 2 and 3.

Both $U_3$ and $U_4$ are upper bounded by the $S_2$ term in Lemma 2, as

$$U_3 = \|(T+\lambda)^{-1/2}(T-\hat{T})[I]\|_{\mathcal{H}_\Theta} \leq \|(T+\lambda)^{-1/2}(T-\hat{T})\|_{\text{op}}\|I\|_{\mathcal{H}_\Theta} = S_2\|I\|_{\mathcal{H}_\Theta},$$

$$U_4 = \|(T+\lambda)^{-1/2}(T-\hat{T})(T+\lambda)^{-1}T[I]\|_{\mathcal{H}_\Theta} \leq \|(T+\lambda)^{-1/2}(T-\hat{T})\|_{\text{op}}\|(T+\lambda)^{-1}T\|_{\text{op}}\|I\|_{\mathcal{H}_\Theta} \overset{(A)}{\leq} S_2\|I\|_{\mathcal{H}_\Theta},$$

where $(A)$ holds by Lemma 3. Putting the upper bounds on $U_0, U_1, U_2, U_3, U_4$ together, we get

$$\|I_{\text{CBQ}} - I_\lambda\|_{\mathcal{L}^2(\Theta, \mathbb{Q}_w)} \leq U_0 \times U_1 \times (U_2 + U_3 + U_4) \leq 4\left((S_2 + 1)\|I_{\text{BQ}} - I\|_{\mathcal{H}_\Theta} + 2S_2\|I\|_{\mathcal{H}_\Theta}\right).$$

By applying the union bound, we get that that with probability at least $1 - \delta$,

$$\|I_{\text{CBQ}} - I_\lambda\|_{\mathcal{L}^2(\Theta, \mathbb{Q}_w)} \leq U_0 \times U_1 \times (U_2 + U_3 + U_4)$$

$$\leq (1 - C_6)^{-1}\left((S_2 + 1)\|I_{\text{BQ}} - I\|_{\mathcal{H}_\Theta} + 2S_2\|I\|_{\mathcal{H}_\Theta}\right)$$

$$\overset{(A)}{\leq} (1 - C_6)^{-1}\left(\left(C_6\sqrt{\lambda} + 1\right)\|I_{\text{BQ}} - I\|_{\mathcal{H}_\Theta} + \frac{8\kappa}{\sqrt{\lambda M}}\log(4/\delta)\|I\|_{\mathcal{H}_\Theta}\right),$$

where $(A)$ holds by Lemma 2. Inserting this and the bound in Equation (A.10) into Equation (A.9) gives

$$\|I_{\text{CBQ}} - I\|_{\mathcal{L}^2(\Theta, \mathbb{Q}_w)} \leq (1 - C_6)^{-1}\left(C_6\sqrt{\lambda} + 1\right)\|I_{\text{BQ}} - I\|_{\mathcal{H}_\Theta} + \left(\frac{8(1 - C_6)^{-1}\kappa}{\sqrt{\lambda M}}\log(4/\delta) + \sqrt{\lambda}\right)\|I\|_{\mathcal{H}_\Theta}.$$

$\square$

Finally, we use the $\mathcal{L}^2(\Theta, \mathbb{Q}_w)$ bound in Theorem 5 to establish a bound in $\mathcal{L}^2(\Theta, \mathbb{Q})$ in terms of the BQ variance.

**Corollary 1.** *Suppose Assumptions B0.(b), B1.(b), B2.(a), B2.(b), and B4.(c), and $\lambda\sqrt{M} > (4/C_6)\kappa\log(4/\delta)$ for some $C_6 \le 1$. Then*

$$\|I_{\mathrm{CBQ}} - I\|_{\mathcal{L}^2(\Theta,\mathbb{Q})} \le \left(1 + \frac{1}{c\sqrt{M}}\sup_{\theta\in\Theta}\mathbb{E}_{y_{1:N}^\theta \sim \mathbb{P}_\theta}\sigma^2_{\mathrm{BQ}}(\theta; y_{1:N}^\theta)\right)$$
$$\times \left((1 - C_6)^{-1}\left(C_6\sqrt{\lambda} + 1\right)\|I_{\mathrm{BQ}} - I\|_{\mathcal{H}_\Theta} + \left(\frac{8(1 - C_6)^{-1}\kappa}{\sqrt{\lambda M}}\log(4/\delta) + \sqrt{\lambda}\right)\|I\|_{\mathcal{H}_\Theta}\right)$$

*Proof.* Observe that for any $g \in \mathcal{L}^2(\Theta, \mathbb{Q})$, it holds that $\|g\|^2_{\mathcal{L}^2(\Theta,\mathbb{Q}_w)} \ge (\inf_{\theta\in\Theta} w(\theta)) \times \|g\|^2_{\mathcal{L}^2(\Theta,\mathbb{Q})}$. Then, since

$$w(\theta) = \mathbb{E}_{y_{1:N}^\theta \sim \mathbb{P}_\theta}\frac{\lambda_\Theta}{\lambda_\Theta + \sigma^2_{\mathrm{BQ}}(\theta; y_{1:N}^\theta)} \ge \frac{\lambda_\Theta}{\lambda_\Theta + \mathbb{E}_{y_{1:N}^\theta \sim \mathbb{P}_\theta}\sigma^2_{\mathrm{BQ}}(\theta; y_{1:N}^\theta)} = \frac{1}{1 + \lambda_\Theta^{-1}\mathbb{E}_{y_{1:N}^\theta \sim \mathbb{P}_\theta}\sigma^2_{\mathrm{BQ}}(\theta; y_{1:N}^\theta)},$$

the bound in Theorem 5, the definition of $\lambda$ in Equation (A.7), and Assumption B5.(a) give the desired statement. $\qquad\square$

## A.4 PROOF OF THEOREM 1

We are now ready to prove our main convergence result, which is a version of Theorem 1 for $\lambda_\mathcal{X} \ge 0$. We start by restating it for the convenience of the reader.

*Restatement of Theorem 2.* Suppose all technical assumptions in Appendix A.1 hold. Then for any $\delta \in (0, 1)$ there is an $N_0 > 0$ such that for any $N \ge N_0$, with probability at least $1 - \delta$ it holds that

$$\|I_{\mathrm{CBQ}} - I\|_{\mathcal{L}^2(\Theta,\mathbb{Q})} \le \left(1 + c^{-1}M^{-\frac{1}{2}}\left(\lambda_\mathcal{X} + C_2 N^{-1+2\varepsilon}\left(N^{-\frac{s_\mathcal{X}}{d}+\frac{1}{2}+\varepsilon} + C_3\lambda_\mathcal{X}\right)^2\right)\right)$$
$$\times \left(C_7(\delta)N^{-\frac{1}{2}+\varepsilon}\left(N^{-\frac{s_\mathcal{X}}{d}+\frac{1}{2}+\varepsilon} + C_5\lambda_\mathcal{X}\right) + C_8(\delta)M^{-\frac{1}{4}}\|I\|_{\mathcal{H}_\Theta}\right)$$

for any arbitrarily small $\varepsilon > 0$, constants $C_2, C_3, C_5, C_7(\delta) = O(1/\delta)$ and $C_8(\delta) = O(\log(1/\delta))$ independent of $N, M, \varepsilon$. $\qquad\square$

*Proof of Theorem 2.* By inserting Theorems 3 and 4 into Corollary 1 and applying the union bound, we get that the result holds with probability at least $1 - \delta$ and

$$C_7(\delta) = (1 - C_6)^{-1}\left(C_6 c^{\frac{1}{2}} + 1\right)C_4(2/\delta),$$
$$C_8(\delta) = \left(8c^{-\frac{1}{2}}(1 - C_6)^{-1}\kappa\log(4/\delta) + c^{\frac{1}{2}}\right).$$

$\qquad\square$

As discussed in the main text, convergence is fastest when the regulariser $\lambda_\mathcal{X}$ is set to 0; $\lambda_\mathcal{X} > 0$ ensures greater stability at the cost of a lower speed of convergence. For clarity we show how Theorem 1 in the main text follows from the more general Theorem 2 by setting $\lambda_\mathcal{X} = 0$.

*Proof of Theorem 1.* In Theorem 2, take $\lambda_\mathcal{X} = 0$. Then

$$\|I_{\mathrm{CBQ}} - I\|_{\mathcal{L}^2(\Theta,\mathbb{Q})} \le \left(1 + c^{-1}M^{-\frac{1}{2}}C_2 N^{-\frac{2s_\mathcal{X}}{d}+\varepsilon}\right) \times \left(C_7(\delta)N^{-\frac{s_\mathcal{X}}{d}+\varepsilon} + C_8(\delta)M^{-\frac{1}{4}}\|I\|_{\mathcal{H}_\Theta}\right).$$

As $\mathbb{Q}$ was assumed equivalent to the uniform distribution in Assumption B2.(a), the error in uniform measure is bounded by the error in $\mathbb{Q}$. Therefore, the result holds for

$$C_0(\delta) = \left(1 + c^{-1}C_2\right)C_7(\delta) = O(1/\delta),$$
$$C_1(\delta) = \left(1 + c^{-1}C_2\right)\|I\|_{\mathcal{H}_\Theta}C_8(\delta) = O(\log(1/\delta)).$$

$\qquad\square$

# APPENDIX B  PRACTICAL CONSIDERATIONS FOR CONDITIONAL BAYESIAN QUADRATURE

We now discuss important practical considerations which can have significant impact on the performance of CBQ. Firstly, in Appendix B.1 we discuss how to ensure a closed-form expression for kernel mean embeddings and initial errors of BQ estimators. Then, we discuss the selection of all kernel hyperparameters in Appendix B.2.

## B.1  TRACTABLE KERNEL MEANS

In the main text, we discussed the requirement for both BQ and CBQ that the kernel mean embedding $\mu$ and its integral (called initial error) are known in closed-form. A list of well-known pair can be found in Table 1 in [Briol et al., 2019] or the `ProbNum` package [Wenger et al., 2021]. However, even when none of these pairs are appropriate for the problem at hand, there are still multiple solutions:

- First, for a fixed $k$, when the embedding of $\mathbb{P}$ is intractable but the embedding of some other distribution $\mathbb{Q}$ is known, we can use the 'importance sampling trick' which consists of writing the integral as $I = \mathbb{E}_{X \sim \mathbb{P}}[f(X)] = \mathbb{E}_{X \sim \mathbb{Q}}[g(X)]$ where $g(x) = f(x)p(x)/q(x)$ and $p, q$ are the densities of $\mathbb{P}, \mathbb{Q}$. This allows us to use BQ on the integral of $g$, which is tractable by construction.

- Secondly, again for a fixed $k$ and assuming that we know the quantile function $\Phi^{-1}$ of the distribution $\mathbb{P}$ and that the embedding of the uniform distribution is available, we can use the 'inverse transform trick' which consists of writing $I = \mathbb{E}_{X \sim \mathbb{P}}[f(X)] = \mathbb{E}_{U \sim \mathbb{U}}[g(U)]$ where $g(u) = f(\Phi^{-1}(u))$ and $\mathbb{U}$ is a uniform distribution on some hypercube. Once again, BQ can now be applied to the transformed problem.

- Finally, for any distribution $\mathbb{P}$ whose density is known up to the normalisation constant (for example most posterior distributions), then specialised kernels with closed-form embeddings can be constructed. This is true of Stein reproducing kernels Anastasiou et al. [2023]. Suppose we have a distribution $\mathbb{P}$ with density $p : \mathcal{X} \to \mathbb{R}^+$ and a function $f : \mathcal{X} \to \mathbb{R}$ with the property that $\lim_{x \to \infty} p(x)f(x) = 0$. The Langevin Stein kernel $k : \mathcal{X} \times \mathcal{X} \to \mathbb{R}$ [Anastasiou et al., 2023] is given by:

$$k_p(x, x') := \nabla_x \log p(x)^\top k(x, x') \nabla_{x'} \log p(x') + \nabla_x \log p(x)^\top \nabla_{x'} k(x, x')$$
$$+ \nabla_{x'} \log p(x')^\top \nabla_x k(x, x') + \nabla_x \cdot \nabla_{x'} k(x, x'),$$

where $\nabla_x = (\partial/\partial x_1, \cdots, \partial/\partial x_d)^\top$ and $\nabla_x \cdot \nabla_{x'} k(x, x') = \sum_{i=1}^d \frac{\partial k(x, x')}{\partial x_i \partial x'_i}$.

The main advantage of using Stein kernel is that the mean embedding $\mu(x') = \int_{\mathcal{X}} k_p(x, x')p(x)dx = 0$ by construction. However, this means our GP prior on $f$ encodes beliefs that the function has mean zero. To weaken this, we can add a constant $c \in \mathbb{R}$; i.e $\tilde{k}_p(x, x') = k_p(x, x') + c$, so that the kernel mean embedding becomes $\mu(x') = c$. The constant $c$ can then be treated as a kernel hyperparameter and estimated alongside all other parameters.

## B.2  MODEL AND HYPERPARAMETER SELECTION

We now discuss our approach for model and hyperparameter selection for CBQ and baseline methods.

**Conditional Bayesian quadrature**   The hyperparameter selection for CBQ boils down to the choice of GP interpolation hyperparameters at stage 1 and the choice of GP regression hyperparameters at stage 2. To simplify this choice, we renormalise all our function values before performing GP regression and interpolation. This is done by first subtracting the empirical mean and then dividing by the empirical standard deviation. All of our experiments then use prior mean functions $m_\Theta$ and $m_{\mathcal{X}}$ which are zero functions, a reasonable choice given the function was renormalised using the empirical mean. This choice is made for simplicity, and we might expect further improvements in accuracy if more information is available.

The choice of covariance functions $k_{\mathcal{X}}$ and $k_\Theta$ is made on a case-by-case basis in order to both encode properties we expect the target functions to have, but also to ensure that the corresponding kernel mean is available in closed-form (as per the previous section). Once this is done, we typically still need to make a choice of hyperparameters for both kernel: lengthscales $l_{\mathcal{X}}, \ell_\Theta$ and amplitudes $A_{\mathcal{X}}, A_\Theta$. We also need to select the regularizer $\lambda_{\mathcal{X}}, \lambda_\Theta$. $\lambda_{\mathcal{X}}$ is fixed to be 0 as suggested

by Theorem 1. The rest of the hyperparameters are selected through empirical Bayes, which consists of maximising the log-marginal likelihood. For stage 1, the log-marginal likelihood can be written as [Rasmussen and Williams, 2006]:

$$L(l_\mathcal{X}, A_\mathcal{X}) = -\frac{1}{2} \log |k_\mathcal{X}(x_{1:N}, x_{1:N}; l_\mathcal{X}, A_\mathcal{X})| - \frac{N}{2} \log(2\pi)$$
$$- \frac{1}{2}(f(x_{1:N}) - m_\mathcal{X}(x_{1:N}))^\top \left(k_\mathcal{X}(x_{1:N}, x_{1:N}; l_\mathcal{X}, A_\mathcal{X}) + \lambda_\mathcal{X} \mathrm{Id}_N\right)^{-1} (f(x_{1:N}) - m_\mathcal{X}(x_{1:N})),$$

where $|\cdot|$ denotes the determinant of the matrix, and we write $k_\mathcal{X}(x_{1:N}, x_{1:N}; l_\mathcal{X}, A_\mathcal{X})$ to emphasise the hyperparameters used to compute the Gram matrix. The optimisation is implemented through a grid search over $[1.0, 10.0, 100.0, 1000.0]$ for the amplitude $A_\mathcal{X}$ and a grid search over $[0.1, 0.3, 1.0, 3.0, 10.0]$ for the lengthscale $l_\mathcal{X}$.

If $k_\mathcal{X}$ is a Stein reproducing kernel, we have an extra hyperparameter $c_\mathcal{X}$. In this case, we use stochastic gradient descent on the log-marginal likelihood to find the optimal value for $c_\mathcal{X}, l_\mathcal{X}, A_\mathcal{X}$, which is implemented with JAX autodiff library [Bradbury et al., 2018]. The reason we are using gradient based optimization instead of grid search for Stein kernel is that Stein kernel requires an accurate estimate of $c_\mathcal{X}$ to work well. In order to return accurate results, grid search would require finer grid which is very expensive, while gradient based methods would require good initialization to avoid getting stuck in local minima. Fortunately, since $c_\mathcal{X}$ indicates the mean of functions in the RKHS, we know that $c_\mathcal{X} = 0$ is a good initialisation point since we have subtracted the empirical mean when normalising.

Additionally, it is important to note that we could technically use $T$ different kernels $k_\mathcal{X}^1, \cdots, k_\mathcal{X}^T$ for each integral in stage 1. However, the hyperparameters of each kernel $k_\mathcal{X}^t$ would need to be selected using empirical Bayes under the observations $x_{1:N}^t$, which means we would need to repeat the above optimization $T$ times. In practice, when performing initial experiments, we observed that the estimated hyperparameters were very similar. Our strategy is therefore to select the hyperparameters of $k_\mathcal{X}^1$ and subsequently reuse them across all $T$ integrals in stage 1. This is done for computational reasons, and we expect CBQ to show better performances if hyperparameters are optimised separately.

For the kernel $k_\Theta$, we also select the hyperparameters by maximising the log-marginal likelihood:

$$L(l_\Theta, A_\Theta) = -\frac{1}{2} \log |k_\Theta(\theta_{1:T}, \theta_{1:T}; l_\Theta, A_\Theta)| - \frac{T}{2} \log(2\pi)$$
$$- \frac{1}{2}(I_{\mathrm{BQ}}(\theta_{1:T}) - m_\Theta(\theta_{1:T}))^\top \left(k_\Theta(\theta_{1:T}, \theta_{1:T}; l_\Theta, A_\Theta) + \left(\lambda_\Theta + \sigma_{\mathrm{BQ}}^2(\theta_{1:T})\right) \mathrm{Id}_T\right)^{-1} (I_{\mathrm{BQ}}(\theta_{1:T}) - m_\Theta(\theta_{1:T})).$$

Similar to above, we also do a grid search over $[1.0, 10.0, 100.0, 1000.0]$ for amplitude $A_\Theta$, a grid search over $[0.1, 0.3, 1.0, 3.0, 10.0]$ for lengthscale $l_\Theta$ and a grid search over $[0.01, 0.1, 1.0]$ for $\lambda_\Theta$, so we select the value that gives the largest log-marginal likelihood.

**Least-squares Monte Carlo**    LSMC implements Monte Carlo in the first stage and polynomial regression in the second stage. In the second stage, the hyperparameters include the regularisation coefficient $\lambda_\Theta$ and the order of the polynomial $p \in \{1, 2, 3, 4\}$. These hyperaparameters are also selected with grid search to give the lowest RMSE on a separate held out validation set.

**Kernel least-squares Monte Carlo**    KLSMC implements Monte Carlo in the first stage and kernel ridge regression in the second stage. In the second stage, the hyperparameters are analogous to the hyperparameters in the second stage of CBQ, namely $A_\Theta, l_\Theta, \lambda_\Theta$. These hyperaparameters are selected with grid search to give the lowest RMSE on a separate held out validation set.

**Importance sampling**    For IS, there are no hyperparameters to select.

## APPENDIX C    ADDITIONAL EXPERIMENTS

We now provide detailed description of all experiments in the main text, as well as further results and ablation studies. All figures reported in the paper are created using the median values obtained from 20 separate runs with different random seeds. Standard error is shown as shaded area around the median.

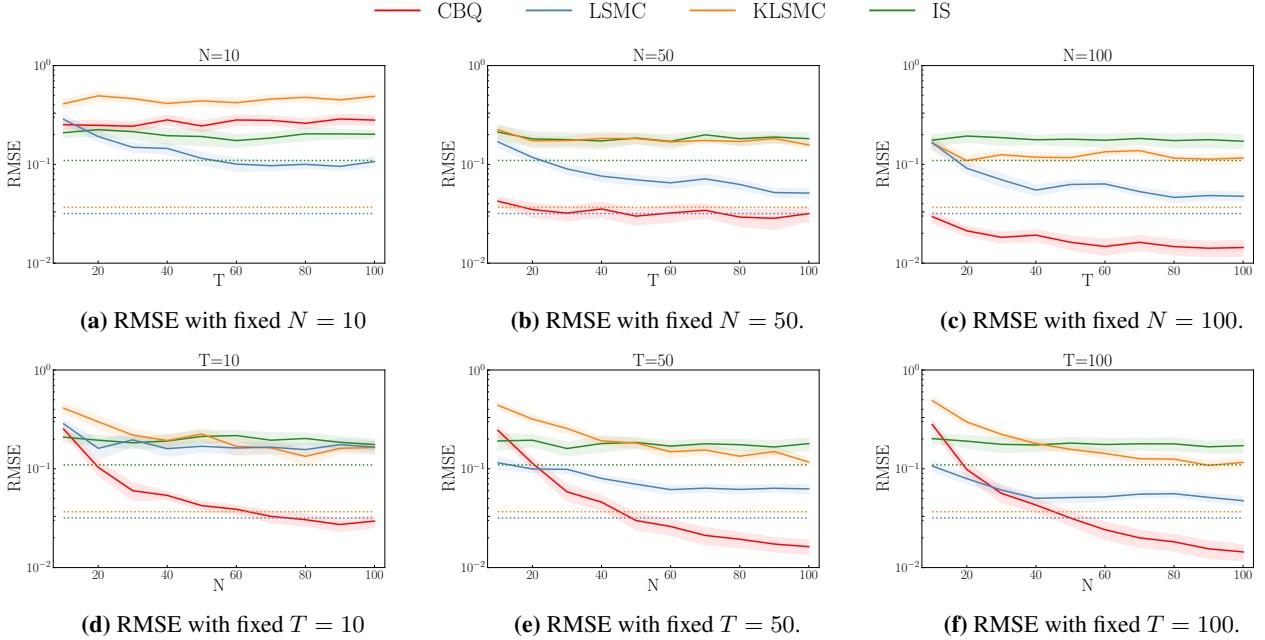

**Figure 7:** *Bayesian sensitivity analysis for linear models.* **First Row:** Dimension $d = 2$ with fixed $N = 10, 50, 100$ and increasing $T$. **Second Row:** Dimension $d = 2$ with fixed $T = 10, 50, 100$ and increasing $N$. The intergral is $f(x) = x^\top x$.

## C.1 SYNTHETIC EXPERIMENT: BAYESIAN SENSITIVITY ANALYSIS FOR LINEAR MODELS

### C.1.1 Experimental Setting

In this synthetic experiment, we do sensitivity analysis on the hyperparameters in Bayesian linear regression. The observational data for the linear regression are $Y \in \mathbb{R}^{m \times d}$, $Z \in \mathbb{R}^m$ with $m$ being the number of observations and $d$ being the dimension. We use $x$ to denote the regression weight; this is unusual but is done so as to keep the notation consistent with the main text. By placing a $\mathcal{N}(x; 0, \theta \mathrm{Id}_d)$ prior on the regression weights $x \in \mathbb{R}^d$ with $\theta \in (1, 3)^d$, and assuming independent $\mathcal{N}(0, \eta)$ observation noise for some known $\eta > 0$, we can obtain (via conjugacy) a multivariate Gaussian posterior $\mathbb{P}_\theta$ whose mean and variance have a closed form expression [Bishop, 2006].

$$\mathbb{P}_\theta = \mathcal{N}(\tilde{m}, \tilde{\Sigma}), \quad \tilde{\Sigma}^{-1} = \frac{1}{\theta}\mathrm{Id}_d + \eta Y^\top Y, \quad \tilde{m} = \eta \tilde{\Sigma} Y^\top Z.$$

We can then analyse sensitivity by computing the conditional expectation $I(\theta) = \int_{\mathcal{X}} f(x) \mathbb{P}_\theta(dx)$ of some quantity of interest $f$. For example, if $f(x) = x^\top x$, then $I(\theta)$ is the second moment of the posterior and the results are already reported in the main text. If $f(x) = x^\top y^*$ for some new observation $y^*$, then $I(\theta)$ is the predictive mean. In these simple settings, $I(\theta)$ can be computed analytically, making this a good synthetic example for benchmarking. We sample parameter values $\theta_{1:T}$ from a uniform distribution $\mathbb{Q} = \mathrm{Unif}(\Theta)$ where $\Theta = (1, 3)^d$, and for each such parameter $\theta_t$, we obtain $N$ observations $x_{1:N}^t$ from $\mathbb{P}_{\theta_t}$. In total, we have $N \times T$ samples.

For conditional Bayesian quadrature (CBQ), we need to carefully choose two kernels $k_\Theta$ and $k_{\mathcal{X}}$. Firstly, we choose the kernel $k_{\mathcal{X}}$ to be an isotropic Gaussian kernel: $k(x, x') = A_{\mathcal{X}} \exp(-\frac{1}{2l_{\mathcal{X}}^2}(x - x')^\top (x - x'))$ for the purpose that the Gaussian kernel mean embedding has a closed form under the Gaussian posterior $\mathbb{P}_\theta$:

$$\mu_\theta(x) = A_{\mathcal{X}} \left| Id_d + l_{\mathcal{X}}^{-2} \tilde{\Sigma} \right|^{-1/2} \exp\left(-\frac{1}{2}(x - \tilde{m})^\top (\tilde{\Sigma} + l_{\mathcal{X}}^2 \mathrm{Id}_d)^{-1}(x - \tilde{m})\right) \tag{C.11}$$

In addition, the integral of the kernel mean embedding $\mu_\theta$ (known as the initial error) also has a closed form $\int_{\mathcal{X}} \mu_\theta(x) \mathbb{P}_\theta(dx) = A_{\mathcal{X}} l_{\mathcal{X}} / \sqrt{|l_{\mathcal{X}}^2 \mathrm{Id}_d + 2\tilde{\Sigma}|}$.

This leaves us with a choice for $k_\Theta$. In this synthetic setting, we know that $I(\theta)$ is infinitely times differentiable, but we

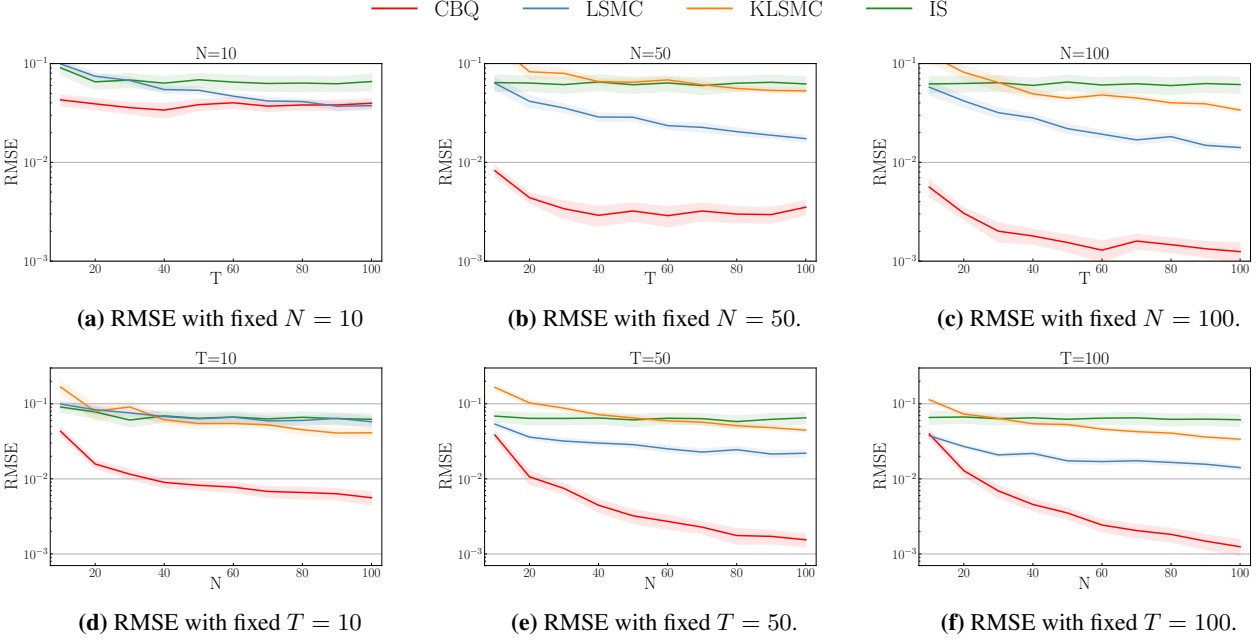

**Figure 8:** *Bayesian sensitivity analysis for linear models.* **First Row:** Dimension $d = 2$ with fixed $N = 10, 50, 100$ and increasing $T$. **Second Row:** Dimension $d = 2$ with fixed $T = 10, 50, 100$ and increasing $N$. The intergral is $f(x) = x^\top y^*$.

opt for Matérn-3/2 kernel $k_\Theta(\theta, \theta') = A_\Theta(1 + \sqrt{3}|\theta - \theta'|/l_\Theta) \exp(-\sqrt{3}|\theta - \theta'|/l_\Theta)$ to encode a more conservative prior information on the smoothness of $I(\theta)$.

### C.1.2    Assumptions from Theorem 1

We would like to check whether the assumptions made in Theorem 1 hold in this experiment.

- A1: Although $\mathcal{X} = \mathbb{R}$ is not a compact domain, $\mathbb{P}_\theta$ is a Gaussian distribution so the probability mass outside a large compact subset of $\mathcal{X}$ decays exponentially. $\Theta = (1, 3)^d$ is a compact domain. A1 is therefore approximately satisfied.

- A2: A2 is satisfied due to the sampling mechanism of $\theta_{1:T}$ and $\{x_{1:N}^t\}_{t=1}^T$.

- A3: $\mathbb{Q}$ is a uniform distribution so its density $q$ is constant and hence upper bounded and strictly positive. $\mathbb{P}_\theta$ is a Gaussian distribution so its density $p_\theta$ is strictly positive on a compact and large domain with finite second moment. A3 is approximately satisfied.

- A4: Both $f(x)$ and $I(\theta)$ are infinitely times differentiable, so $s_I = s_f = \infty$. Although $k_\mathcal{X}$ is Gaussian kernel which does not satisfy the assumption of Theorem 1, we have ablation study in Appendix C.7 showing similar performance when $k_\mathcal{X}$ is Matérn-3/2 kernel so $s_\mathcal{X} = \frac{3}{2} + \frac{d}{2}$, and $k_\Theta$ is Matérn-3/2 kernel so $s_\Theta = \frac{3}{2} + \frac{d}{2}$, where $d$ is the dimension. A4 is satisfied.

- A5: $\lambda_\mathcal{X}$ is picked to be 0 and $\lambda_\Theta$ is found via grid search among $\{0.01, 0.1, 1.0\}$. A5 is satisfied.

### C.1.3    Additional Experimental Results

We now provide additional experimental results for Bayesian sensitivity analysis in linear models. Figure 7 provides the result when the integrand is chosen to be $f(x) = x^\top x$ so $I(\theta)$ represents the posterior second moment, and Figure 8 provides the result when the integrand is chosen to be $f(x) = x^\top y^*$ so $I(\theta)$ represents the predictive mean. We can see that CBQ has demonstrated consistent smaller RMSE for both tasks under the same number of samples and faster convergence rate compared to all other baseline methods. The conclusions that we draw from the main text also hold for different values of $N$ and $T$. By comparing the performance of CBQ and KLSMC, where the second stage of both methods are identical, and the main difference lies in the first stage, we believe that CBQ shows better performances mainly due to using Bayesian

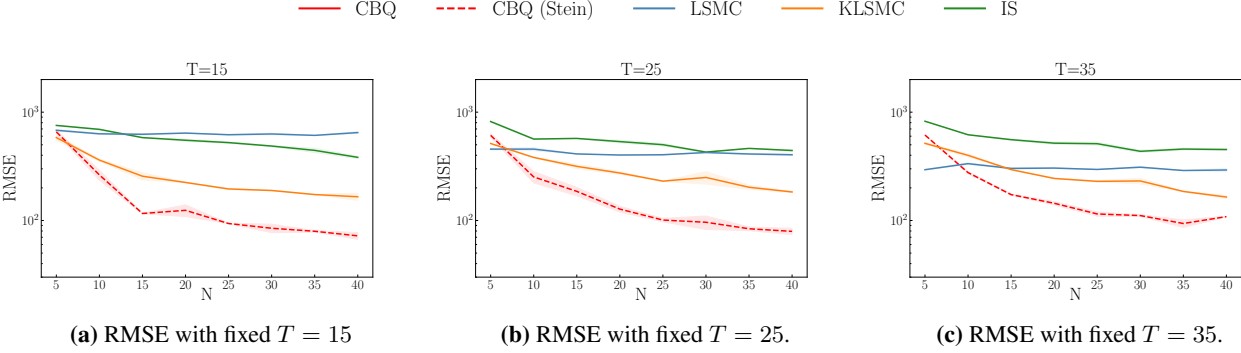

**Figure 9:** *Bayesian sensitivity analysis for SIR model. $T = 15, 25, 35$ and increasing $N$.*

quadrature instead of Monte Carlo in the first stage. Also by comparing the first and second row in both Figure 7 and Figure 8, we can confirm the theory we proved in Appendix A that CBQ has a faster convergence rate in $N$ than in $T$.

In general, CBQ is more computationally expensive than baselines (KLSMC, LSMC and IS), so in this simple setting it is more efficient to spend more budget on obtaining more samples. Nonetheless, in scenarios where the expense of sample collection constitutes a significant fraction of the computational budget, or when the evaluation of the integrand proves to be highly costly, it becomes more cost-effective to spend a larger share of the budget towards CBQ. For example, sampling can become expensive easily when the prior and likelihood are not conjugate, so Markov chain Monte Carlo methods are needed to sample from unnormalized posterior. Also, we show in the next section Appendix C.2 a real world example when sampling is particularly costly and hence using CBQ is overall more efficient.

## C.2 BAYESIAN SENSITIVITY ANALYSIS FOR SUSCEPTIBLE-INFECTIOUS-RECOVERED (SIR) MODEL

### C.2.1 Experimental Setting

The SIR model is commonly used to simulate the dynamics of infectious diseases through a population Kermack and McKendrick [1927]. It divides the population into three sections. Susceptibles (S) represents people who are not infected but can be infected after getting contact with an infectious individual. Infectious (I) represents people who are currently infected and can infect susceptible individuals. Recovered (R) represents individuals who have been infected and then removed from the disease, either by recovering or dying. The dynamics are governed by a system of ordinary differential equations (ODE) as below.

$$\frac{\mathrm{d}S}{\mathrm{d}r} = -xSI, \quad \frac{\mathrm{d}I}{\mathrm{d}r} = xSI - \gamma I, \quad \frac{\mathrm{d}R}{\mathrm{d}r} = \gamma I$$

with $x$ being the infection rate, $\gamma$ being the recovery rate and $r$ is the time. The solution to the SIR model would be a vector of $(N_I^r, N_S^r, N_R^r)$ representing the number of infectious, susceptibles and recovered at day $r$.

In this experiment, we assume that the recovery rate $\gamma$ is fixed and we place a Gamma prior distribution on $x$; i.e. $\mathbb{P}_\theta = \mathrm{Gamma}(\theta, \xi)$ where $\theta$ represents the initial belief of the infection rate deduced from the study of the virus in the laboratory at the beginning of the outbreak, and $\xi$ represents the amount of uncertainty on the initial belief. We fix the parameter $\xi = 10$, the total population is set to be $10^6$ and the recovery rate $\gamma = 0.05$. The target of interest is the expected peak number of infected individuals under the prior distribution on $x$:

$$I(\theta) = \mathbb{E}_x \left[ \max_r N_I^r(x) \mid \theta \right] = \int_{\mathcal{X}} \max_r N_I^r(x) \mathbb{P}_\theta(dx)$$

with the integrand $f(x) = \max_r N_I^r(x)$. We are interested in the sensitivity analysis of the shape parameter $\theta$ to the final estimate of the expected peak number of infected individuals. The initial belief of the infection rate $\theta_{1:T}$ are sampled from the uniform distribution $\mathbb{Q} = \mathrm{Unif}(2, 9)$ and then $N$ number of $x_{1:N}$ are sampled from $\mathbb{P}_{\theta_t} = \mathrm{Gamma}(\theta_t, \xi)$. In this setting, sampling $x$ is very expensive as it necessarily involves solving the system of SIR ODEs, which can be very slow as the discretization step gets finer. In the middle panel of Figure 5, we have shown that obtaining one sample from SIR ODEs under discretization time step $\tau = 0.1$ takes around 3.0s, whereas running the whole CBQ algorithm takes 1.0s, not to

mention that sampling from SIR ODEs need to be repeated $N \times T$ times. Therefore, using CBQ is ultimately more efficient overall within the same period of time.

For CBQ, we need to carefully choose two kernels $k_\Theta$ and $k_\mathcal{X}$. First we choose $k_\mathcal{X}$, we use Matérn-3/2 as the base kernel and then apply a Langevin Stein operator to both arguments of the base kernel to obtain $k_\mathcal{X}$. The reason we use a Langevin Stein kernel is that Stein kernel gives an RKHS which is a subset on the Sobolev space with one order less smoothness than the base kernel, and since the smoothness of the integrand $f(x) = \max_r N_I^r(x)$ is unknown, using a Stein kernel enforces weaker prior information than Matérn-3/2. Furthermore, the kernel mean embedding of a Stein kernel $\mu(x)$ is a constant $c$ by construction as per the discussion in Appendix B. The initial error is also a constant $c$ by construction. Then we choose $k_\Theta$. Since $I(\theta)$ represents the peak number of infections so $I(\theta)$ is expected to be smooth and continuous, and hence we choose $k_\Theta$ as Matérn-3/2 kernel. All hyperparameters in $k_\mathcal{X}$ and $k_\Theta$ are selected according to Appendix B.2. We use a MC estimator with 5000 samples as the pseudo ground truth and evaluate the RMSE across all methods.

### C.2.2 Assumptions from Theorem 1

We would like to check whether the assumptions made in Theorem 1 hold in this experiment.

- A1: Although $\mathcal{X} = \mathbb{R}^+$ is not a compact domain, $\mathbb{P}_\theta$ is a Gamma distribution so the probability mass outside a large compact subset of $\mathcal{X}$ around the origin decays exponentially. $\Theta = (2, 9)^d$ is a compact domain. A1 is approximately satisfied.

- A2: A2 is satisfied due to the sampling mechanism of $\theta_{1:T}$ and $\{x_{1:N}^t\}_{t=1}^T$.

- A3: $\mathbb{Q}$ is a uniform distribution so its density $q$ is constant and hence upper bounded and strictly positive. $\mathbb{P}_\theta$ is a Gamma distribution so its density $p_\theta$ is strictly positive within a large compact subset of $\mathcal{X}$ and has finite second moment. A3 is approximately satisfied.

- A4: $f(x) = \max_r N_I^r(x)$ is the maximum number of infections so $f(x)$ is not necessarily smooth. $I(\theta)$ represents the peak number of infections with varying initial estimate of the infection rate, so $I(\theta)$ is smooth and continuous with $s_I \leq 1$. $k_\mathcal{X}$ is Stein kernel with Matern-3/2 kernel as the base, so the corresponding RKHS will have functions which are rough (i.e. of smoothness $1/2$) but is only a subset of a Sobolev space. In addition, $k_\Theta$ is Matern-3/2 kernel so $s_\Theta = \frac{3}{2} + \frac{1}{2} = 2$. It is therefore unclear if A4 is satisfied.

- A5: $\lambda_\mathcal{X}$ is picked to be 0 and $\lambda_\Theta$ is found via grid search among $\{0.01, 0.1, 1.0\}$. A5 is satisfied.

### C.2.3 Additional Experimental Results

We report more results in Figure 9 with fixed $T = 15, 25, 35$ and increasing $N$, to showcase that CBQ consistently exhibits smaller RMSE than baseline methods. The conclusions that we draw from the main text also hold for different values of N and T for this experiment.

## C.3 OPTION PRICING IN MATHEMATICAL FINANCE

### C.3.1 Experimental Setting

In this experiment, we consider specifically an asset whose price $S(\tau)$ at time $\tau$ follows the Black-Scholes formula $S(\tau) = S_0 \exp\left(\sigma W(\tau) - \sigma^2 \tau / 2\right)$ for $\tau \geq 0$, where $\sigma$ is the underlying volatility, $S_0$ is the initial price and $W$ is the standard Brownian motion. The financial derivative we are interested in is a butterfly call option whose payoff at time $\tau$ can be expressed as $\psi(S(\tau)) = \max(S(\tau) - K_1, 0) + \max(S(\tau) - K_2, 0) - 2\max(S(\tau) - (K_1 + K_2)/2, 0)$.

In addition to the expected payoff, insurance companies are interested in computing the expected loss of their portfolios if a shock would occur in the economy. We follow the setting in Alfonsi et al. [2021, 2022] assuming that a shock occur at time $\eta$, at which time the option price is $S(\eta) = \theta$, and this shock multiplies the option price by $1 + s$. The option price at maturity time $\zeta$ is denoted as $S(\zeta) = x$. The expected loss caused by the shock can be expressed as

$$\mathcal{L} = \mathbb{E}[\max(I(\theta), 0)], \quad I(\theta) = \int_0^\infty \psi(x) - \psi\left((1 + s)x\right) \mathbb{P}_\theta(dx)$$

So the integrand is $f(x) = \psi(x) - \psi((1 + s)x)$.

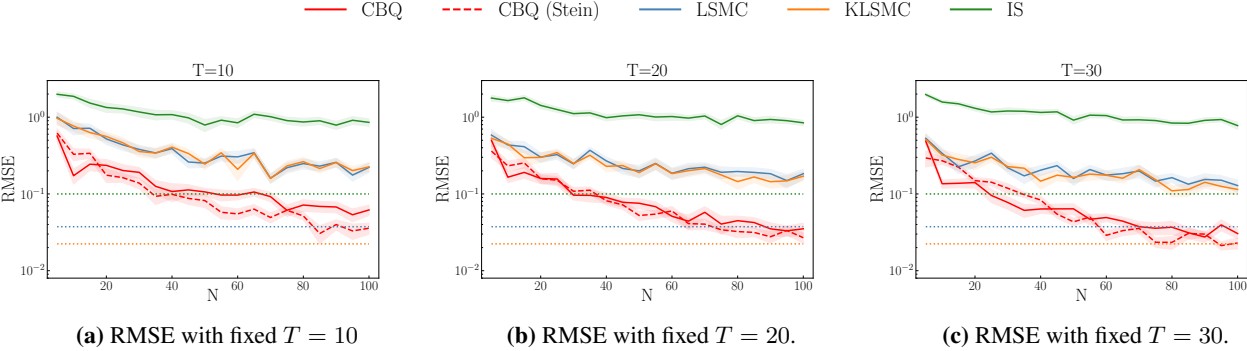

**(a)** RMSE with fixed $T = 10$       **(b)** RMSE with fixed $T = 20$.       **(c)** RMSE with fixed $T = 30$.

**Figure 10:** *Option pricing in mathematical finance.* $T = 10, 20, 30$ *and increasing* $N$.

Following the setting in Alfonsi et al. [2021, 2022], we consider the initial price $S_0 = 100$, the volatility $\sigma = 0.3$, the strikes $K_1 = 50, K_2 = 150$, the option maturity $\zeta = 2$ and the shock happens at $\eta = 1$ with strength $s = 0.2$. The option price at which the shock occurs are $\theta_{1:T}$ sampled from the log normal distribution deduced from the Black-Scholes formula $\theta_{1:T} \sim \mathbb{Q} = \text{Lognormal}(\log S_0 - \frac{\sigma^2}{2}\eta, \sigma^2\eta)$. Then $x_{1:N}^t$ are sampled from another log normal distribution also deduced from the Black-Scholes formula $x_{1:N}^t \sim \mathbb{P}_{\theta_t} = \text{Lognormal}(\log \theta_t - \frac{\sigma^2}{2}(\zeta - \eta), \sigma^2(\zeta - \eta))$.

For CBQ, we need to carefully choose two kernels $k_{\mathcal{X}}$ and $k_{\Theta}$. First we choose the kernel $k_{\mathcal{X}}$ to be a log-Gaussian kernel for the purpose that the log-Gaussian kernel mean embedding has a closed form under log-normal distribution $\mathbb{P}_\theta = \text{Lognormal}(\bar{m}, \bar{\sigma}^2)$ with $\bar{m} = \log \theta - \frac{\sigma^2}{2}(\zeta - \eta)$ and $\bar{\sigma}^2 = \sigma^2(\zeta - \eta)$. The log Gaussian kernel is defined as $k_{\mathcal{X}}(x, x') = A_{\mathcal{X}} \exp(-\frac{1}{2l_{\mathcal{X}}^2}(\log x - \log x')^2)$ and the kernel mean embedding has the form

$$\mu_\theta(x) = \frac{A_{\mathcal{X}}}{\sqrt{1 + \frac{\bar{\sigma}^2}{l_{\mathcal{X}}^2}}} \exp\left(-\frac{\bar{m}^2 + (\log x)^2}{2(\bar{\sigma}^2 + l_{\mathcal{X}}^2)}\right) x^{\frac{\bar{m}}{\bar{\sigma}^2 + l_{\mathcal{X}}^2}}$$

The initial error, which is the integral of kernel mean $\mu_\theta(x)$ does not have a closed form expression, so we use the empirical average as an approximation. Then, we choose the kernel $k_{\Theta}$ to be a Matérn-3/2 kernel.

For this experiment, we also implement CBQ with Langevin Stein reproducing kernel. We use Matérn-3/2 as the base kernel and then apply the Langevin Stein operator to both arguments of the base kernel to obtain $k_{\mathcal{X}}$. The reason we use a Stein kernel is that Stein kernels have an RKHS whose functions have one order less smoothness than the base kernel, and since the integrand has very low smoothness (due to the maximum function), we do not want to use an overly smooth kernel. The kernel mean embedding of a Stein kernel is a constant $c$ by construction as per the discussion in Appendix B. The kernel $k_{\Theta}$ is selected as Matérn-3/2 kernel. All hyperparameters in $k_{\mathcal{X}}$ and $k_{\Theta}$ for CBQ and hyperparameters for baseline methods are selected according to Appendix B.2.

### C.3.2 Assumptions from Theorem 1

We would like to check whether the assumptions made in Theorem 1 hold in this experiment.

- A1: Although $\mathcal{X} = \mathbb{R}^+$ is not a compact domain, $\mathbb{P}_\theta$ is a lognormal distribution so the probability mass outside a large compact subset of $\mathcal{X}$ decays super exponentially. A similar argument can be made for $\Theta$ as well. A1 is therefore approximately satisfied.

- A2: A2 is satisfied due to the sampling mechanism of $\theta_{1:T}$ and $\{x_{1:N}^t\}_{t=1}^T$.

- A3: $\mathbb{Q}$ is a lognormal distribution so its density $q$ is upper bounded and strictly positive within a large compact subset of $\Theta$. $\mathbb{P}_\theta$ is also a lognormal distribution so its density $p_\theta$ is strictly positive within a large compact subset of $\mathcal{X}$ and has finite second moment. A3 is approximately satisfied.

- A4: $f(x)$ is a combination of piecewise linear functions so $s_f = 1$ and $I(\theta)$ is infinitely times differentiable so $s_f = \infty$. When $k_{\mathcal{X}}$ is Stein kernel with Matern-3/2 kernel as the base, the functions in the corresponding RKHS have smoothness 1/2, whereas when $k_{\mathcal{X}}$ is the log Gaussian kernel, the functions are infinitely differentiable. Neither of these choices

satisfy the assumption, although Stein kernel contain many (but not necessarily all) function of smoothness $1/2$. $k_\Theta$ is Matern-3/2 kernel so $s_\Theta = \frac{3}{2} + \frac{1}{2} = 2$. It is therefore unclear if A4 is satisfied.

- A5: $\lambda_\mathcal{X}$ is picked to be 0 and $\lambda_\Theta$ is found via grid search among $\{0.01, 0.1, 1.0\}$. A5 is satisfied.

### C.3.3 More Experimental Results

We report more results in Figure 10 with fixed $T = 10, 20, 30$ and increasing $N$, to showcase that CBQ consistently exhibits smaller RMSE than baseline methods. The conclusions that we draw from the main text also hold for different values of $N$ and $T$ for this experiment. The performance of CBQ is similar between $k_\mathcal{X}$ being Stein kernel and $k_\mathcal{X}$ being log Gaussian kernel. It would be interesting to further investigate the performance of CBQ in estimating the future price of other financial derivatives, and we leave it for future work.

### C.4 UNCERTAINTY DECISION MAKING IN HEALTH ECONOMICS

#### C.4.1 Experimental Settings

In the medical world, it is important to compare the cost and the relative advantages of conducting extra medical experiments. The expected value of partial perfect information (EVPPI) quantifies the expected gain from conducting extra experiments to obtain precise knowledge of some unknown variables [Brennan et al., 2007]:

$$\text{EVPPI} = \mathbb{E}\Big[\max_c I_c(\theta)\Big] - \max_c \mathbb{E}\Big[I_c(\theta)\Big], \ I_c(\theta) = \int_\mathcal{X} f_c(x, \theta)\mathbb{P}_\theta(dx)$$

where $c \in \mathcal{C}$ is a set of potential treatments and $f_c$ measures the potential outcome of treatment $c$. Our method is applicable for estimating the conditional expectation $I_c(\theta)$ of the first term.

We adopt the same experimental setup as delineated in Giles and Goda [2019], wherein $x$ and $\theta$ have a joint 19-dimensional Gaussian distribution, meaning that $\mathbb{P}_\theta$ is a Gaussian distribution. The specific meanings of all $x$ and $\theta$ are outlined in Table 1. All these variables are independent except that $\theta_1, \theta_2, x_6, x_{14}$ are pairwise correlated with a correlation coefficient 0.6. The observations $\theta_{1:T}$ are sampled from the marginal Gaussian distribution $\mathbb{Q}$ and then $N$ observations of $x_{1:N}^t$ are sampled from $\mathbb{P}_{\theta_t}$.

We are interested in a binary decision-making problem ($\mathcal{C} = \{1, 2\}$) with $f_1(x, \theta) = 10^4(\theta_1 x_5 x_6 + x_7 x_8 x_9) - (x_1 + x_2 x_3 x_4)$ and $f_2(x, \theta) = 10^4(\theta_2 x_{13} x_{14} + x_{15} x_{16} x_{17}) - (x_{10} + x_{11} x_{12} x_4)$. In computing EVPPI, we estimate $I_c(\theta)$ with CBQ and baselines, and then use standard MC for the rest of the expectations. We draw $10^6$ samples from the joint distribution to generate a pseudo ground truth, and evaluate the RMSE across different methods. Note that IS is no longer applicable here because $f_c$ now depends on both $x$ and $\theta$, so we only comparing CBQ against KLSMC and LSMC.

For CBQ, we need to carefully choose two kernels. First, we take $k_\mathcal{X}$ to be a Matérn-3/2 to ensure that the kernel mean embedding under a Gaussian distribution $\mathbb{P}_\theta = \mathcal{N}(\tilde{\mu}, \tilde{\Sigma})$ has a closed form if we use the 'inverse transform trick' as outlined in Appendix B. Specifically speaking, we initially sample $u$ from $\mathcal{N}(0, \text{Id}_d)$, then calculate $x = \tilde{m} + L^\top u$ where $L$ is the lower triangular matrix derived from the Cholesky decomposition of the covariance matrix $\tilde{\Sigma}$. The integral now becomes

$$I_c(\theta) = \int_{\mathbb{R}^d} f(x)\mathcal{N}(x; \tilde{m}, \tilde{\Sigma})dx = \int_{\mathbb{R}^d} f(\tilde{m} + L^\top u)\mathcal{N}(u; 0, \text{Id}_d)du \tag{C.12}$$

The closed form expression of kernel mean embedding for a Matérn-3/2 kernel and isotropic Gaussian can be found in the Appendix S.3 of Ming and Guillas [2021]. Then we pick $k_\Theta$. We know there is a high chance that $I_c(\theta)$ is infinitely times differentiable, but we opt for Matérn-3/2 kernel to encode a more conservative prior information on the smoothness of $I_c(\theta)$ because we do not have a closed form of it. All hyperparameters in $k_\mathcal{X}$ and $k_\Theta$ are selected according to Appendix B.2.

#### C.4.2 Assumptions from Theorem 1

We would like to check whether the assumptions made in Theorem 1 hold in this experiment.

- A1: Although $\mathcal{X} = \mathbb{R}$ is not a compact domain, but $\mathbb{P}_\theta$ is a Gaussian distribution so the probability mass outside a large compact subset of $\mathcal{X}$ decays exponentially. Similarly, $\Theta = \mathbb{R}$ is not a compact domain, but $\mathbb{Q}$ is a Gaussian distribution so the probability mass outside a large compact subset of $\Theta$ decays exponentially. A1 is approximately satisfied.

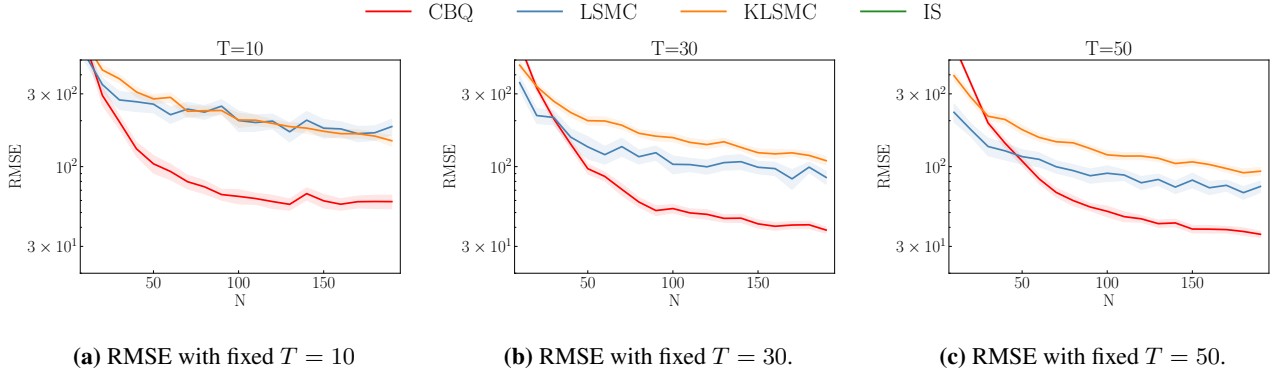

**(a)** RMSE with fixed $T = 10$

**(b)** RMSE with fixed $T = 30$.

**(c)** RMSE with fixed $T = 50$.

**Figure 11:** *Uncertainty decision making in health economics.* $T = 10, 30, 50$ and increasing $N$.

| Variables | Mean | Std | Meaning |
|:---:|:---:|:---:|:---:|
| $X_1$ | 1000 | 1.0 | Cost of treatment |
| $X_2$ | 0.1 | 0.02 | Probability of admissions |
| $X_3$ | 5.2 | 1.0 | Days of hospital |
| $X_4$ | 400 | 200 | Cost per day |
| $X_5$ | 0.3 | 0.1 | Utility change if response |
| $X_6$ | 3.0 | 0.5 | Duration of response |
| $X_7$ | 0.25 | 0.1 | Probability of side effects |
| $X_8$ | -0.1 | 0.02 | Change in utility if side effect |
| $X_9$ | 0.5 | 0.2 | Duration of side effects |
| $X_{10}$ | 1500 | 1.0 | Cost of treatment |
| $X_{11}$ | 0.08 | 0.02 | Probability of admissions |
| $X_{12}$ | 6.1 | 1.0 | Days of hospital |
| $X_{13}$ | 0.3 | 0.05 | Utility change if response |
| $X_{14}$ | 3.0 | 1.0 | Duration of response |
| $X_{15}$ | 0.2 | 0.05 | Probability of side effects |
| $X_{16}$ | -0.1 | 0.02 | Change in utility if side effect |
| $X_{17}$ | 0.5 | 0.2 | Duration of side effects |
| $\theta_1$ | 0.7 | 0.1 | Probability of responding |
| $\theta_2$ | 0.8 | 0.1 | Probability of responding |

**Table 1:** Variables in the health economics experiment.

- A2: A2 is satisfied due to the sampling mechanism of $\theta_{1:T}$ and $\{x_{1:N}^t\}_{t=1}^T$.

- A3: $\mathbb{Q}$ is also a Gaussian distribution so its density $q$ is upper bounded and strictly positive on a compact and large domain. $\mathbb{P}_\theta$ is a Gaussian distribution so its density $p_\theta$ is strictly positive on a compact and large domain with finite second moment. A3 is approximately satisfied.

- A4: Both the integrand $f$ and the conditional expectation $I_c(\theta)$ are infinitely times differentiable, so $s_f = s_I = \infty$. On the other hand, due to the choice of Matérn-3/2 kernels, $s_\Theta = 3/2 + 1/2 = 2$ and $s_\mathcal{X} = 3/2 + 9/2 = 6$. A4 is therefore satisfied.

- A5: $\lambda_\mathcal{X}$ is picked to be 0 and $\lambda_\Theta$ is found via grid search among $\{0.01, 0.1, 1.0\}$. A5 is satisfied.

### C.4.3 Additional Experimental Results

We report more results in Figure 11 with fixed $T = 10, 30, 50$ and increasing N, to showcase that CBQ consistently exhibits smaller RMSE than baseline methods. The conclusions that we draw from the main text also hold for different values of $N$ and $T$ for this experiment.

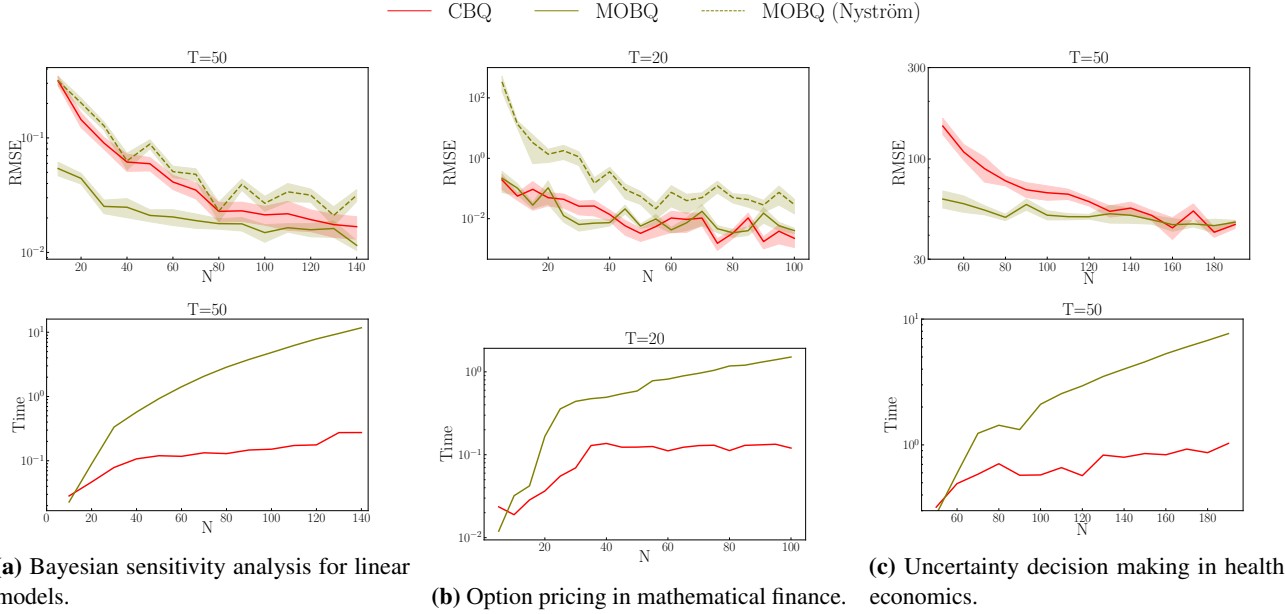

**(a)** Bayesian sensitivity analysis for linear models.

**(b)** Option pricing in mathematical finance.

**(c)** Uncertainty decision making in health economics.

**Figure 12:** Comparison of CBQ and MOBQ in terms of RMSE (first row) and computational time (second row). **Left (a):** Bayesian sensitivity analysis for linear models. **Middle (b):** Option pricing in mathematical finance. **Right (c):** Uncertainty decision making in health economics.

## C.5 COMPARISON OF CONDITIONAL BAYESIAN QUADRATURE AND MULTI-OUTPUT BAYESIAN QUADRATURE

In Section 3 in the main text, we mentioned a comparison of CBQ and multi-output Bayesian quadrature Xi et al. [2018] (MOBQ) in terms of their computational complexity. For $T$ parameter values $\theta_1, \cdots, \theta_T$ and $N$ samples from each probability distribution $\mathbb{P}_{\theta_1}, \ldots, \mathbb{P}_{\theta_T}$, the computational cost is $\mathcal{O}(TN^3 + T^3)$ for CBQ and $\mathcal{O}(N^3T^3)$ for MOBQ. We now give a more thorough comparison of CBQ and MOBQ in this section.

When the integrand $f$ only depends on $x$ (Bayesian sensitivity analysis for linear models, option pricing in mathematical finance), MOBQ only requires one kernel $k_{\mathcal{X}}$.

$$I_{\mathrm{MOBQ}}(\theta^*) = \left( \int_{\mathcal{X}} k_{\mathcal{X}}(x, x_{1:NT}) \mathbb{P}_{\theta^*}(dx) \right) \left( k_{\mathcal{X}}(x_{1:NT}, x_{1:NT}) + \lambda_{\mathcal{X}} \mathrm{Id}_{NT} \right)^{-1} f(x_{1:NT})$$

where $x_{1:NT} \in \mathbb{R}^{NT}$ is a concatenation of $x_{1:N}^1, \cdots, x_{1:N}^T$. When the integrand $f$ depends on both $x$ and $\theta$ (uncertainty decision making in health economics), MOBQ requires two kernels $k_{\mathcal{X}}$ and $k_{\Theta}$.

$$I_{\mathrm{MOBQ}}(\theta^*) = \left( \int_{\mathcal{X}} k_{\mathcal{X}}(x, x_{1:NT}) \odot k_{\Theta}(\theta^*, \theta_{1:NT}) \mathbb{P}_{\theta^*}(dx) \right)$$
$$\left( k_{\mathcal{X}}(x_{1:NT}, x_{1:NT}) \odot k_{\Theta}(\theta_{1:NT}, \theta_{1:NT}) + \lambda_{\mathcal{X}} \mathrm{Id}_{NT} \right)^{-1} f(x_{1:NT})$$

where $\odot$ denotes element-wise product, and $\theta_{1:NT} = [\theta_1, \cdots, \theta_1, \cdots, \theta_T, \cdots, \theta_T] \in \mathbb{R}^{NT}$. From the above two equations, we can see that the computation cost of $\mathcal{O}(N^3T^3)$ mainly comes from the inversion of a $NT \times NT$ kernel matrix. All the MOBQ hyperparameters in $k_{\mathcal{X}}$ and $k_{\Theta}$ are selected by empirical Bayes in the same way as CBQ outlined in Appendix B.2. It's crucial to note that the MOBQ computational cost is significantly higher for Stein reproducing kernel during hyperparameter selection (an approach analogous to the "vector-valued control variates" of Sun et al. [2023a]), as evaluating the log marginal likelihood at every iteration would require the inversion of a $NT \times NT$ matrix. Therefore, we do not include the experiment of Bayesian sensitivity analysis for the SIR model in this section. All the hyperparameters for CBQ are reused as in Appendix C.

For Bayesian sensitivity analysis in linear models, the integrand is $f(x) = x^\top x$, the dimension is fixed $d = 2$ and $T = 50$. In Figure 12a, we can see that MOBQ indeed achieves lower RMSE at the beginning, but CBQ catches up when $N$ grows

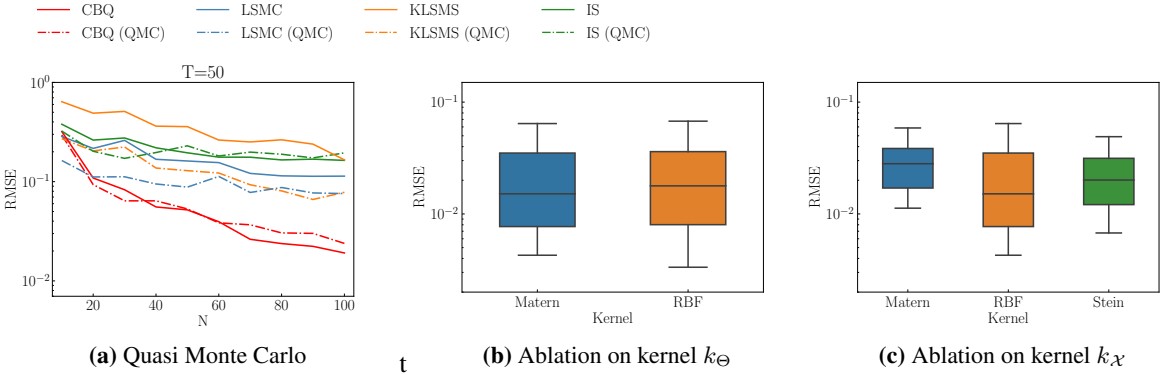

**(a)** Quasi Monte Carlo    t    **(b)** Ablation on kernel $k_\Theta$    **(c)** Ablation on kernel $k_{\mathcal{X}}$

Figure 13: **Left:** Comparison of all methods with standard i.i.d. sampling and Quasi-Monte Carlo samples. **Middle and Right:** Ablation study for CBQ with different $k_\Theta$ and $k_{\mathcal{X}}$ kernels in Bayesian sensitivity analysis for linear models.

higher. For option pricing in mathematical finance, we only compare MOBQ and CBQ when $k_{\mathcal{X}}$ is the log Gaussian kernel and $T = 20$. For uncertainty decision making in health economics, we compare MOBQ and CBQ when $T = 50$. In Figure 12b and Figure 12c, we can see that CBQ and MOBQ achieves similar performances in terms of RMSE. Additionally, in the second row of Appendix C.4.3, we compare the computational cost of MOBQ and CBQ, where we can see that the computational time of MOBQ is much larger than CBQ as $N$ grows across all settings, due to the complexity of $\mathcal{O}(N^3 T^3)$ for MOBQ.

Additionally, as the main computational bottleneck of MOBQ is the inversion of the kernel matrix, so it would be interesting to see if MOBQ combined with scalable GP methods can reduce the computational time while still preserving the same level of accuracy. The scalable approximation method used here is Nyström approximation Williams and Seeger [2000]. We report the performance of MOBQ (Nyström) in both Figure 12a and Figure 12b, and we can see that MOBQ (Nyström) performs worse than CBQ in terms of RMSE. The reason of worse performance of MOBQ (Nyström) is that the use of scalable GP methods would introduce an extra layer of approximation that slows down the convergence rate. Additionally, most scalable GP methods are used in the "regression" setting, while quadrature methods like BQ or CBQ belong to the "interpolation" setting Kanagawa et al. [2018], so the quadrature problem will be more sensitive to the approximation error introduced.

## C.6 QUASI MONTE CARLO

Quasi Monte Carlo (QMC) is another line of research on improving the precision of approximating intractable integrals. While quadrature methods like BQ and CBQ aim at finding a smart way to combine the function values, QMC aims to find samples that can more uniformly cover the integration domain than random sampling [Niu et al., 2023, Hickernell, 1998, Gerber and Chopin, 2015]. In the development of CBQ, we don't make any assumptions about the sampling of observations; specifically, we don't mandate i.i.d sampling. Therefore, it would be interesting to see whether combining quadrature algorithms with QMC could further improve the accuracy for estimating conditional expectation.

For a fair comparison in the experiment of Bayesian sensitivity analysis for linear models, we implement QMC sampling for all methods including CBQ and baseline methods. The samples $x_{1:N}^t$ are generated from a Sobol sequence which is a low-discrepancy sequence commonly used in QMC to cover the multidimensional space more uniformly than random sequences. We follow the technique introduced in randomized QMC Lemieux [2004] to shift the Sobol sequence by a random amount.

It can be observed in Figure 13a that replacing random sampling with QMC significantly enhances the performance of baseline methods, such as LSMC and KLSMC, while subtly improves the performance of CBQ. The limited degree of improvement seen in CBQ with QMC sampling can be attributed to the fact that CBQ already yields a remarkably low RMSE. Consequently, the margin of improvement offered by QMC sampling is not as evident in CBQ as in the baseline methods. We have only studied the effect of combining QMC and CBQ in the experiment of Bayesian sensitivity analysis in linear models. It would be interesting to see if combining QMC and CBQ would result in higher accuracy in other settings, and we leave it for future work.

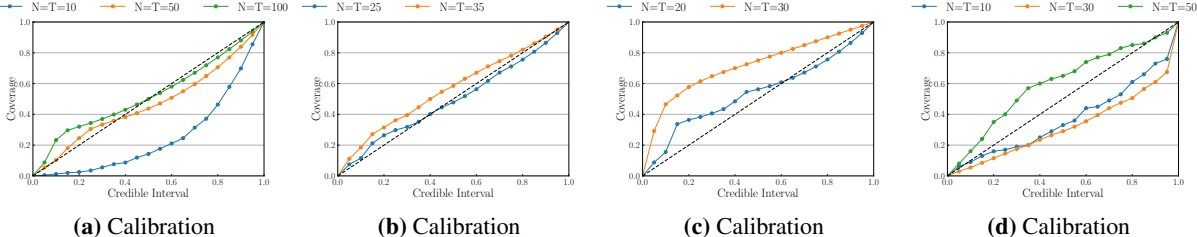

**(a)** Calibration  **(b)** Calibration  **(c)** Calibration  **(d)** Calibration

**Figure 14:** Calibration plots. **Top Left:** Bayesian sensitivity analysis in linear models. **Top Right:** Bayesian sensitivity analysis for SIR model. **Bottom Left:** Option pricing in mathematical finance. **Bottom Right:** Uncertainty decision making in health economics.

### C.7  ABLATIONS ON KERNELS

We present an ablation study evaluating the impact of distinct kernel choices $k_\mathcal{X}$ and $k_\Theta$ within the framework of Bayesian sensitivity analysis in linear models. The integrand is $f(x) = x^\top x$, the dimension $d = 2$ and $N = T = 50$. First, we choose $k_\Theta$ to be Matérn-3/2 kernel and Gaussian kernel. Figure 13b shows that the performance of CBQ remains consistent across different $k_\Theta$ kernels.

Subsequently, we opt for Matérn-3/2 kernel, Gaussian kernel and Stein kernel (with Matérn-3/2 as the base kernel) as choices for $k_\mathcal{X}$. When $k_\mathcal{X}$ is Gaussian kernel, the formula for kernel mean embedding $\mu_\theta(x)$ is presented in Equation (C.11). When $k_\mathcal{X}$ is Matérn-3/2 kernel, a closed form expression for the kernel mean embedding does not exist for the non-isotropic Gaussian distribution $\mathcal{N}(\tilde{m}, \tilde{\Sigma})$, but the 'inverse transform trick' can be employed as in Equation (C.12). When $k_\mathcal{X}$ is Stein kernel, we choose Matérn-3/2 as the base kernel and then apply Stein operator on both arguments of kernel $k_0$. All hyperparameters are selected according to Appendix B.2. From Figure 13c, we can see that CBQ performs best when $k_\mathcal{X}$ is Matérn-3/2 kernel, and we know that $k_\mathcal{X}$ being Matérn-3/2 kernel satisfies the assumptions of Theorem 1. When $k_\mathcal{X}$ is Gaussian RBF kernel or Stein kernel, whether the assumptions of Theorem 1 still hold is unknown, but in this ablation study, CBQ under both kernels have shown good performances in terms of RMSE. The ablation study is only implemented in this very simple setting, so we encourage practitioners to be careful in the selection of kernels in real world applications.

### C.8  CALIBRATION

CBQ falls in the area of probabilistic numeric algorithms that can provide finite-sample Bayesian quantification of uncertainty, where the uncertainty arises from having access to only a finite number of function values of the integrand. Since CBQ is a two-stage hierarchical Gaussian process method in nature, and the final estimate $I_{\text{CBQ}}$ is treated as Gaussian distributed, so the standard deviation $\sigma^2_{\text{CBQ}}$ is a measure of uncertainty Kendall and Gal [2017]. The calibration plots in Figure 14 are obtained by altering the width of the credible interval and then computes the percentage of times a credible interval contains the true value $I(\theta)$ under repetitions of the experiment. The black diagonal line represents the ideal case, with any curve lying above the black line indicating underconfidence and any curve lying below indicating overconfidence. It is generally regarded more preferable to be underconfident than overconfident.

In Figure 14a, we show the calibration of the CBQ posterior for the integrand $f(x) = x^\top x$ when dimension $d = 2$. We observe that when the number of samples is as small as 10, CBQ is overconfident, which can be explained by the poor performance of using empirical Bayes to select hyperparameters in the small sample regime. On the other hand, when $N$ and $T$ increase, CBQ becomes underconfident, meaning that our posterior variance is more inflated than needed from a frequentist viewpoint. The calibration plots for other experiments are all demonstrated in Figure 14, and the conclusions are consistent across different experiments.