# OpenReview forum: "Conditional Bayesian Quadrature"
_auai.org/UAI/2024/Conference — UAI 2024 poster_

### Official Review · Reviewer_ZqG2 · 2024-03-20

**Q2-1 Originality-Novelty:** 3
**Q2-2 Correctness-Technical Quality:** 3
**Q2-5 Clarity Of Writing:** 4

**Q1 Summary And Contributions:**

The paper proposes "conditional Bayesian quadrature", a method for estimation conditional expectations through probabilistic numerical methods. The authors propose a two-state procedure that first computes expectations for each parameter separately using Gaussian process (GP) regression models, and then uses the estimated posterior means and variances to build a second GP model allowing to, e.g., "interpolate" to unseen parameter values. The authors demonstrate that their method convincingly outperforms other state-of-the-art methods.

**Q2-3 Extent To Which Claims Are Supported By Evidence:**

3: Good: the main claims are supported by convincing evidence (in the form of adequate experimental evaluation, proofs, (pseudo-)code, references, assumptions).

**Q2-4 Reproducibility:**

4: Excellent: key resources (e.g. proofs, code, data) are available and key details (e.g. proof sketches, experimental setup) are comprehensively described for competent researchers to confidently and easily reproduce the main results.

**Q3 Main Strengths:**

- The paper is very well written.
- The proposed method is convincing and outperforms state-of-the-art competitors.
- The method is very intuitive. It importantly considers correlations in theta in the second stage. The idea to incorporate the uncertainty estimate for the variance in the second stage is, in my opinion, well thought through.

**Q4 Main Weakness:**

- The novelty of the method seems a bit limited and incremental.
- While the experimental section is in my opinion sufficient enough, it could benefit from more experiments.
- The authors state that their method still has a computational cost of $O(TN^3 + T^3)$ which seems prohibitive. Are there approaches how this could be reduced, e.g., using Bayesian neural networks which also yield credible intervals for mean and variance?

**Q5 Detailed Comments To The Authors:**

--

**Q9 Complying With Reviewing Instructions:**

Yes

---

> ### Author Rebuttal · Authors · 2024-04-08
>
> Thank you for taking the time to consider our paper, and for your kind words.
>
> ## 1. The novelty of the method seems a bit limited and incremental.
> We actually see the fact that the method is natural as a significant strength, and we hope this will encourage potential users to adopt it. In addition, our paper provides a thorough theoretically and empirically study of the algorithm demonstrating that it can lead to much more accurate estimates compared to baselines.
>
> ## 2. While the experimental section is in my opinion sufficient enough, it could benefit from more experiments.
> Thanks for the suggestion. We have already provided experiments for four different problems, and also provided extensive additional results in the appendix. We would be happy to provide more, but it is unclear which other results would actually enhance the paper given the limited page limit.
>
> ## 3. The method still has a computational cost of $O\left(T N^3+T^3\right)$ which seems prohibitive. Using Bayesian neural networks?
>
> Please see our response to Reviewer fkqH regarding computational complexity.
>
> Bayesian neural networks (BNN) can indeed be used as a replacement for GP in Bayesian quadrature to provide uncertainty estimates for the integral, see for example https://arxiv.org/pdf/2305.13248.pdf. These do not however benefit from strong theoretical results. We will add more discussion in the paper on how BNN can be potentially used in our setting to estimate conditional expectations.

---

### Official Review · Reviewer_a8Md · 2024-03-21

**Q2-1 Originality-Novelty:** 3
**Q2-2 Correctness-Technical Quality:** 3
**Q2-5 Clarity Of Writing:** 4

**Q1 Summary And Contributions:**

This paper proposes a new approach for estimating conditional expectation in the setting where obtaining samples or evaluating integrands is costly. The proposed method extends Bayesian Quadrature method, which is ued to estimate unconditional expectation, to estimate conditional expectation. The key component of the method is a two-stage hiearchical Gaussian process. The authors showed theoretical guarantees on the convergence of the proposed method under certain technical conditions. Besides, the authors also implemented various numerical experiments to empirically support their proposed method.

**Q2-3 Extent To Which Claims Are Supported By Evidence:**

3: Good: the main claims are supported by convincing evidence (in the form of adequate experimental evaluation, proofs, (pseudo-)code, references, assumptions).

**Q2-4 Reproducibility:**

4: Excellent: key resources (e.g. proofs, code, data) are available and key details (e.g. proof sketches, experimental setup) are comprehensively described for competent researchers to confidently and easily reproduce the main results.

**Q3 Main Strengths:**

1. The key idea of extending BQ method to estimate conditional expectation by a two-stage hiearchical Gaussian process seems novel. Although it seems natural and not technically complicated, it is very intuitive and clean. For this reason, I appreciate the proposed method.

2. The authors provided theoretical convergence guarantees on the proposed method. The technical assumptions of the theorem are carefully discussed, both on whether they are practical and on ways to potentially relax them. Besides, the authors also checked whether those conditions can be satisfied in their numerical experiments.

3. The authors implemented extensive numerical experiments to compare their method with several baseline methods. The code of the paper is made available for checking, though I did not check the code myself.

4. The writing of the paper is clear and easy to follow.

**Q4 Main Weakness:**

1. The key idea may not be technically hard to construct.

2. Like demonstrated by the authors themsevels, the proposed method is restrictive to low-dimensional applications.

3. The proposed method can be computationally expensive.

**Q5 Detailed Comments To The Authors:**

The authors may consider addressing aformentioned problems in Q4. While problem 2 and 3 might be intrinsic problems of GP and can be hard to address, the authors can emphasize the main technical challenges to help readers better appreciate the paper's technical contribution.

**Q9 Complying With Reviewing Instructions:**

Yes

---

> ### Author Rebuttal · Authors · 2024-04-08
>
> Thank you for taking the time to consider our paper, and for your very strong backing.
>
> ## 1. The key idea may not be technically hard to construct.
> We actually see this as a significant strength. Indeed, we hope the fact that this approach is natural will encourage potential users to adopt it.
>
> ## 2. The proposed method is restrictive to low-dimensional applications. The proposed method can be computationally expensive.
> Indeed, this is a common limitation of these methods which we acknowledge in the paper. There have been some attempts to scale BQ to high dimensions; see section 5.4 of [Briol et al. 2019] where the integrand can be decomposed into a sum of low-dimensional functions. This is however only doable in limited settings. We will discuss this point further in the camera-ready version of the paper.
>
> The computational complexity of CBQ is $\mathcal{O}\left(T N^3+T^3\right)$, which is higher than alternative methods such as LSMC and KLSMC. High computational cost is a common issue for all GP based methods due to the inversion of the Gram matrix. However, the higher cost of CBQ will be offset by faster convergence, shown by our theory and experiments. And in many applications (like the SIR model), the cost of evaluating the integrand will be much larger than the matrix inversion cost, so CBQ will be more efficient overall.
>
> In the camera-ready version of the paper, we are happy to put more discussion on the technical challenges including both limitations to low-dimensional inputs and high computational complexity to help readers better appreciate the paper's technical contributions.

---

### Official Review · Reviewer_8akG · 2024-03-21

**Q2-1 Originality-Novelty:** 3
**Q2-2 Correctness-Technical Quality:** 3
**Q2-5 Clarity Of Writing:** 3

**Q1 Summary And Contributions:**

This paper proposes Conditional Bayesian Quadrature (CBQ), a method for computing *conditional* expectations ($I(\Theta) = E_{x \sim P_\Theta } [f(x, \Theta)]$) based on standard BQ.
Being fully Bayesian, CBQ has the advantage of quantifying uncertainty on the estimator. Compared to sampling or regression-based approaches to the problem, the proposed approach is more computationally expensive for a fixed number of samples, but exhibits faster convergence rate. Therefore, CBQ should be preferable whenever $f$ is low-dimensional but expensive to evaluate.

**Q2-3 Extent To Which Claims Are Supported By Evidence:**

3: Good: the main claims are supported by convincing evidence (in the form of adequate experimental evaluation, proofs, (pseudo-)code, references, assumptions).

**Q2-4 Reproducibility:**

3: Good: key resources (e.g. proofs, code, data) are available and key details (e.g. proofs, experimental setup) are sufficiently well-described for competent researchers to confidently reproduce the main results.

**Q3 Main Strengths:**

- The overall presentation and writing is good overall. The motivation and problem statement is clear, CBQ is well-contextualized with respect to the relevant literature. I appreciate that the shortcomings of the proposed approach are also clearly stated.
- Theoretical bounds on the convergence rate are formally derived.
- The empirical evaluation is solid.

**Q4 Main Weakness:**

- The paper is quite technical and not very accessible to non-experts in BQ, in my opinion.

**Q5 Detailed Comments To The Authors:**

***Minors***

"The usual approach is to use classical Monte Carlo methods ... but in many applications we are also interested in estimating either $I$ for a fixed $\Theta$ ... As a result, a second step combining the estimates ... is often required to complete the task."

Why $T$ instantiations need to be combined in this case when evaluating for a single $\Theta$?


---

Fig.1 Right should have axis labels and a more informative caption. Also, I missed the reference in the main text.

---

The formal definition of the space of square-integrable functions is recursive (possibly a typo?).

---

"and we can only use $N$ rather than $NT$ points to estimate each $I (\Theta_t )$"

I would rephrase this. Why can't we use as many samples as we want in each estimation $I (\Theta_t )$?


---

"but alternatives beyond this parametric family of distributions could also be used "

Which parametric family?

---

"RKHS" undefined.

**Q9 Complying With Reviewing Instructions:**

Yes

---

> ### Author Rebuttal · Authors · 2024-04-08
>
> Thank you for taking the time to consider our paper, and for your very strong backing.
> ## 1. The paper is quite technical and not very accessible to non-experts in BQ.
> We are happy to provide more guidance for non BQ experts given the extra space we have in the camera-ready version.
>
> ## 2. Why $T$ instantiations need to be combined in this case when evaluating for a single $\theta$?
> By using only 1 instantiation of $\theta$ is wasteful by throwing away much information on the integrand if the $T$ instantiations are related to each other. Our CBQ shows both theoretically and empirically that combining all instantiations results in higher accuracy.
>
> ## 3. Why can't we use as many samples as we want in each estimation $I(\theta_t)$?
> In the context of the standard Monte Carlo estimate of $I(\\theta_t)=\\int f(x, \\theta_t)dP_{\\theta_t}(x)$, it is only allowed to use $N$ samples from $P_{\theta_t}$ because other $NT - N$ samples are from a different distribution.
>
> ## 4. Which parametric family?
> The parametric family is $\\{P_\theta: \theta \in \Theta \\}$. For importance sampling, the importance distribution can be selected to be other distributions apart from the conditional distributions: $P_{\theta_1}, \cdots, P_{\theta_T}$.
>
> ## 5. Other minor comments
> Thank you, and we will fix these in the camera-ready version of our paper.

---

### Official Review · Reviewer_CAkf · 2024-03-22

**Q2-1 Originality-Novelty:** 3
**Q2-2 Correctness-Technical Quality:** 3
**Q2-5 Clarity Of Writing:** 3

**Q1 Summary And Contributions:**

This paper provides an approach to approximate conditional expectations of black-box functions which are expensive to evaluate. The approach is based on Gaussian processes (GPs) and the related framework of Bayesian quadrature (BQ). The proposed method applies a two-stage approach using GPs first to approximate conditional expectations pointwise and a second stage to provide estimates for the conditional expectations at points not available in the training set. Theoretical convergence guarantees are provided, and experiments demonstrate superior performance on a range of benchmark problems.

**Q2-3 Extent To Which Claims Are Supported By Evidence:**

3: Good: the main claims are supported by convincing evidence (in the form of adequate experimental evaluation, proofs, (pseudo-)code, references, assumptions).

**Q2-4 Reproducibility:**

3: Good: key resources (e.g. proofs, code, data) are available and key details (e.g. proofs, experimental setup) are sufficiently well-described for competent researchers to confidently reproduce the main results.

**Q3 Main Strengths:**

* The paper provides a detailed discussion about the architectural choices and relation to Monte Carlo methods.
* Theoretical results guarantee convergence of the method's estimates under a reasonable set of assumptions.
* Assumptions and limitations of theoretical analysis are well discussed.
* Experiments show significant performance improvements against baselines on a variety of benchmark problems with simulated data.

**Q4 Main Weakness:**

* Method depends on closed-form expressions for kernel expectations, though a few alternatives based on closed-form approximations are provided in the appendix.
* Lack of discussion and comparison against conditional kernel mean embeddings, which have an extensive literature on approximating conditional expectations.
* Experiments provide no real-data benchmark.

**Q5 Detailed Comments To The Authors:**

* **Need for closed-form expressions:** In Sec. 2.2, the need for closed-form expressions of the kernel means is pointed out as an issue with related work in BQ. However, the proposed method also requires access to closed-form expressions of the kernel mean embeddings, as mentioned in Sec. 3. Alternatives for when that is not possible are provided in the appendix, but to me it seems that the main paper should also have a brief discussion on this limitation or at least an explicit mention of the discussion in Appendix B for the alternatives. At the moment, the paragraph mentioning the requirement only mentions that pairs of kernels and distributions for what this is possible are present in the appendix, leaving the impression that the method is not applicable otherwise.
* **Comparison with conditional mean embeddings:** Have conditional kernel mean embeddings (CMEs) been considered as a baseline method? Given a set of joint samples of $x$ and $\theta$, it is possible to construct a CME estimator for conditional expectations, especially for the case where $f$ does not depend on $\theta$ directly, as in the first 3 examples. There's an extensive literature on CMEs for approximations of conditional expectations, and a detailed review is available in Muandet et al. [2017], cited in the paper. I believe at least a discussion in the related work section should be present, though ideally comparison experiments should also be included, if possible.
* **Independent estimates assumption:** In the formulation of the CBQ posterior, there's an implicit independence assumption in the estimates of $I_{\mathrm{BQ}}$ by the BQ GP, assumption which is not discussed about. Estimates of $I(\theta)$ for values of $\theta$ that are close to each other should be highly correlated, but the covariance matrix for the first-stage estimates is incorporated into the CBQ estimator as a diagonal matrix. I understand that correlations are still modelled via the second-stage kernel $k_\Theta$. My concern is whether the estimates for $\hat{I}_{BQ}$ can truly be considered as independent and if that'd lead to any unintended effect on the theoretical guarantees.
* **Experiment benchmarks:** Would it be possible to include real-data benchmarks? Have they been considered?

Minor:
* *Citations format*: A few citations in the text are incorrectly formatted. Some citations that should be parenthetical are formatted as in-text citations. Example: "... known as the kernel mean embedding Muandet et al. [2017]" in the first paragraph of the second column in page 3. And others that should be in-text are showing up as parenthetical. Example: also in the same paragraph, "Fortunately, there are multiple solutions for this problem; see Table 1 in [Briol et al., 2019], [Nishiyama and Fukumizu, 2016]".
* *RKHS methods and smoothness assumption*: In Sec. 2.1, last paragraph, "Notably, they do not explicitly utilise the property that $I(\theta)$ is a smooth function of $\theta$, and will therefore be sub-optimal for our setting." RKHS methods, as referred in the previous sentence, usually do take into account smoothness assumptions on the modelled function, which the paper's proposed method also makes use of. Therefore, this sentence is confusing and seems a bit misleading.
* The sentence "Examples range from differential equation solvers [Kersting and Hennig, 2016], variational inference [Acerbi, 2018] and simulator-based inference [Bharti et al., 2023] to..." is a bit confusing, since variational inference and simulator-based inference are not per se examples of application problems for BQ, but the mentioned papers propose methods that use some version of BQ.

**Q9 Complying With Reviewing Instructions:**

Yes

---

> ### Author Rebuttal · Authors · 2024-04-08
>
> Thank you for taking the time to consider our paper, and for your very kind comments.
> ## 1. Closed-form expressions for kernel means
> We agree that the need (typical for BQ) to have a closed-form kernel means is a limitation. This is why we introduce Stein kernels in Appendix B.1 (two of our experiments use it). We will move our discussion in Appendix B.1 to the main text so that the readers can better understand this limitation, as well as practical ways to get over it.
>
> ## 2. Comparison with conditional mean embeddings:
> We already provide this comparison in the paper, where it is presented under the name “kernel least-squares Monte Carlo (KLSMC)”. We call this KLSMC because our target audience is the Monte Carlo community and we therefore wanted to follow the terminology in this area. However, we will further clarify that these methods are in fact identical in the camera ready version.
>
> ## 3. Independent estimates assumption:
> The estimator $I_{BQ}(\theta_t)$ is a deterministic function of $\theta_t$ and $x_t^1, \cdots, x_t^n$. Since $\theta_{t}$ are independently sampled and $x_{t}^1, \cdots, x_{t}^n$ are sampled from $\mathbb{P}(x | \theta_t)$, therefore $I_{BQ}(\theta_{t})$ are also independent. The correlation between integral values (as opposed to other estimators) is indeed captured by our posterior $I_{CBQ}(\theta)$ (see Figure1).
> ## 4. Would it be possible to include real-data benchmarks?
> The SIR and option pricing experiments are both real-world experiments. The reason is that, in those areas, real world-experiments are based on expensive simulations of complex models (the ‘data’), and this is exactly what is done in our paper.
> ## 5. RKHS methods usually do take into account smoothness assumptions on the modeled function.
> Alternative approaches [Xi et al., 2018, Gessner et al., 2020, Sun et al., 2023a] aim at approximating multiple related integrals through a vector valued RKHS. Their setting is different from ours in that our integral $I$ is explicitly a function of $\theta$, but one could, in certain settings, obtain an equivalency. We therefore agree that this sentence is a bit misleading, and we will rewrite it in the camera-ready version of our paper.
> ## 6. Citations format and other minor issues.
> Thank you, and we will fix these in the camera-ready version of our paper.

---

### Official Review · Reviewer_fkqH · 2024-03-26

**Q2-1 Originality-Novelty:** 3
**Q2-2 Correctness-Technical Quality:** 4
**Q2-5 Clarity Of Writing:** 4

**Q1 Summary And Contributions:**

The authors consider the estimation of parametric expectations I(w)=E[f(X,w)] where the expectation is computed w.r.t. some P_w. In contrast to other approaches, e.g., importance sampling, the authors consider the case where not only P_w but also f depends on w. Technically, they derive a univariate Gaussian posterior distribution on I(w) whose mean and variance are parametrised by w. The authors show (theoretically and empirically) that, under relatively mild conditions, the proposed method is more sample efficient than baselines.

**Q2-3 Extent To Which Claims Are Supported By Evidence:**

4: Excellent: all claims are supported by very convincing evidence (in the form of comprehensive experimental evaluation, rigorous mathematical proofs, detailed (pseudo-)code, precise references, well-motivated and realistic assumptions) and the authors deliver what they promise.

**Q2-4 Reproducibility:**

4: Excellent: key resources (e.g. proofs, code, data) are available and key details (e.g. proof sketches, experimental setup) are comprehensively described for competent researchers to confidently and easily reproduce the main results.

**Q3 Main Strengths:**

A theoretically and practically sound method for an important problem class. The provided bound helps in practice and shows that this method outperforms competing methods.

**Q4 Main Weakness:**

The scaling of this whole class of methods is not practical in the presence of a large number of dimensions.

**Q5 Detailed Comments To The Authors:**

The paper itself is very good and I have nothing to complain about. However, re-running the option pricing experiment, it becomes clear that the proposed method is almost always the slowest, especially when we consider the Stein kernel. This is not a bad thing, since the method has theoretical guarantees and produces very good results. However, there should be a more explicit discussion of computational complexity in the paper.

**Q9 Complying With Reviewing Instructions:**

Yes

---

> ### Author Rebuttal · Authors · 2024-04-08
>
> Thank you for taking the time to consider our paper, and for your very strong backing.
>
> ## 1. Scaling in a large number of dimensions.
> Indeed, this is a common limitation of these methods which we acknowledge in the paper. There have been some attempts to scale BQ to high dimensions; see section 5.4 of [Briol et al. 2019] where the integrand can be decomposed into a sum of low-dimensional functions. This is however only doable in limited settings. We will discuss this point further in the camera-ready version of the paper.
>
> ## 2. Discussion of computational complexity.
> The computational complexity of CBQ is $\mathcal{O}\left(T N^3+T^3\right)$, which is higher than alternative methods such as LSMC and KLSMC. High computational cost is a common issue for all GP based methods due to the inversion of the Gram matrix. However, the higher cost of CBQ will be offset by faster convergence, shown by our theory and experiments. And in many applications (like the SIR model), the cost of evaluating the integrand will be much larger than the matrix inversion cost, so CBQ will be more efficient overall. We will extend our discussion of computational complexity in the camera-ready version of our paper.

---

### Meta-Review · Area_Chair_tR3e · 2024-04-17

This work proposes estimation of conditional expectations via a Bayesian quadrature-based framework.  Overall, the reviewers felt that this work presented worthwhile theoretical contributions, but were concerned that the computational complexity of the method limited it to low-dimensional problems.  I concur with this assessment, and while I agree that this is a limitation of this family of methods more generally, the authors should still address these computational issues in more detail in subsequent revisions of the paper.